# Wflow_sbm v0.7.3, a spatially distributed hydrological model: from global data to local applications

**Willem J. van Verseveld**[1], **Albrecht H. Weerts**[1,2], **Martijn Visser**[1], **Joost Buitink**[1], **Ruben O. Imhoff**[1], **Hélène Boisgontier**[1], **Laurène Bouaziz**[1], **Dirk Eilander**[1,3], **Mark Hegnauer**[1], **Corine ten Velden**[1], and **Bobby Russell**[1]

[1]Department of Inland Water Systems, Deltares, P.O. Box 177, 2600 MH Delft, the Netherlands
[2]Hydrology and Environmental Hydraulics Group, Department of Environmental Sciences, Wageningen University & Research, P.O. Box 47, 6700 AA Wageningen, the Netherlands
[3]Institute for Environmental Studies (IVM), Vrije Universiteit Amsterdam, 1081 HV Amsterdam, the Netherlands

**Correspondence:** Willem J. van Verseveld (willem.vanverseveld@deltares.nl)

**Abstract.** The wflow_sbm hydrological model, recently released by Deltares, as part of the Wflow.jl (v0.7.3) modelling framework, is being used to better understand and potentially address multiple operational and water resource planning challenges from a catchment scale to national scale to continental and global scale. Wflow.jl is a free and open-source distributed hydrological modelling framework written in the Julia programming language. The development of wflow_sbm, the model structure, equations and functionalities are described in detail, including example applications of wflow_sbm. The wflow_sbm model aims to strike a balance between low-resolution, low-complexity and high-resolution, high-complexity hydrological models. Most wflow_sbm parameters are based on physical characteristics or processes, and at the same time wflow_sbm has a runtime performance well suited for large-scale high-resolution model applications. Wflow_sbm models can be set a priori for any catchment with the Python tool HydroMT-Wflow based on globally available datasets and through the use of point-scale (pedo)transfer functions and suitable upscaling rules and generally result in a satisfactory ($0.4 \geq$ Kling–Gupta efficiency (KGE) $< 0.7$) to good (KGE $\geq 0.7$) performance for discharge a priori (without further tuning). Wflow_sbm includes relevant hydrological processes such as glacier and snow processes, evapotranspiration processes, unsaturated zone dynamics, (shallow) groundwater, and surface flow routing including lakes and reservoirs. Further planned developments include improvements on the computational efficiency and flexibility of the routing scheme, implementation of a water demand and allocation module for water resource modelling, the addition of a deep groundwater concept, and computational efficiency improvements through for example distributed computing and graphics processing unit (GPU) acceleration.

## 1 Introduction

Hydrological models have proven to be useful tools in better understanding multiple operational and water resource planning challenges including drought (e.g. Trambauer et al., 2015) and flood forecasting (e.g. Alfieri et al., 2013), the assessment of water availability (e.g. van Beek et al., 2011), and analysing the impact of food production on river systems (e.g. Jägermeyr et al., 2017). An advantage of spatially distributed (gridded) hydrological models, in contrast to spatially lumped models, is the ability to directly use the spatially varying information contained in spatial datasets for model setup, forcing and validation. High-resolution spatial datasets have become increasingly available, often at a global scale, and can be used to represent land cover, vegetation (e.g. leaf area index (LAI)) and soil properties in spatially distributed hydrological models. For example, SoilGrids provides gridded soil information at 250 m resolution globally (Hengl et al., 2017). With regard to forcing, the release of the fifth-generation ECMWF atmospheric reanalysis of the global climate (ERA5) (Hersbach et al., 2020) dataset (1959–present), with a spatial resolution of $\sim 31\,\mathrm{km} \times 31\,\mathrm{km}$ and a

temporal resolution of 1 h, and ERA5-Land with a spatial resolution of $\sim 9\,\text{km} \times 9\,\text{km}$ are worth mentioning. Recently, it has been argued that the development of a hyperresolution global hydrological model at $1\,\text{km}^2$ or finer is a "grand challenge for hydrology" and is needed to address the water problems facing society (Wood et al., 2011; Bierkens et al., 2014).

Notwithstanding the advantages of and need for (hyperresolution) spatially distributed hydrological models, parameterization of these models is not straightforward because of overparameterization and as a result overfitting (Jakeman and Hornberger, 1993; Beven, 1993, 2006). Furthermore, transferability of hydrological parameters across spatial and temporal scales is important for reducing calibration time and the application of hydrological models in ungauged or poorly gauged basins. However the impact of transferring model parameters across spatial and temporal resolutions on model performance is unequivocal, and high parameter transferability across spatial resolution may also be the result of inadequate representation of spatial variability in (large-scale) hydrological models (Melsen et al., 2016). Finally, there is the scientific debate on the "best" approach to process-based hydrological modelling leading to appropriate physical realism, especially related to model structure and model solutions (Kirchner, 2006; Clark et al., 2016, 2017).

Concerning hydrological model structure and solutions, Hrachowitz and Clark (2017) classified hydrological models along two dimensions, process complexity and spatial resolution. Hydrological models with high process complexity and spatial resolution are for example ParFlow (Kollet and Maxwell, 2006), HydroGeoSphere (Brunner and Simmons, 2012) and HYDRUS (2D/3D) (Šimůnek et al., 2008), while for example HBV (Bergström, 1992), SUPERFLEX (Fenicia et al., 2011) and FLEX-Topo (Gao et al., 2014) are characterized by low spatial resolution and low process complexity. For the high-resolution, high-complexity hydrological models, the majority of the parameters are based on physical characteristics and may be estimated directly or by upscaling from field or remotely sensed observations, depending on the model resolution. For low-resolution, low-complexity hydrological models, the majority of parameters are effective parameters at the catchment scale and calibration is required to identify parameter values. Generally, high-resolution, high-complexity hydrological models are computationally demanding, which limits their application to smaller domains or requires either a reduction in model resolution or high-performance computing resources for large-scale applications. Free and open-source spatially distributed hydrological models that require a low calibration effort (parameters based on physical characteristics) and have fast runtimes applicable to large-scale high-resolution modelling (medium complexity) are not available or have very limited availability to our knowledge.

Wflow_sbm represents a family of spatially distributed hydrological models that have the vertical hydrological simple bucket model (SBM; Vertessy and Elsenbeer, 1999) concept in common but can have different lateral concepts that control how water is routed for example over the land or river domain. This paper presents the wflow_sbm model configuration that makes use of the kinematic-wave approach for river, overland and lateral subsurface flow. It is part of the open-source distributed hydrological model platform Wflow.jl (van Verseveld et al., 2024) developed at Deltares. Wflow_sbm strikes a balance between low-resolution, low-complexity and high-resolution, high-complexity hydrological models, giving an answer to most of the aforementioned challenges. In this model, the soil part is largely based on Topog_SBM (Vertessy and Elsenbeer, 1999), with gravity-based infiltration and vertical flow through the soil column as well as capillary rise representing a simplified version of Richards' equation. Furthermore it uses a 1-D kinematic-wave approach for channel, overland and lateral subsurface flows that is similar to TOPKAPI (Benning, 1995; Todini and Ciarapica, 2002), G2G (Bell et al., 2007), 1K-DHM (Tanaka and Tachikawa, 2015) and Topog_SBM (Vertessy and Elsenbeer, 1999), as an approximation for dynamic waves and variably saturated subsurface flow (Richards' equation). Its advantage is that most wflow_sbm parameters are based on physical characteristics and at the same time wflow_sbm has a runtime performance well suited for large-scale high-resolution modelling.

Furthermore, in line with the need to improve the transparency, reproducibility and ease of setting up hydrological models (Clark et al., 2017; Knoben et al., 2021), we use the wflow plugin (HydroMT-Wflow; Eilander et al., 2022) of the HydroMT Python package (Eilander and Boisgontier, 2022) to set up wflow_sbm models for any catchment based on globally available datasets, e.g. SoilGrids (Hengl et al., 2017), GlobCover 2009 (Arino et al., 2010) and MERIT Hydro (Yamazaki et al., 2019). Point-scale (pedo)transfer functions (PTFs) from the literature are used to derive model parameters at the highest available resolution of the data and are scaled with suitable upscaling operators (Imhoff et al., 2020) to the desired model resolution. The advantage of this is that transfer functions are only constrained by field and laboratory measurements, although we acknowledge that the scale at which these PTFs can be applied remains uncertain (van Looy et al., 2017; Samaniego et al., 2017). Nevertheless, the application of this method to the Rhine Basin resulted, for most discharge gauging stations in the central and northern part of the basin, in Kling–Gupta efficiency (KGE; Gupta et al., 2009) values between 0.6 and 0.9 (Imhoff et al., 2020). In the meantime, wflow_sbm and the aforementioned approach were used and tested to model the basins in the upper region of the greater Chao Phraya River in Thailand (Wannasin et al., 2021a, b) and the Citarum River in Indonesia (Rusli et al., 2021). Meijer et al. (2021) used wflow_sbm to rapidly develop a water resource model for the upper Niger Basin using global online data. Sperna Weiland et al. (2021) used wflow_sbm to assess climate change impacts in nine

river basins across Europe, while Aerts et al. (2022) used wflow_sbm to assess the impact of various model resolutions (200 m, 1 km, 3 km) on wflow_sbm performance for the CAMELS-US dataset.

The objective of this paper is to describe the wflow_sbm model in detail (model structure and equations) and to present some applications and envisaged future developments. Section 2 describes the development of the wflow_sbm model within the Wflow.jl framework and its model structure, model equations and functionalities. In Sect. 3 we describe the computational performance of wflow_sbm. Several applications of wflow_sbm are demonstrated in Sect. 4, followed by conclusions and foreseen future work in Sect. 5.

## 2 Model description

### 2.1 Overview

Wflow.jl (v0.7.3) (van Verseveld et al., 2024) is an open-source modelling framework for distributed hydrological modelling, containing multiple distributed hydrological model concepts implemented in the programming language Julia (Bezanson et al., 2017). It is a continuation of the wflow framework (Schellekens et al., 2020), which is based on the PCRaster Python framework (Karssenberg et al., 2010). The switch to the programming language Julia was made because Julia offers high performance (speed of C), required for large-scale high-resolution hydrological model applications, and is an "easy-to-use" language. Julia also opens up opportunities to parallelize the code for further improved computational performance. Wflow.jl provides several different vertical and lateral concepts that can be used for hydrological modelling and is compliant with the Basic Model Interface (BMI). Three vertical hydrological concepts are available within Wflow.jl: HBV-96 (wflow_hbv), FLEXTopo (wflow_flextopo) and SBM (wflow_sbm).

Wflow_sbm is the main hydrological model concept of the Wflow.jl framework and represents a family of hydrological models that have the vertical SBM concept in common. Wflow_sbm can have different lateral concepts that control how water (river, overland and subsurface flow) is routed, easily enabled by the modular structure of Wflow.jl. The wflow_sbm model presented here (Fig. 1) consists of the vertical SBM concept, and for the lateral components, the kinematic-wave approach is used for river, overland and lateral subsurface flow, similarly to TOPKAPI (Benning, 1995; Todini and Ciarapica, 2002), G2G (Bell et al., 2007), 1K-DHM (Tanaka and Tachikawa, 2015) and Topog_SBM (Vertessy and Elsenbeer, 1999). The vertical SBM concept is largely based on Topog_SBM (Vertessy and Elsenbeer, 1999), which considers the soil to be a "bucket" with a saturated and unsaturated store. While Topog_SBM is specifically designed to simulate fast-runoff processes during discrete storm events in small catchments ($< 10\,\text{km}^2$) as evapotranspiration losses are ignored, wflow_sbm can be applied to a wider variety of catchments. The main differences between wflow_sbm and Topog_SBM are as follows:

- the addition of evapotranspiration and interception losses;

- the addition of a root water uptake reduction function (Feddes et al., 1978);

- the addition of capillary rise;

- the addition of glacier, snow build-up and melting processes and an avalanche option for downhill snow transport;

- water being routed downstream over an eight-direction (D8) network, instead of the element network being based on contour lines and trajectories, used by Topog_SBM;

- the introduction of an option to divide the soil column into different layers to allow for transfer of water within the unsaturated zone.

Wflow_sbm has been applied in various catchments around the world showing satisfactory ($0.4 \geq \text{KGE} < 0.7$) to good ($\text{KGE} \geq 0.7$) performance (e.g. López López et al., 2016; Hassaballah et al., 2017; Giardino et al., 2019; Gebremicael et al., 2019; Imhoff et al., 2020; Laverde-Barajas et al., 2020; Wannasin et al., 2021a, b; Rusli et al., 2021; Meijer et al., 2021).

Figure 1 presents the different processes and fluxes in the wflow_sbm model. Precipitation enters each grid cell through the interception routine (total precipitation is first intercepted), based on the Gash model (Gash, 1979) or a modified Rutter model (Rutter et al., 1971, 1975) depending on the time stamp the model is using. Throughfall and stemflow from the interception routine are transferred to the optional snow (based on the HBV-96 hydrological model concept; Bergström, 1992) and glacier routines (based on the HBV-light degree-day-based model; Seibert et al., 2018). The soil in every grid cell is considered a single bucket, divided into a saturated and unsaturated store, with the option to divide the soil column into different layers. Available infiltration (stemflow and throughfall not converted into snow, including meltwater) infiltrates into the soil or becomes direct runoff based on the river fraction or open-water (excluding rivers) fraction. Soil infiltration is determined separately for the paved and nonpaved areas, as these have different infiltration capacities. Naturally, only the water that can be stored in the soil can infiltrate. If not all water can infiltrate, this is added as saturation excess water to the runoff routing scheme for overland flow. Infiltration excess occurs when the infiltration capacity is smaller than the available infiltration rate, and this amount of water is also added to the runoff routing scheme for overland flow. An exponential decay of the

saturated hydraulic conductivity with soil depth is assumed. Transfer of water to the unsaturated store and to the saturated store is based on Brooks and Corey (1964) when the soil column is divided into different layers, and in the case of one soil layer the original Topog_SBM vertical transfer formulation can also be used. Part of the water evaporates through soil evaporation, transpiration that is first derived from the saturated store if roots intersect with the saturated store and then from the unsaturated store, and open-water (excluding rivers) and river evaporation. Besides transpiration, capillary rise and leakage result in a flux from the saturated store to the unsaturated store and outside of the model domain, respectively. The kinematic-wave approach is used to route subsurface flow laterally. Saturation excess water occurs when the water table of lateral subsurface flow reaches the surface, and exfiltration of water from the unsaturated store to the surface because of a changing water table is added as saturation excess water to the runoff routing scheme for overland flow. For overland and river routing, the kinematic-wave approach is also used. Reservoir and lake models (optional) can be included within the kinematic-wave river routing.

The wflow_sbm model is described in more detail, including equations, in Sect. 2.2–2.7. These sections link to the main routines of wflow_sbm (Fig. 1):

1. "Interception" (Sect. 2.2)

2. "Snow and glaciers" (Sect. 2.3)

3. "The soil module and evapotranspiration" (Sect. 2.4)

4. "Lateral subsurface flow" (Sect. 2.5)

5. "Surface flow routing" (Sect. 2.6)

6. "Reservoirs and lakes" (Sect. 2.7).

Table A1 lists wflow_sbm state and flux variables (nonexhaustive). Additionally, wflow_sbm model inputs and parameters are listed in Table A2, including default values. Tables A1 and A2 both list the symbols that are used in Sect. 2.2–2.7 as well as the corresponding Wflow.jl names. It is possible to provide wflow_sbm with cyclic model parameters, for example monthly LAI maps (part of the HydroMT-Wflow default model setup), to estimate time-dependent interception parameters (see also Sect. 2.2.3) or to provide model parameters as part of forcing (besides precipitation, potential-evapotranspiration and temperature fields). For symbols that represent model parameters in Sect. 2.2–2.7 (except in Sect. 2.2.3), a time-dependent notation is not used because providing time-dependent model parameters is optional.

The equations of most hydrological processes of the wflow_sbm model are solved using the explicit Euler method for each model time step $t$ of length $\Delta t$ [s] (model time step size) specified in the configuration file. For the unsaturated flow through multiple soil layers, variable internal time stepping is used, as a maximum change in soil water per unsaturated soil layer is allowed to prevent the overestimation of vertical unsaturated flows (Sect. 2.4.2). The kinematic-wave equations (Sect. 2.5 and 2.6) are solved using Newton's method. For the river and overland flow kinematic wave, variable internal time stepping based on the Courant number or a fixed time step size [s] specified in the configuration file is used when iterations of the kinematic wave for surface flow are enabled in the configuration file; otherwise these equations are solved at model time step $t$ of length $\Delta t$ [s]. Reservoir and lake equations (Sect. 2.7) are solved using the explicit Euler method, except for the lake model with a parabolic weir, where the modified Puls approach is used. The reservoirs and lakes are part of the kinematic-wave river routing, and these equations are solved at the internal time step of the kinematic-wave river routing. The lateral subsurface flow is solved for each model time step using Newton's method, and depending on the model time step size, resolution and model parameters related to lateral subsurface flow, this may result in loss of accuracy. With a daily model time step, we estimate that using a minimum grid resolution of 200 m gives generally accurate lateral-subsurface-flow results. The use of the explicit Euler method for most equations means that results may differ between simulations that use a different model time step (for example daily vs. hourly). Most flux variables in wflow_sbm are defined per model time step $t$, and external flux parameters (expressed per day (the model base time step size)) are converted to the user-defined model time step size, during model initialization. For model equations in this section handling these flux variables, the model time step $t$ is implicitly embedded (e.g. for subtracting a flux variable from a storage variable, the flux variable is not multiplied by $t$ in the equation), which is identical to the code implementation of these equations.

To run a wflow_sbm model, several files are required: (1) a configuration file in the Tom's Obvious Minimal Language (TOML) format; (2) a netCDF file containing static and (optional) cyclic data, for example model parameters, flow direction, river network and gauges; and (3) a netCDF file containing forcing data (precipitation, potential-evapotranspiration and temperature fields). Storage and rating curves for lakes should be provided in CSV format. The static and forcing maps should have the same spatial domain and resolution; i.e. the regridding of forcing data is not supported. The focus of Wflow.jl is on the computations (computational engine), and the modular structure of the code simplifies extending the base code for pre- and post-processing purposes.

In the TOML file the following aspects are defined: simulation period and model time step size, model-specific settings like the model type (e.g. "sbm" for wflow_sbm) or whether to include snow modelling, locations and names of input and output files, mapping of internal model variables and parameters to external netCDF variables, optional modification of input model parameters and forcing, and output

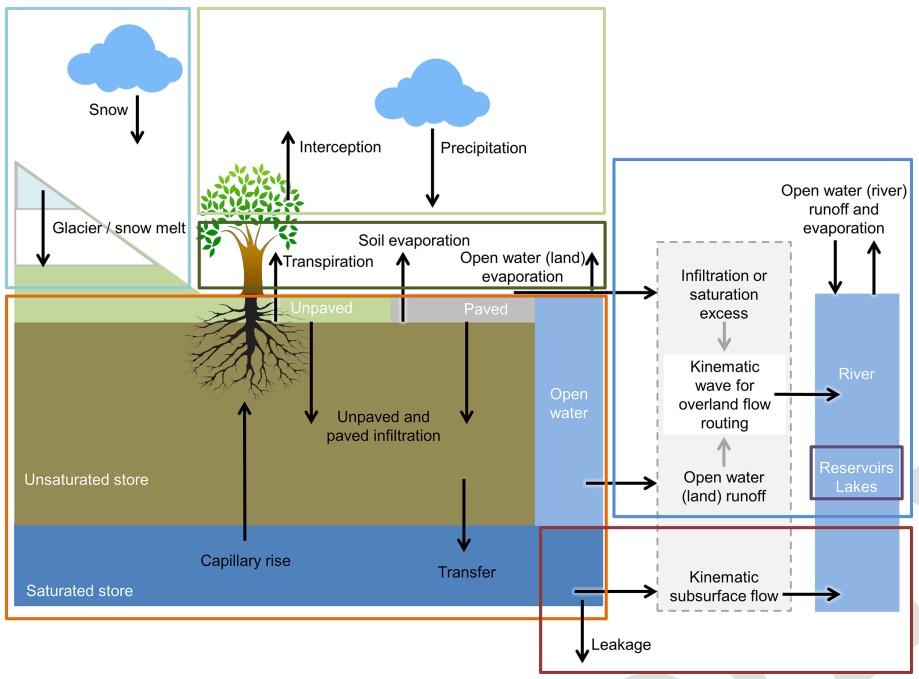

**Figure 1.** An overview of the different processes and fluxes in the wflow_sbm model (adopted from van Verseveld et al., 2024). The model includes the following routines: interception (green, Sect. 2.2), snow and glaciers (light blue, Sect. 2.3), soil module and evapotranspiration (orange, Sect. 2.4), lateral subsurface flow (brown, Sect. 2.5), surface routing (dark blue, Sect. 2.6), and reservoirs and lakes (black, Sect. 2.7).

options. Glacier and snow modelling and lake and reservoir modelling are optional (specific model settings). Wflow_sbm typically runs at a daily time step (recommended maximum model time step size) and a spatial resolution of $\sim 1$ km (we recommend a maximum grid resolution of $\sim 5$ km). Sub-daily model time steps are supported, for example for flow forecasting purposes or small (fast-responding) catchments. Output options consist of gridded data (netCDF) and scalar data (netCDF or CSV). Scalar data can be generated for individual grid cells or areas (e.g. sub-catchment).

For users that mainly want to run simulations without installing Julia, Wflow.jl is available as a compiled executable (cross-platform). Users that want to explore and modify the code and want to extend Wflow.jl (e.g. writing your own Julia scripts around the Wflow.jl package), we recommend installing Wflow.jl as a Julia package. The Wflow.jl documentation provides more details about the Wflow.jl installation and usage in the "User guide" section (van Verseveld et al., 2024).

## 2.2 Interception

For interception the Gash model (Gash, 1979) for daily (or larger) time steps and a modified Rutter model (Rutter et al., 1971, 1975) for sub-daily time steps are available within the Wflow.jl framework. The Gash model is a storm-based interception model, assuming one precipitation event per model time step and is applied when wflow_sbm runs at a daily (or larger) time step. For sub-daily time steps, a modified Rutter model is used.

### 2.2.1 Gash model

The original Gash model considers precipitation input to be a series of discrete storm events, where each storm event is divided into three sequential phases: (1) wetting phase during which precipitation saturates the canopy, (2) saturation phase during which the canopy is saturated and the precipitation intensity is higher than the average evaporation rate of the saturated canopy, and (3) drying phase during the period after precipitation has ceased. The precipitation rate to completely saturate the canopy $P_{\text{sat, max}}^t$ [mm $t^{-1}$] at time step $t$ is defined as

$$P_{\text{sat,max}}^t = \frac{-P_{\text{sat}}^t S_{\text{canopy,max}}}{E_{\text{sat}}^t}$$
$$\times \ln\left[1 - \frac{E_{\text{sat}}^t}{P_{\text{sat}}^t}(1 - f_{\text{canopygap}} - f_{\text{stemflow}})^{-1}\right], \quad (1)$$

where $P_{\text{sat}}^t$ [mm $t^{-1}$] is the average precipitation intensity and $E_{\text{sat}}^t$ [mm $t^{-1}$] is the average evaporation rate during saturation of the canopy, $S_{\text{canopy, max}}$ [mm] is the canopy storage capacity, $f_{\text{canopygap}}$ [–] is the canopy gap fraction, and $f_{\text{stemflow}}$ [–] is the stemflow fraction. When wflow_sbm is not provided with the leaf area index (LAI), the ratio $\frac{E_{\text{sat}}^t}{P_{\text{sat}}^t}$ [–] is used as the model parameter (Table A2); otherwise $E_{\text{sat}}^t =$

$(1 - f_{\text{canopygap}})E^t_{\text{pot, total}}$, where $E^t_{\text{pot, total}}$ [mm $t^{-1}$] is potential evapotranspiration (for the computation of $f_{\text{canopygap}}$ as a function of LAI, see Sect. 2.2.3), $P^t_{\text{sat}}$ is then equal to the total precipitation input rate $P^t$ [mm $t^{-1}$], and $f_{\text{canopygap}}$ is time dependent when LAI is provided as a cyclic (e.g. monthly) parameter or as part of forcing. The stemflow fraction $f_{\text{stemflow}}$ in wflow_sbm is defined as a fixed fraction (0.1) of $f_{\text{canopygap}}$ limited by the canopy fraction $(1 - f_{\text{canopygap}})$, and stemflow $P^t_{\text{stemflow}}$ [mm $t^{-1}$] at time step $t$ is calculated as follows:

$$f_{\text{stemflow}} = \min(0.1 f_{\text{canopygap}}, 1 - f_{\text{canopygap}}), \tag{2}$$

$$P^t_{\text{stemflow}} = f_{\text{stemflow}} P^t. \tag{3}$$

Interception during the wetting phase $I^t_{\text{wet}}$ [mm $t^{-1}$], saturation phase $I^t_{\text{sat}}$ [mm $t^{-1}$] and dry phase $I^t_{\text{dry}}$ [mm $t^{-1}$] at time step $t$ is given by Eqs. (4)–(6):

$$I^t_{\text{wet}} = \begin{cases} (1 - f_{\text{canopygap}} - f_{\text{stemflow}})P^t_{\text{sat,max}} - S_{\text{canopy,max}} & \text{if } P^t > P^t_{\text{sat, max}} \\ (1 - f_{\text{canopygap}} - f_{\text{stemflow}})P^t & \text{otherwise,} \end{cases} \tag{4}$$

$$I^t_{\text{sat}} = \begin{cases} \frac{E^t_{\text{sat}}}{P^t_{\text{sat}}}(P^t - P^t_{\text{sat,max}}) & \text{if } P^t > P^t_{\text{sat, max}} \\ 0 & \text{otherwise,} \end{cases} \tag{5}$$

$$I^t_{\text{dry}} = \begin{cases} S_{\text{canopy,max}} & \text{if } P^t > P^t_{\text{sat, max}} \\ 0 & \text{otherwise.} \end{cases} \tag{6}$$

The total interception $I^t_{\text{total}}$ [mm $t^{-1}$] at time step $t$, assuming that trunk interception can be neglected, is the sum of the interception in all three phases, bounded by $E^t_{\text{pot, total}}$:

$$I^t_{\text{total}} = \min(I^t_{\text{wet}} + I^t_{\text{dry}} + I^t_{\text{sat}}, E^t_{\text{pot, total}}). \tag{7}$$

Throughfall $P^t_{\text{throughfall}}$ [mm $t^{-1}$] at time step $t$ is the remainder after subtracting the total interception and stemflow from the precipitation:

$$P^t_{\text{throughfall}} = P^t - I^t_{\text{total}} - P^t_{\text{stemflow}}. \tag{8}$$

The remaining potential evaporation $E^t_{\text{pot, remainder}}$ [mm $t^{-1}$] at time step $t$ is given by

$$E^t_{\text{pot, remainder}} = E^t_{\text{pot, total}} - I^t_{\text{total}}. \tag{9}$$

### 2.2.2 Modified Rutter model

For sub-daily time steps, a modified Rutter interception model is used, which compared to the Gash model keeps track of the canopy storage $S^t_{\text{canopy}}$ [mm] and is updated in two steps. Stemflow is calculated in the same way as the Gash model; see Eq. (3). The precipitation rate on the canopy $P^t_{\text{canopy}}$ [mm $t^{-1}$] at time step $t$ is a function of the total precipitation input rate and the canopy gap and stemflow fractions:

$$P^t_{\text{canopy}} = (1 - f_{\text{canopygap}} - f_{\text{stemflow}})P^t. \tag{10}$$

The initial drainage rate $D^t_{\text{canopy,s1}}$ [mm $t^{-1}$] from the canopy storage at time step $t$ is the surplus of canopy storage at the previous time step compared to the canopy storage capacity $S_{\text{canopy, max}}$ [mm]:

$$D^t_{\text{canopy,s1}} = \begin{cases} S^{t-1}_{\text{canopy}} - S_{\text{canopy,max}} & \text{if } S^{t-1}_{\text{canopy}} > S_{\text{canopy,max}} \\ 0 & \text{otherwise.} \end{cases} \tag{11}$$

This check is required because $S_{\text{canopy, max}}$ (and $f_{\text{canopygap}}$) can change over time (see Sect. 2.2.3). The canopy storage is then updated based on the initial canopy drainage rate and precipitation that falls on the canopy (Eq. 12); then the evaporation rate from the canopy storage $E^t_{\text{canopy}}$ [mm $t^{-1}$] is determined (Eq. 13) and subtracted from $S^t_{\text{canopy}}$ (Eq. 14):

$$S^t_{\text{canopy}} = S^{t-1}_{\text{canopy}} + P^t_{\text{canopy}} - D^t_{\text{canopy,s1}}, \tag{12}$$

$$E^t_{\text{canopy}} = \min(S^t_{\text{canopy}}, E^t_{\text{pot,total}}), \tag{13}$$

$$S^t_{\text{canopy}} = S^t_{\text{canopy}} - E^t_{\text{canopy}}. \tag{14}$$

The remaining potential evaporation $E^t_{\text{pot, remainder}}$ [mm $t^{-1}$] at time step $t$ is given by

$$E^t_{\text{pot, remainder}} = E^t_{\text{pot,total}} - E^t_{\text{canopy}}. \tag{15}$$

The canopy storage $S^t_{\text{canopy}}$ is drained again if required with drainage rate $D^t_{\text{canopy,s2}}$ at time step $t$,

$$D^t_{\text{canopy,s2}} = \begin{cases} S^t_{\text{canopy}} - S_{\text{canopy,max}} & \text{if } S^t_{\text{canopy}} > S_{\text{canopy,max}} \\ 0 & \text{otherwise,} \end{cases} \tag{16}$$

and $S^t_{\text{canopy}}$ is updated to get the final canopy storage $S^t_{\text{canopy}}$ [mm]:

$$S^t_{\text{canopy}} = S^t_{\text{canopy}} - D^t_{\text{canopy,s2}}. \tag{17}$$

Throughfall $P^t_{\text{throughfall}}$ [mm $t^{-1}$] at time step $t$ is given by adding the total drainage rate from the canopy and the precipitation rate directly on the ground:

$$P^t_{\text{throughfall}} = D^t_{\text{canopy,s1}} + D^t_{\text{canopy,s2}} + f_{\text{canopygap}} P^t. \tag{18}$$

The total interception $I^t_{\text{total}}$ [mm $t^{-1}$] at time step $t$ is given by

$$I^t_{\text{total}} = P^t + D^t_{\text{canopy,s1}} - P^t_{\text{stemflow}} - P^t_{\text{throughfall}}. \tag{19}$$

### 2.2.3 Interception model parameters from leaf area index (LAI)

Within wflow_sbm it is possible to estimate interception model parameters based on LAI [m$^2$ m$^{-2}$] as static input (generally not used), as time-dependent input such as cyclic time series (climatology) or as part of forcing. It is assumed that the canopy capacity for leaves $S^t_{\text{leaf, max}}$ is linearly related to LAI$^t$ at time step $t$ through the specific leaf storage $S_{\text{leaf}}$ [mm] (van Dijk and Bruijnzeel, 2001):

$$S^t_{\text{leaf,max}} = S_{\text{leaf}} \text{LAI}^t. \tag{20}$$

The specific leaf storage is related to land cover type. Also the storage for the woody part of the vegetation $S_{\text{wood, max}}$ [mm] is required to estimate total canopy capacity $S_{\text{canopy, max}}^t$ [mm] at time step $t$. The relations between land cover and $S_{\text{leaf}}$ and $S_{\text{wood, max}}$ are based on Pitman (1989) and Liu (1998). The canopy gap fraction $f_{\text{canopygap}}^t$ [–] at time step $t$ is determined using the extinction coefficient $k$ based on van Dijk and Bruijnzeel (2001), and the value of $k$ is related to land cover type:

$$f_{\text{canopygap}}^t = e^{(-k\text{LAI}^t)}. \tag{21}$$

## 2.3 Snow and glaciers

### 2.3.1 Snow

Snow processes are adopted from the HBV-96 hydrological model concept (Bergström, 1992). The effective precipitation rate $P_{\text{effective}}^t$ [mm $t^{-1}$] (throughfall and stemflow) occurs as snowfall $P_{\text{snow}}^t$ [mm $t^{-1}$] at time step $t$ if the air temperature $T_{\text{air}}^t$ [°C] at time step $t$ is below a user-defined temperature threshold $s_{\text{fall, }T\text{ threshold}}$ [°C]. An interval parameter $s_{\text{fall, }T\text{ interval}}$ [°C] defines the range over which precipitation falls partly as snow and partly as rain, with 100 % snow at the lower end and decreasing linearly to 0 % at the upper end. The fraction of precipitation that occurs as rainfall $f_{\text{rain}}^t$ [–] at time step $t$ is calculated as TS1

$$f_{\text{rain}}^t = \begin{cases} 0 & \text{if } s_{\text{fall, }T\text{ interval}} = 0 \text{ and} \\ & T_{\text{air}}^t \leq s_{\text{fall, }T\text{ threshold}} \\ 1 & \text{if } s_{\text{fall, }T\text{ interval}} = 0 \text{ and} \\ & T_{\text{air}}^t > s_{\text{fall, }T\text{ threshold}} \\ \max\Big(\min \\ \Big(\frac{T_{\text{air}}^t - (s_{\text{fall, }T\text{ threshold}} - 0.5\, s_{\text{fall, }T\text{ interval}})}{s_{\text{fall, }T\text{ interval}}}, 1\Big), 0\Big) & \text{if } s_{\text{fall, }T\text{ interval}} \neq 0. \end{cases} \tag{22}$$

This fraction is used to calculate snowfall $P_{\text{snow}}^t$ and rainfall $P_{\text{rain}}^t$ [mm $t^{-1}$] at time step $t$ as follows:

$$P_{\text{snow}}^t = (1 - f_{\text{rain}}^t) P_{\text{effective}}^t, \tag{23}$$
$$P_{\text{rain}}^t = f_{\text{rain}}^t P_{\text{effective}}^t. \tag{24}$$

For snowmelt, HBV-96 uses a degree-day approach, an empirical relationship between melt and air temperature. If $T_{\text{air}}^t$ is above a melting temperature threshold $s_{\text{melt, }T\text{ threshold}}$ [°C], snowmelt occurs. The potential snowmelt rate $M_{\text{snow,pot}}^t$ [mm $t^{-1}$] at time step $t$, using the degree-day factor $s_{\text{ddf}}$ [mm $t^{-1}$ °C$^{-1}$], is calculated as follows:

$$M_{\text{snow,pot}}^t = \begin{cases} s_{\text{ddf}}(T_{\text{air}}^t - s_{\text{melt, }T\text{ threshold}}) & \text{if } T_{\text{air}}^t > s_{\text{melt, }T\text{ threshold}} \\ 0 & \text{otherwise.} \end{cases} \tag{25}$$

The actual snowmelt rate $M_{\text{snow,act}}^t$ [mm $t^{-1}$] at time step $t$ is limited by the snow storage $S_{\text{snow}}^{t-1}$ [mm] at the previous time step and is calculated by taking the minimum of $M_{\text{snow,pot}}^t$ and $S_{\text{snow}}^{t-1}$. The snowpack retains water that can refreeze if $T_{\text{air}}^t$ is below $s_{\text{melt, }T\text{ threshold}}$. The potential refreezing rate $M_{\text{refreeze,pot}}^t$ [mm $t^{-1}$] at time step $t$ is controlled by $s_{\text{ddf}}$, a coefficient of refreezing $s_{\text{refreeze}}$ [–] (fixed: 0.05), $T_{\text{air}}^t$ and $s_{\text{melt, }T\text{ threshold}}$, as follows:

$$M_{\text{refreeze,pot}}^t = \begin{cases} s_{\text{ddf}} s_{\text{refreeze}}(s_{\text{melt, }T\text{ threshold}} - T_{\text{air}}^t) & \text{if } T_{\text{air}}^t < s_{\text{melt, }T\text{ threshold}} \\ 0 & \text{otherwise.} \end{cases} \tag{26}$$

The actual refreezing rate $M_{\text{refreeze,act}}^t$ [mm $t^{-1}$] is based on the liquid water content of snow $S_{\text{snow,liquid}}^{t-1}$ [mm] at the previous time step and the potential refreezing rate $M_{\text{refreeze,pot}}^t$, by taking the minimum of $M_{\text{refreeze,pot}}^t$ and $S_{\text{snow,liquid}}^{t-1}$. Snow storage $S_{\text{snow}}^t$ [mm] at time step $t$ is then a function of snow storage $S_{\text{snow}}^{t-1}$ at the previous time step, snowfall, the actual refreezing rate and actual snowmelt at time step $t$:

$$S_{\text{snow}}^t = S_{\text{snow}}^{t-1} + P_{\text{snow}}^t + M_{\text{refreeze,act}}^t - M_{\text{snow,act}}^t. \tag{27}$$

The liquid water content of snow $S_{\text{snow,liquid}}^t$ at time step $t$ is a function of the liquid water content of snow $S_{\text{snow,liquid}}^{t-1}$ at the previous time step; the actual refreezing rate, actual snowmelt and rainfall rate at time step $t$; and the maximum amount of water that the snowpack can hold. This maximum amount is controlled by the water-holding capacity $s_{\text{whc}}$ [–] of snow and snow storage at time step $t$:

$$S_{\text{snow,liquid}}^t = S_{\text{snow,liquid}}^{t-1} + M_{\text{snow,act}}^t + P_{\text{rain}}^t - M_{\text{refreeze,act}}^t, \tag{28}$$
$$S_{\text{snow,liquid}}^t = S_{\text{snow,liquid}}^t - \max(S_{\text{snow,liquid}}^t - S_{\text{snow}}^t s_{\text{whc}}, 0). \tag{29}$$

The amount that does exceed the water-holding capacity of snow ($\max(S_{\text{snow,liquid}}^t - S_{\text{snow}}^t s_{\text{whc}}, 0)$) is available as rainfall at time step $t$.

To control unlimited build-up of the snowpack at high altitude where temperature rarely reaches above the melting temperature, the optional avalanche routine can be used to transport snow, based on the local drain network, downhill, where it becomes available for snowmelt. The fraction of snow that can be transported downhill is calculated as

$$f_{\text{snow transport}}^t = \min\left(0.5, \frac{c_{\text{land slope}}}{\tan(80°)}\right) \min\left(1, \frac{S_{\text{snow}}^t}{s_{\text{max}}}\right), \tag{30}$$

with $f_{\text{snow transport}}^t$ [–] the fraction of snow at time step $t$ that is available for transport downhill, $c_{\text{land slope}}$ [m m$^{-1}$] the slope of the land surface and $s_{\text{max}}$ [mm] the maximum snowpack with a fixed value of 10 000 mm. The fraction of snow that can be transported downhill is multiplied with the snowpack storage and represents the transport capacity of snow $M_{\text{snow,downhill}}^t$ [mm $t^{-1}$] at time step $t$:

$$M_{\text{snow,downhill}}^t = S_{\text{snow}}^t f_{\text{snow transport}}^t. \tag{31}$$

Snow is then transported downhill based on the local drain network and the transport capacity of snow, which limits the

snow transport, updating the snow storage $S_{\text{snow}}^t$ [mm] and liquid water content of snow $S_{\text{snow,liquid}}^t$ [mm] in each grid cell at time step $t$.

### 2.3.2 Glaciers

Glacier modelling considers two main processes: glacier build-up from snow turning into firn (adopted from the HBV-light model; Seibert et al., 2018) and glacier melt (using a temperature degree-day model). This glacier modelling approach considers firn to be a part of the accumulated glacier mass. First, a fixed fraction $g_{\text{snow to firn}}$ [$t^{-1}$], which typically ranges between 0.001 and 0.006 for a daily time step, of snow storage $S_{\text{snow}}^t$ [mm] on top of the glacier is converted into firn for each time step $t$ of length $\Delta t$ [s]:

$$S_{\text{snow to firn}}^t = \min\left(g_{\text{snow to firn}} S_{\text{snow}}^t, 8\frac{\Delta t}{\Delta t_{\text{b}}}\right), \tag{32}$$

where $S_{\text{snow to firn}}^t$ [mm $t^{-1}$] is the snow-into-firn conversion rate at time step $t$, with a maximum conversion rate of $8 \text{ mm d}^{-1}$. This maximum conversion rate is scaled by $\Delta t$ and the model base time step size $\Delta t_{\text{b}}$ of $86\,400$ [s]. The snow storage from the snow module (Sect. 2.3.1) $S_{\text{snow}}^t$ [mm] at time step $t$ is then updated as follows:

$$S_{\text{snow}}^t = S_{\text{snow}}^t - S_{\text{snow to firn}}^t f_{\text{glacier}}, \tag{33}$$

with $f_{\text{glacier}}$ [–] the fraction of a grid cell covered by a glacier. When the snowpack on top of the glacier is almost all melted ($S_{\text{snow}}^t < 10 \text{ mm}$), glacier melt is enabled and estimated with a degree-day model. If the air temperature $T_{\text{air}}^t$ is above a melting temperature threshold $g_{\text{melt, }T\text{ threshold}}$ [°C], glacier melt occurs. The potential glacier melt $M_{\text{glacier,pot}}^t$ [mm $t^{-1}$], using the degree-day factor $g_{\text{ddf}}$ [mm $t^{-1}$ °C$^{-1}$], is calculated as

$$M_{\text{glacier,pot}}^t = \begin{cases} g_{\text{ddf}}(T_{\text{air}}^t - g_{\text{melt, }T\text{ threshold}}) & \text{if } T_{\text{air}}^t > g_{\text{melt, }T\text{ threshold}} \\ 0 & \text{otherwise.} \end{cases} \tag{34}$$

The actual glacier melt $M_{\text{glacier,act}}^t$ [mm $t^{-1}$] at time step $t$ is limited by the glacier storage at the previous time step $S_{\text{glacier}}^{t-1}$ [mm] (expressed in mm water equivalent) and $S_{\text{snow to firn}}^t$ as follows:

$$M_{\text{glacier,act}}^t = \min(M_{\text{glacier,pot}}^t, S_{\text{glacier}}^{t-1} + S_{\text{snow to firn}}^t). \tag{35}$$

The glacier storage $S_{\text{glacier}}^t$ [mm] at time step $t$ is then updated as follows:

$$S_{\text{glacier}}^t = S_{\text{glacier}}^{t-1} + S_{\text{snow to firn}}^t - M_{\text{glacier,act}}^t. \tag{36}$$

A map with $S_{\text{glacier}}$ values can be provided as an initial state (default: 5500 mm) when wflow_sbm is initialized with default values in the code ("cold" start); see also Table A2.

### 2.4 The soil module and evapotranspiration

#### 2.4.1 Infiltration

The infiltration rate of available water $F_{\text{available}}^t$ [mm $t^{-1}$] at time step $t$ into the soil (throughfall, stemflow, snowmelt and glacier melt) is first added to the saturated parts of the grid cell: the river flow and overland flow components of wflow_sbm. This is based on the river fraction $f_{\text{river}}$ [–] (river flow component) and open-water fraction (excluding rivers) $f_{\text{open water}}$ [–] (overland flow component) within a grid cell as follows:

$$R_{\text{river}}^t = f_{\text{river}} F_{\text{available}}^t, \tag{37}$$

$$R_{\text{open water}}^t = f_{\text{open water}} F_{\text{available}}^t, \tag{38}$$

where $R_{\text{river}}^t$ [mm $t^{-1}$] is runoff from the river fraction in a cell at time step $t$ and $R_{\text{open water}}^t$ [mm $t^{-1}$] is runoff from the open-water fraction in a cell at time step $t$. $R_{\text{river}}^t$ and $R_{\text{open water}}^t$ are later added to the wflow_sbm river and overland flow components, respectively. The infiltration rate of the remaining available water $F_{\text{available}}^t$ [mm $t^{-1}$] at time step $t$ into the soil is determined as

$$F_{\text{available}}^t = F_{\text{available}}^t - R_{\text{river}}^t - R_{\text{open water}}^t. \tag{39}$$

The soil in wflow_sbm is considered a bucket with a depth $z_{\text{soil}}$ [mm] and is divided into a saturated store $S_{\text{sat}}$ [mm] and an unsaturated store $S_{\text{unsat}}$ [mm]. The top of the saturated store forms a pseudo-water table at depth $z_{\text{watertable}}$ [mm] such that the value of $S_{\text{sat}}$ is given by

$$S_{\text{sat}} = (z_{\text{soil}} - z_{\text{watertable}})(\theta_{\text{s}} - \theta_{\text{r}}), \tag{40}$$

where $\theta_{\text{s}}$ and $\theta_{\text{r}}$ are the saturated and residual soil water contents, respectively, both expressed in mm mm$^{-1}$. The amount of water that can infiltrate is a function of the infiltration capacity $c_{\text{infiltration, paved}}$ [mm $t^{-1}$] of the compacted soil (or paved area) fraction ($f_{\text{paved}}$ [–]) of each grid cell, the infiltration capacity $c_{\text{infiltration, unpaved}}$ [mm $t^{-1}$] of the non-compacted soil fraction (or unpaved area) ($(1 - f_{\text{paved}})$) of each grid cell, the initial storage capacity of the unsaturated zone $S_{\text{unsat,max}}^t$ [mm] (Eq. 43) at time step $t$ and an optional reduction factor $f_{\text{frozen}}$ applied to the infiltration capacity when snow is modelled, and the model setting soilinfreduction is set to true. The parameter $f_{\text{frozen}}$ depends on the near-surface soil temperature, which is modelled based on the approach of Wigmosta et al. (2009):

$$T_{\text{soil}}^t = T_{\text{soil}}^{t-1} + w_{\text{soil}}(T_{\text{air}}^t - T_{\text{soil}}^{t-1}), \tag{41}$$

where $T_{\text{soil}}^t$ [°C] is the near-surface soil temperature at time step $t$, $T_{\text{air}}^t$ [°C] is the air temperature at time step $t$, $T_{\text{soil}}^{t-1}$ [°C] is the near-surface soil temperature at the previous time step and $w_{\text{soil}}$ is a weighting coefficient [$t^{-1}$]. The optional infiltration capacity reduction factor $f_{\text{frozen}}^t$ at time step $t$ is

based on the model parameter $f_{\text{red, frozen}}$ [–] and the near-surface soil temperature as follows:

$$f_{\text{frozen}}^t = \begin{cases} \frac{1}{b + e^{(-c(T_{\text{soil}}^t - a))}} + f_{\text{red,frozen}} & \text{if snow and soilinfreduction} \\ 1 & \text{otherwise,} \end{cases} \quad (42)$$

where $b = \frac{1}{(1 - f_{\text{red,frozen}})}$, $a = 0$ and $c = 8$. The initial storage capacity of the unsaturated zone $S_{\text{unsat, max}}^t$ [mm] at time step $t$ is based on the saturated storage $S_{\text{sat}}^{t-1}$ [mm] at the previous time step, the sum of unsaturated storage in each soil layer $n$ for $m$ unsaturated soil layers $\sum_{n=1}^{m} S_{\text{unsat, }n}^{t-1}$ [mm] at the previous time step, and the total soil water capacity of the wflow_sbm soil bucket. $S_{\text{unsat, max}}^t$ is calculated as follows:

$$S_{\text{unsat, max}}^t = z_{\text{soil}}(\theta_s - \theta_r) - S_{\text{sat}}^{t-1} - \sum_{n=1}^{m} S_{\text{unsat,}n}^{t-1}. \quad (43)$$

The infiltration rate of remaining available water is split into two parts, the part that falls on compacted areas $F_{\text{available}}^t f_{\text{paved}}$ [mm $t^{-1}$] and the part that falls on non-compacted areas $F_{\text{available}}^t (1 - f_{\text{paved}})$ [mm $t^{-1}$] at time step $t$. The maximum infiltration rate in these areas is calculated by taking the minimum of the infiltration capacity and the infiltration rate for these areas:

$$F_{\text{unpaved}}^t = \min(c_{\text{infiltration,unpaved}}\, f_{\text{frozen}}^t, F_{\text{available}}^t (1 - f_{\text{paved}})), \quad (44)$$

$$F_{\text{paved}}^t = \min(c_{\text{infiltration,paved}}\, f_{\text{frozen}}^t, F_{\text{available}}^t f_{\text{paved}}). \quad (45)$$

The actual total infiltration rate $F_{\text{total}}^t$ [mm $t^{-1}$] is a function of the total infiltration rate (compacted and non-compacted areas) and the initial unsaturated storage capacity:

$$F_{\text{total}}^t = \min(F_{\text{unpaved}}^t + F_{\text{paved}}^t, S_{\text{unsat, max}}^t). \quad (46)$$

Finally, the infiltration excess water rate $F_{\text{excess}}^t$ [mm $t^{-1}$] at time step $t$ is determined as

$$\begin{aligned} F_{\text{excess}}^t = &(F_{\text{available}}^t (1 - f_{\text{paved}}) - F_{\text{unpaved}}^t) \\ &+ (F_{\text{available}}^t f_{\text{paved}} - F_{\text{paved}}^t). \end{aligned} \quad (47)$$

### 2.4.2 Soil water accounting scheme

The soil bucket in wflow_sbm with a depth $z_{\text{soil}}$ [mm] can be split up into different layers. Assuming a unit head gradient, the potential transfer of water $Q_{\text{transfer, pot, }n}^t$ [mm $t^{-1}$] from an unsaturated layer $n$ at time step $t$ is controlled by the vertical saturated hydraulic conductivity $K_{\text{vz, }n}^t$ [mm $t^{-1}$] at time step $t$ (Eq. 50) the effective saturation degree of layer $n$ at time step $t$, and a Brooks–Corey power coefficient $c_n$ [–] based on the pore size distribution index $\lambda_n$ [–] (Brooks and Corey, 1964) of layer $n$:

$$Q_{\text{transfer, pot, }n}^t = K_{\text{vz, }n}^t \left(\frac{\theta_n^t - \theta_r}{\theta_s - \theta_r}\right)^{c_n}, \quad (48)$$

$$c_n = \frac{2 + 3\lambda_n}{\lambda_n}, \quad (49)$$

where $\theta_n^t$ [mm mm$^{-1}$] is the soil water content of unsaturated soil layer $n$ at time step $t$ and $\theta_s$ and $\theta_r$ are as previously defined. The vertical saturated hydraulic conductivity $K_{\text{vz, }n}^t$ of unsaturated soil layer $n$ for $m$ unsaturated soil layers (based on the pseudo-water-table depth at the previous time step $z_{\text{watertable}}^{t-1}$ [mm]; $n = 1$ refers to the upper soil layer) at time step $t$ is given by

$$K_{\text{vz, }n}^t = \begin{cases} f_{\text{Kv, }n} K_{\text{v0}} e^{(-f_{\text{Kv}} z_{\text{bottom, }n})} & \text{if } n < m \\ f_{\text{Kv, }n} K_{\text{v0}} e^{(-f_{\text{Kv}} z_{\text{watertable}}^{t-1})} & \text{if } n = m, \end{cases} \quad (50)$$

where the model parameter $K_{\text{v0}}$ [mm $t^{-1}$] is the vertical saturated hydraulic conductivity at the soil surface (that declines exponentially with depth), $f_{\text{Kv, }n}$ is an optional (default: 1.0) multiplication factor [–] for each soil layer $n$ to correct the vertical saturated hydraulic conductivity, $f_{\text{Kv}}$ is a scaling parameter [mm$^{-1}$] and $z_{\text{bottom, }n}$ [mm] is the soil depth at the bottom of soil layer $n$. The thickness $z_{n,\text{ thickness}}^t$ [mm] of unsaturated soil layer $n$ for $m$ unsaturated soil layers at time step $t$ is given by

$$z_{n,\text{ thickness}}^t = \begin{cases} z_{n,\text{ thickness}} & \text{if } n < m \\ z_{\text{watertable}}^{t-1} - z_{\text{bottom, }n-1} & \text{if } n = m \text{ and } n > 1 \\ z_{\text{watertable}}^{t-1} & \text{if } n = 1 \text{ and } m = 1, \end{cases} \quad (51)$$

where $z_{n,\text{ thickness}}$ is the actual thickness of soil layer $n$ and $z_{\text{bottom, }n-1}$ [mm] is the soil depth at the bottom of the soil layer above soil layer $n$. For each unsaturated soil layer $n$, the transfer of water at time step $t$ for $m$ unsaturated soil layers is calculated as follows:

$$S_{\text{unsat, }n}^t = S_{\text{unsat,}n}^{t-1} + Q_{\text{in, }n}^t, \quad (52)$$

$$Q_{\text{transfer, pot, }n}^t = K_{\text{vz, }n}^t \min\left(\left(\frac{S_{\text{unsat, }n}^t}{z_{n,\text{ thickness}}^t (\theta_s - \theta_r)}\right)^{c_n}, 1\right), \quad (53)$$

$$Q_{\text{transfer, act, }n}^t = \min\left(Q_{\text{transfer,pot,}n}^t, S_{\text{unsat, }n}^t\right), \quad (54)$$

$$S_{\text{unsat, }n}^t = S_{\text{unsat, }n}^t - Q_{\text{transfer, act, }n}^t, \quad (55)$$

where $Q_{\text{in, }n}^t$ for unsaturated soil layer $n = 1$ (upper soil layer) is the actual total infiltration rate $F_{\text{total}}^t$ (Eq. 46) and, for unsaturated soil layer $n > 1$, $Q_{\text{in, }n}^t$ is the actual transfer of water from the soil layer above layer $n$ ($Q_{\text{transfer,act,}n-1}^t$ [mm $t^{-1}$]); $S_{\text{unsat, }n}^{t-1}$ [mm] is the unsaturated storage of layer $n$ at the previous time step; and $S_{\text{unsat, }n}^t$ [mm] is the subsequent updated unsaturated storage of layer $n$ at time step $t$. During each model time step $t$, internal time stepping (iterations) is applied to Eqs. (53), (54) and (55), under unsaturated conditions based on a maximum allowed change in $S_{\text{unsat, }n}$ of 0.2 mm for each internal time step to prevent an overestimation of $Q_{\text{transfer, pot, }n}^t$.

When the soil bucket in wflow_sbm is not split up into different layers, it is possible to use the original Topog_SBM vertical transfer formulation. The transfer of water from the unsaturated store to the saturated store is in that case controlled by the vertical saturated hydraulic conductivity at

depth $z_{\text{watertable}}^{t-1}$, an optional multiplication factor $f_{\text{Kv}, 1}$ [–] to correct the vertical saturated hydraulic conductivity, the ratio between the unsaturated storage $S_{\text{unsat}, 1}^t$ [mm] (resulting from Eq. 52) and the saturation deficit $S_{\text{deficit}}^t$ [mm], and the available unsaturated storage $S_{\text{unsat}, 1}^t$ at time step $t$:TS2

$$S_{\text{deficit}}^t = (\theta_s - \theta_r) z_{\text{soil}} - S_{\text{sat}}^{t-1}, \tag{56}$$

$$Q_{\text{transfer, pot}, 1}^t = f_{\text{Kv}, 1} K_{\text{v}0} e^{(-f_{\text{Kv}} z_{\text{watertable}}^{t-1})} \frac{S_{\text{unsat}, 1}^t}{S_{\text{deficit}}^t}, \tag{57}$$

$$Q_{\text{transfer, act}, 1}^t = \min(Q_{\text{transfer, pot}, 1}^t, S_{\text{unsat}, 1}^t). \tag{58}$$

### 2.4.3 Evapotranspiration

Open-water evaporation from waterbodies (excluding rivers) $E_{\text{open water}}^t$ [mm $t^{-1}$] and rivers $E_{\text{river}}^t$ [mm $t^{-1}$] at time step $t$ is based on the fraction of open water $f_{\text{open water}}$ [–], the fraction of rivers $f_{\text{river}}$ [–], the water level in the kinematic reservoir of the river flow component $S_{\text{wl, river}}^{t-1}$ [mm] and the overland flow component $S_{\text{wl, land}}^{t-1}$ [mm] at the previous time step, and the remaining potential evaporation after interception $E_{\text{pot, remainder}}^t$ [mm $t^{-1}$] as follows:

$$E_{\text{river}}^t = \min(f_{\text{river}} S_{\text{wl, river}}^{t-1}, f_{\text{river}} E_{\text{pot, remainder}}^t), \tag{59}$$

$$E_{\text{open water}}^t = \min(f_{\text{open water}} S_{\text{wl, land}}^{t-1}, f_{\text{open water}} E_{\text{pot, remainder}}^t). \tag{60}$$

The potential-evaporation rate remaining after interception (Eq. 9 or 15) and open-water evaporation (rivers and waterbodies (excluding rivers)) $E_{\text{pot, remainder}}^t$ [mm $t^{-1}$] at time step $t$ is then

$$E_{\text{pot, remainder}}^t = E_{\text{pot, remainder}}^t - E_{\text{river}}^t - E_{\text{open water}}^t. \tag{61}$$

Potential soil evaporation $E_{\text{pot, soil}}^t$ [mm $t^{-1}$] at time step $t$ is based on $E_{\text{pot, remainder}}^t$ and the canopy gap fraction $f_{\text{canopygap}}$ [–] (assumed to be identical to the amount of bare soil and can vary in time; Sect. 2.2.3). When the soil bucket in wflow_sbm is not split up into different layers, soil evaporation $E_{\text{act, soil}}^t$ [mm $t^{-1}$] is calculated as follows:

$$E_{\text{pot, soil}}^t = E_{\text{pot, remainder}}^t f_{\text{canopygap}}, \tag{62}$$

$$E_{\text{act, soil}}^t = \min(E_{\text{pot, soil}}^t \frac{S_{\text{deficit}}^t}{z_{\text{soil}}(\theta_s - \theta_r)}, S_{\text{unsat}, 1}^t), \tag{63}$$

with $S_{\text{deficit}}^t$, $z_{\text{soil}}$, $\theta_s$, $\theta_r$ and $S_{\text{unsat}, 1}^t$ (Eq. 55) as previously defined. As such, soil evaporation will be potential if the soil is fully wetted, and it decreases linearly with increasing soil moisture deficit, limited by $S_{\text{unsat}, 1}^t$. When the soil bucket in wflow_sbm is split up into different layers, soil evaporation $E_{\text{act, soil}}^t$ [mm $t^{-1}$] is restricted to the upper layer. As with the case of a single soil layer, potential soil evaporation is scaled according to the wetness of the soil layer, based on

the unsaturated layer storage from Eq. (55), as follows:TS3

$$E_{\text{act, soil}}^t =$$

$$\begin{cases} \min(E_{\text{pot, soil}}^t \frac{S_{\text{unsat}, 1}^t}{z_{\text{watertable}}^{t-1}(\theta_s - \theta_r)}, S_{\text{unsat}, 1}^t) & \text{if } z_{\text{watertable}}^{t-1} \leq z_{1, \text{thickness}} \\ \min(E_{\text{pot, soil}}^t \frac{S_{\text{unsat}, 1}^t}{z_{1, \text{thickness}}(\theta_s - \theta_r)}, S_{\text{unsat}, 1}^t) & \text{if } z_{\text{watertable}}^{t-1} > z_{1, \text{thickness}}, \end{cases} \tag{64}$$

where $z_{1, \text{thickness}}$ [mm] is the actual thickness of the upper layer and $\theta_s$, $\theta_r$ and $z_{\text{watertable}}^{t-1}$ are as previously defined. Soil evaporation $E_{\text{act, soil}}^t$ is subtracted from the unsaturated storage in the upper soil layer $S_{\text{unsat}, 1}^t$ (Eq. 55), and the remaining potential soil evaporation $E_{\text{remainder, soil}}^t$ is determined as follows:

$$S_{\text{unsat}, 1}^t = S_{\text{unsat}, 1}^t - E_{\text{act, soil}}^t, \tag{65}$$

$$E_{\text{remainder, soil}}^t = E_{\text{pot, soil}}^t - E_{\text{act, soil}}^t. \tag{66}$$

When the soil bucket in wflow_sbm is split up into different layers, soil evaporation $E_{\text{act, soil, sat}}^t$ [mm $t^{-1}$] from the saturated store is possible when the water table $z_{\text{watertable}}^{t-1}$ is present in the upper soil layer with actual thickness $z_{1, \text{thickness}}$ [mm], and it is calculated as follows:

$$E_{\text{act, soil, sat}}^t = \min(E_{\text{remainder, soil}}^t \frac{z_{1, \text{thickness}} - z_{\text{watertable}}^{t-1}}{z_{1, \text{thickness}}},$$

$$(z_{1, \text{thickness}} - z_{\text{watertable}}^{t-1})(\theta_s - \theta_r)), \tag{67}$$

with $\theta_s$ and $\theta_r$ as previously defined. In the case of a single soil layer or when $z_{\text{watertable}}^{t-1}$ is not present in the upper soil layer, $E_{\text{act, soil, sat}}^t$ is set at zero. $E_{\text{act, soil, sat}}^t$ is subtracted from the saturated store (at the previous time step):

$$S_{\text{sat}}^t = S_{\text{sat}}^{t-1} - E_{\text{act, soil, sat}}^t. \tag{68}$$

Potential transpiration $E_{\text{pot trans}}^t$ [mm $t^{-1}$] at time step $t$ is based on the remaining potential evaporation after interception and open-water evaporation (Eq. 61) and the canopy gap fraction as follows:

$$E_{\text{pot trans}}^t = E_{\text{pot, remainder}}^t (1 - f_{\text{canopygap}}). \tag{69}$$

In wflow_sbm, transpiration is first taken from the saturated store if the roots reach the water table $z_{\text{watertable}}^{t-1}$ at the previous time step. The fraction of wet roots $f_{\text{wet roots}}$ [–] (ranging between 0 and 1) is determined using a sigmoid function that defines the sharpness of the transition between fully wet and fully dry roots. Transpiration $E_{\text{trans, sat}}^t$ [mm $t^{-1}$] from the saturated store at time step $t$ is calculated as follows:

$$f_{\text{wet roots}} = \frac{1}{1 + e^{(-c_{\text{rd}}(z_{\text{watertable}}^{t-1} - z_{\text{rooting}}))}}, \tag{70}$$

$$E_{\text{trans, sat}}^t = \min(E_{\text{pot trans}}^t f_{\text{wet roots}}, S_{\text{sat}}^t), \tag{71}$$

where $c_{rd}$ [mm$^{-1}$] is a model parameter that controls the sharpness of the sigmoid function and $z_{rooting}$ [mm] is the rooting depth. The saturated store is then updated as follows:

$$S_{sat}^t = S_{sat}^t - E_{trans, sat}^t. \tag{72}$$

The remaining potential transpiration $E_{pot\,trans}^t$ [mm $t^{-1}$] available for transpiration from the unsaturated store is calculated as follows:

$$E_{pot\,trans}^t = E_{pot\,trans}^t - E_{trans, sat}^t. \tag{73}$$

The maximum allowed water extraction rate by roots $E_{root, max, n}^t$ [mm $t^{-1}$] from unsaturated soil layer $n$ at time step $t$ is a function of the fraction of roots $f_{roots, n}^t$ [–] of unsaturated layer $n$ and the available unsaturated storage [mm] of layer $n$:

$$E_{root, max, n}^t = f_{roots, n}^t S_{unsat, n}^t. \tag{74}$$

Next, a root water uptake reduction model based on Feddes et al. (1978) is used to calculate a reduction coefficient as a function of soil matric suction. Soil matric suction is calculated following Brooks and Corey (1964):

$$\frac{(\theta_n - \theta_r)}{(\theta_s - \theta_r)} = \begin{cases} \left(\frac{h_b}{h_n}\right)^{\lambda_n} & \text{if } h_n > h_b \\ 1 & \text{if } h_n \le h_b, \end{cases} \tag{75}$$

where $h_n$ is the soil matric suction [cm] of unsaturated soil layer $n$; $h_b$ is the air entry value [cm]; and $\theta_n$, $\theta_r$, $\theta_s$ and $\lambda_n$ are as previously defined. In wflow_sbm, soil matric suction $h_n^t$ for each unsaturated soil layer $n$ with thickness $z_{n, thickness}^t$ [mm] at time step $t$ is calculated as follows:

$$h_n^t = \frac{h_b}{\left(\frac{S_{unsat, n}^t / z_{n, thickness}^t}{(\theta_s - \theta_r)}\right)^{\lambda_n^{-1}}}, \tag{76}$$

where $S_{unsat, 1}^t$ is given by Eq. (65) and, for unsaturated layers $n > 1$, $S_{unsat, n}^t$ is given by Eq. (55). The root water uptake reduction coefficient $f_{root\,uptake, n}^t$ at time step $t$ with $h_n^t$ below or equal to $h_3$ (400 cm) is set to 1, with $h_n^t$ above or equal to $h_4$ (15 849 cm) is set to 0, and with $h_n^t$ between $h_3$ and $h_4$ declines linearly from 1 to 0. The values for $h_2$ (100 cm), $h_3$ and $h_4$ are fixed, and $h_1$ is set by $h_b$ (default: 10 cm); $h_b$ can be defined as input to the model. In the original transpiration reduction curve of Feddes et al. (1978), root water uptake below $h_1$ is set to zero (oxygen deficit) and between $h_1$ and $h_2$ root water uptake is limited. The assumption that very wet conditions do not affect root water uptake too much is probably generally applicable to natural vegetation; however for crops this assumption is not valid. This could be improved in the Wflow.jl code by applying the reduction to crops only. While the $h_3$ value is fixed, in the original transpiration reduction curve of Feddes et al. (1978), $h_3$ varies

with the potential transpiration rate, and this could also be improved in the code. For unsaturated soil layer $n$, transpiration $E_{trans, unsat, n}^t$ [mm $t^{-1}$] is controlled by $E_{root, max, n}^t$, the remaining transpiration $E_{pot\,trans, remainder}^t$ [mm $t^{-1}$] (for soil layer $n = 1$ see Eq. 73, and for layers $n > 1$ see Eq. 79), the unsaturated storage $S_{unsat, n}^t$ [mm] (for soil layer $n = 1$ see Eq. 65, and for layers $n > 1$ see Eq. 55) and $f_{root\,uptake, n}^t$ at time step $t$: TS4

$$E_{trans, unsat, n}^t = \min(E_{root, max, n}^t, E_{pot\,trans, remainder}^t, S_{unsat, n}^t) \\ \times f_{root\,uptake, n}^t. \tag{77}$$

At the same time $S_{unsat, n}^t$ and the remaining potential transpiration $E_{pot\,trans, remainder}^t$ [mm $t^{-1}$] are updated by subtracting $E_{trans, unsat, n}^t$:

$$S_{unsat, n}^t = S_{unsat, n}^t - E_{trans, unsat, n}^t, \tag{78}$$
$$E_{pot\,trans, remainder}^t = E_{pot\,trans, remainder}^t - E_{trans, unsat, n}^t. \tag{79}$$

After the soil water transfer, evaporation and transpiration computations, a soil water balance check is performed. Unsaturated storage that exceeds the maximum storage per layer is transferred to the layer above (or surface), from the bottom to the top unsaturated soil layer, resulting in an excess water rate at the surface $R_{excess, unsat}^t$ [mm $t^{-1}$]. The actual infiltration rate $F_{act}^t$ [mm $t^{-1}$] is then calculated as follows:

$$F_{act}^t = F_{total}^t - R_{excess, unsat}^t, \tag{80}$$

with $F_{total}^t$ as previously defined. The rate of water that cannot infiltrate due to saturated soil conditions $F_{excess, sat}^t$ [mm $t^{-1}$] is determined as

$$F_{excess, sat}^t = F_{available}^t - F_{total}^t - F_{excess}^t + R_{excess, unsat}^t, \tag{81}$$

with $F_{available}^t$, $F_{total}^t$ and $F_{excess}^t$ as previously defined.

Capillary rise in wflow_sbm is determined when an unsaturated zone occurs in the soil column at time step $t$ based on the water table depth at the previous time step ($z_{watertable}^{t-1} > 0$), using the following approach. First a maximum capillary rise $C_{max}^t$ [mm $t^{-1}$] at time step $t$ is determined from the minimum of $K_{vz, m}^t$ (Eq. 50); the actual transpiration rate from the unsaturated store $\sum_{n=1}^{m} E_{trans, unsat, n}^t$ for $m$ unsaturated soil layers; $S_{sat}^t$ (Eq. 72) and the unsaturated store capacity $S_{unsat, max}^t$, which is based on $S_{sat}^t$ (Eq. 72); and the sum of unsaturated storage $\sum_{n=1}^{m} S_{unsat, n}^t$ for $m$ unsaturated soil layers (soil water balance check after Eq. 78):

$$S_{unsat, max}^t = z_{soil}(\theta_s - \theta_r) - S_{sat}^t - \sum_{n=1}^{m} S_{unsat, n}^t, \tag{82}$$

$$C_{max}^t = \max(0.0, \min(K_{vz, m}^t, \sum_{n=1}^{m} E_{trans, unsat, n}^t, S_{unsat, max}^t, S_{sat}^t)), \tag{83}$$

with $z_{soil}$, $\theta_s$ and $\theta_r$ as previously defined. Then, the maximum capillary rate is scaled using the following empirical

equation (e.g. Zammouri, 2001; Yang et al., 2011; Wang et al., 2016):

$$C_{\text{act}}^t = \begin{cases} C_{\text{max}}^t \left(1 - \dfrac{z_{\text{watertable}}^{t-1}}{z_{\text{cap, maxdepth}}}\right)^{n_{\text{cap}}} & \text{if } (z_{\text{watertable}}^{t-1} > z_{\text{rooting}}) \text{ and} \\ & \quad (z_{\text{watertable}}^{t-1} < z_{\text{cap, maxdepth}}) \\ 0 & \text{otherwise,} \end{cases} \tag{84}$$

where $C_{\text{act}}^t$ [mm $t^{-1}$] is the capillary rate at time step $t$; $z_{\text{cap, maxdepth}}$ [mm] is the critical water depth beyond which capillary rise ceases; $n_{\text{cap}}$ [–] is an empirical coefficient related to soil properties and climate, generally set between 1–3; and $z_{\text{watertable}}^{t-1}$ and $z_{\text{rooting}}$ are as previously defined. When the soil bucket in wflow_sbm is split up into different layers, $C_{\text{act}}^t$ is divided over the different unsaturated soil layers, from the bottom to the top unsaturated soil layer, without exceeding the saturated water content $\theta_s$.

### 2.4.4 Leakage

In wflow_sbm it is possible to have leakage $L^t$ [mm $t^{-1}$] at time step $t$ from the saturated store $S_{\text{sat}}^t$ (Eq. 72) to deeper groundwater by setting the maximum leakage model parameter $L_{\text{max}}$ [mm $t^{-1}$] > 0. This water is lost from the saturated store and runs out of the model domain. $L^t$ is calculated as follows:

$$L^t = \min(f_{\text{Kv}, n_b} K_{v0} e^{(-f_{\text{Kv}} z_{\text{soil}})}, S_{\text{sat}}^t, L_{\text{max}}), \tag{85}$$

where $f_{\text{Kv}, n_b}$ is the optional multiplication factor (to correct the vertical saturated hydraulic conductivity) for the bottom soil layer $n_b$ and $K_{v0}$, $f_{\text{Kv}}$ and $z_{\text{soil}}$ are as previously defined.

## 2.5 Lateral subsurface flow

In wflow_sbm the kinematic-wave approach is used to route subsurface flow laterally. The saturated store can be drained laterally by saturated downslope subsurface flow for a slope with width $w$ [m] according to

$$Q_{\text{subsurface}} = \frac{K_{h0} c_{\text{land slope}}}{f_{\text{ssf, Kv}}} \times \left(e^{(-f_{\text{ssf, Kv}} z_{\text{ssf, watertable}})} - e^{(-f_{\text{ssf, Kv}} z_{\text{ssf, soil}})}\right) w, \tag{86}$$

where $c_{\text{land slope}}$ is the land slope [m m$^{-1}$], $Q_{\text{subsurface}}$ is subsurface flow [m$^3$ d$^{-1}$], $K_{h0} = 0.001 K_{v0} f_{\text{Kh0}} \frac{\Delta t_b}{\Delta t}$ is the horizontal saturated hydraulic conductivity at the soil surface [m d$^{-1}$] based on the vertical saturated hydraulic conductivity at the soil surface $K_{v0}$ [mm $t^{-1}$] and an optional multiplication factor $f_{\text{Kh0}}$ [–] (default: 100) applied to $K_{v0}$, $z_{\text{ssf, watertable}}$ [m] is the water table depth (set by $z_{\text{watertable}}$ [mm] after unit conversion at the start of the lateral-subsurface-flow computation), $z_{\text{ssf, soil}}$ [m] is the soil depth (set by $z_{\text{soil}}$ [mm] after unit conversion), $f_{\text{ssf, Kv}}$ is a scaling parameter [m$^{-1}$] (set by $f_{\text{Kv}}$ [mm$^{-1}$] after unit conversion), and $\Delta t_b$ and $\Delta t$ are as previously defined. This is combined

with the following continuity equation:

$$(\theta_s - \theta_r) w \frac{\delta h}{\delta t} = -\frac{\delta Q_{\text{subsurface}}}{\delta x} + wR, \tag{87}$$

where $h$ is the water table height [m]; $x$ is the distance downslope [m]; $w$ is the flow width [m]; $R = 0.001 \frac{\Delta t_b}{\Delta t} R_{\text{input}}$ is the net input rate [m d$^{-1}$] computed from the net input rate variable $R_{\text{input}}$ [mm $t^{-1}$], which is part of the vertical SBM concept; and $\theta_s$ and $\theta_r$ are as previously defined. Substituting for $h(\frac{\delta Q_{\text{subsurface}}}{\delta h})$ gives

$$\frac{\delta Q_{\text{subsurface}}}{\delta t} = -c \frac{\delta Q_{\text{subsurface}}}{\delta x} + cwR,$$
$$\text{with celerity } c = \frac{K_{h0} e^{(-f_{\text{ssf, Kv}} z_{\text{ssf, watertable}})} c_{\text{land slope}}}{(\theta_s - \theta_r)}. \tag{88}$$

The kinematic-wave equation for lateral subsurface flow is solved iteratively using Newton's method. In wflow_sbm, the flow width $w$ is calculated for each grid cell by dividing the cell area with the distance downslope $x$, based on the flow direction and the grid cell dimensions. The land slope $c_{\text{land slope}}$ needs to be provided for each grid cell in wflow_sbm. The net input rate $R_{\text{input}}$ in wflow_sbm consists of transfer of soil water $Q_{\text{transfer, act,} m}^t$ from unsaturated soil layer $m$ above the water table $z_{\text{watertable}}^{t-1}$ [mm] at the previous time step and the losses through capillary rise $C_{\text{act}}^t$, transpiration $E_{\text{trans, sat}}^t$ from the saturated store, leakage $L^t$ and soil evaporation $E_{\text{act, soil, sat}}^t$ from the saturated store, with the unit mm $t^{-1}$ for these terms. After the lateral-subsurface-flow calculation, which is bounded by the maximum lateral subsurface flow rate based on $z_{\text{ssf, soil}}$, a check is made to determine if saturation of the entire soil column occurs and as a consequence saturation excess overland flow is triggered. Water exfiltrating under saturated conditions $R_{\text{exfilt, sat}}^t$ [m $t^{-1}$] at time step $t$ is calculated as follows:

$$\Delta S_{\text{subsurface}}^t =$$
$$\frac{Q_{\text{subsurface, in}}^t \Delta t_{\text{ssf}} + 0.001 R_{\text{input}} wx - Q_{\text{subsurface, out}}^t \Delta t_{\text{ssf}}}{wx}, \tag{89}$$
$$R_{\text{exfilt, sat}}^t = \max(0, \Delta S_{\text{subsurface}}^t - z_{\text{ssf, watertable}}^{t-1}(\theta_s - \theta_r)), \tag{90}$$

where $\Delta S_{\text{subsurface}}^t$ [m] is the change in subsurface storage at time step $t$; $Q_{\text{subsurface, in}}^t$ [m$^3$ d$^{-1}$] is the subsurface flow into a cell at time step $t$; $Q_{\text{subsurface, out}}^t$ [m$^3$ d$^{-1}$] is the subsurface flow out of a cell at time step $t$; $z_{\text{ssf, watertable}}^{t-1}$ [m] is the water table depth at the previous time step; $\Delta t_{\text{ssf}}$ [d] is the time increment (same length as $\Delta t$ [s], expressed in different units [d]) of the lateral subsurface flow component; and $R_{\text{input}}$, $w$, $x$, $\theta_s$ and $\theta_r$ are as previously defined. Additionally, after the lateral-subsurface-flow calculation, wflow_sbm checks if exfiltration $R_{\text{exfilt, unsat}}^t$ [mm $t^{-1}$] of the unsaturated store onto the land surface occurs because of a change in water table depth $z_{\text{watertable}}$ [mm] (set by $z_{\text{ssf, watertable}}^t$ after unit conversion). This check is performed from the bottom unsaturated layer (at the previous time step) to the top unsaturated

layer, where the excess of unsaturated storage for each layer is transferred from the bottom to the top unsaturated layer, and can result in exfiltration of water onto the land surface.

## 2.6 Surface flow routing

The kinematic-wave approach is used for river and overland flow routing. The kinematic-wave equations (Chow et al., 1988) are

$$\frac{\delta Q}{\delta x} + \frac{\delta A}{\delta t} = Q_{\text{inflow}} \text{ and} \tag{91}$$

$$A = \alpha Q^{\beta}, \tag{92}$$

and they can be combined as

$$\frac{\delta Q}{\delta x} + \alpha \beta Q^{\beta-1} \frac{\delta Q}{\delta t} = Q_{\text{inflow}}, \tag{93}$$

where $Q$ is the surface flow in the kinematic wave [m$^3$ s$^{-1}$], $x$ is the length of the flow pathway [m], $A$ is the cross-section area of the flow pathway [m$^2$], $Q_{\text{inflow}}$ is the lateral inflow per unit length into the kinematic wave [m$^2$ s$^{-1}$], and $\alpha$ and $\beta$ are coefficients. These coefficients can be determined using Manning's equation (Chow et al., 1988), resulting in

$$\alpha = \left( \frac{n P_{\text{w}}^{\frac{2}{3}}}{\sqrt{c_{\text{slope}}}} \right)^{\beta} \text{ and } \beta = 0.6, \tag{94}$$

where $P_{\text{w}}$ [m] is the wetted perimeter, $c_{\text{slope}}$ ($c_{\text{land slope}}$ for overland flow and $c_{\text{river slope}}$ for river flow) is the slope [m m$^{-1}$] and $n$ ($n_{\text{land}}$ for overland flow and $n_{\text{river}}$ for river flow) is Manning's coefficient [s m$^{-1/3}$]. The wetted perimeter $P_{\text{w}}$ for river flow is calculated by adding the river width ($w_{\text{river}}$) and 2 times half of the river bankfull depth ($h_{\text{bankfull}}$). For overland flow, $P_{\text{w}}$ is set equal to the effective flow width, determined by dividing the grid cell area by the flow length and subtracting $w_{\text{river}}$. In wflow_sbm for river flow the parameters $w_{\text{river}}$, length ($x_{\text{river}}$) and $c_{\text{river slope}}$ need to be provided, and for overland flow $c_{\text{land slope}}$ needs to be provided. The lateral inflow per unit flow length for overland flow routing consists of infiltration excess water $F_{\text{excess}}^{t}$, saturation excess water during infiltration $F_{\text{excess, sat}}^{t}$, exfiltration water from the unsaturated store $R_{\text{exfilt, unsat}}^{t}$, water exfiltrating under saturated conditions $R_{\text{exfilt, sat}}^{t}$, runoff from open water $R_{\text{open water}}^{t}$ and open-water evaporation loss $E_{\text{open water}}^{t}$, converted to m$^2$ s$^{-1}$. The lateral inflow per unit length of $x_{\text{river}}$ for river flow routing consists of overland flow [m$^3$ s$^{-1}$], lateral subsurface flow [m$^3$ d$^{-1}$], runoff from the river $R_{\text{river}}^{t}$ [mm $t^{-1}$] and river evaporation loss $E_{\text{river}}^{t}$ [mm $t^{-1}$], converted to m$^2$ s$^{-1}$. Like the lateral subsurface routing, these equations are solved in wflow_sbm using Newton's method. The number of iterations for surface flow in the kinematic wave at time step $t$ of length $\Delta t$ [s] defaults to the Courant number $C$:

$$C = \frac{c_{\text{k}} \Delta t}{\Delta x}, \tag{95}$$

where $c_{\text{k}}$ [m s$^{-1}$] is the kinematic-wave celerity, $c_{\text{k}} = \frac{1}{\alpha \beta Q^{\beta-1}}$, and $\Delta x$ [m] is the space increment. The number of iterations within a time step $t$ is calculated by multiplying the 95th percentile of $C$ (to remove potential very high values (outliers)) for the wflow_sbm model domain with 1.25. The number of iterations can also be fixed to a specific sub-time step [s] for both river and overland flow; this is a model setting in the wflow_sbm configuration file. For river cells in wflow_sbm, where overland and river flow can both be present, lateral subsurface and overland flow into the river cell is partitioned based on the land slope of the river cell $c_{\text{land slope, river}}$ [m m$^{-1}$] and the land slope $c_{\text{land slope, upstream}}$ [m m$^{-1}$] of the upstream cell:

$$f_{\text{to river}} = \frac{c_{\text{land slope, upstream}}}{c_{\text{land slope, upstream}} + c_{\text{land slope, river}}}, \tag{96}$$

$$f_{\text{to land}} = 1 - f_{\text{to river}}, \tag{97}$$

where $f_{\text{to river}}$ [−] is the fraction of lateral subsurface or overland flow from an upstream cell that flows into the river and $f_{\text{to land}}$ [−] is the fraction of lateral subsurface or overland flow from an upstream cell that flows into the downstream kinematic reservoir of lateral subsurface and overland flow, respectively. In the case where a river cell has the same flow direction as the upstream cell, $f_{\text{to river}} = 0$, and thus overland and lateral subsurface flow from the upstream cell does not contribute to flow into the river.

## 2.7 Reservoirs and lakes

### 2.7.1 Reservoirs

In wflow_sbm, reservoirs can be included in the kinematic-wave routing for river flow. The first step in the reservoir module is to calculate the storage $S_{\text{res}}^{t_i}$ [m$^3$] at kinematic-wave time step $t_i$ with length $\Delta t_i$ [s], based on the storage $S_{\text{res}}^{t_i-1}$ at the previous time step of the kinematic wave, inflow $Q_{\text{in, res}}^{t_i}$ [m$^3$ s$^{-1}$] at time step $t_i$, average precipitation $P_{\text{res}}^{t_i}$ [mm $t_i^{-1}$] (converted from $P_{\text{res}}^{t}$ [mm $t^{-1}$]) and potential evapotranspiration $E_{\text{pot, res}}^{t_i}$ [mm $t_i^{-1}$] (converted from $E_{\text{pot, res}}^{t}$ [mm $t^{-1}$]) on the reservoir area $A_{\text{res}}$ [m$^2$] at time step $t_i$: TS5

$$S_{\text{res}}^{t_i} = S_{\text{res}}^{t_i-1} + Q_{\text{in, res}}^{t_i} + 0.001 P_{\text{res}}^{t_i} A_{\text{res}} - 0.001 E_{\text{pot, res}}^{t_i} A_{\text{res}}. \tag{98}$$

Then the storage fraction $f_{\text{res, storage}}^{t_i}$ [−] is calculated based on the maximum storage of the reservoir $S_{\text{res, max}}$ [m$^3$] (above this storage amount water is spilled):

$$f_{\text{res, storage}}^{t_i} = \frac{S_{\text{res}}^{t_i}}{S_{\text{res, max}}}. \tag{99}$$

The minimum release $R_{\text{min}}^{t_i}$ [m$^3$ $t_i^{-1}$] at the kinematic-wave time step $t_i$ is based on a sigmoid function, the minimum flow requirement downstream of the reservoir $Q_{\text{min req.}}$ [m$^3$ s$^{-1}$],

the target minimum storage fraction (of $S_{\text{res, max}}$) $f_{\text{res, min}}$ [–] and $f_{\text{res, storage}}^{t_i}$ at time step $t_i$,

$$R_{\text{min}}^{t_i} = \min\left(\frac{Q_{\text{min req.}}\Delta t_i}{1 + e^{-30(f_{\text{res, storage}}^{t_i} - f_{\text{res, min}})}}, S_{\text{res}}^{t_i}\right), \qquad (100)$$

and $R_{\text{min}}^{t_i}$ is subtracted from the reservoir storage $S_{\text{res}}^{t_i}$:

$$S_{\text{res}}^{t_i} = S_{\text{res}}^{t_i} - R_{\text{min}}^{t_i}. \qquad (101)$$

An additional release $R^{t_i}$ [m$^3$ $t_i^{-1}$] occurs when the reservoir storage is above the target maximum storage fraction $f_{\text{res, max}}$ [–], controlled by the maximum release capacity below the spillway $Q_{\text{max, res}}$ [m$^3$ s$^{-1}$]:

$$R_{\text{pot}}^{t_i} = \max(0, S_{\text{res}}^{t_i} - (S_{\text{res, max}} f_{\text{res, max}})), \qquad (102)$$

$$S_{\text{res, above max}}^{t_i} = \max(0, S_{\text{res}}^{t_i} - S_{\text{res, max}}), \qquad (103)$$

$$R^{t_i} = \min(R_{\text{pot}}^{t_i}, S_{\text{res, above max}}^{t_i} + Q_{\text{max, res}}\Delta t_i - R_{\text{min}}^{t_i}), \quad (104)$$

where $R_{\text{pot}}^{t_i}$ [m$^3$ $t_i^{-1}$] is the potential reservoir release and $S_{\text{res, above max}}^{t_i}$ [m$^3$] is the reservoir storage above the maximum reservoir storage. $R^{t_i}$ is subtracted from the reservoir storage $S_{\text{res}}^{t_i}$:

$$S_{\text{res}}^{t_i} = S_{\text{res}}^{t_i} - R^{t_i}. \qquad (105)$$

The total reservoir inflow $Q_{\text{in, res}}^{t}$ [m$^3$ $t^{-1}$] and outflow $Q_{\text{out, res}}^{t}$ [m$^3$ $t^{-1}$] for model time step $t$ of length $\Delta t$ [s] are calculated as follows:

$$Q_{\text{in, res}}^{t} = \sum_{i=1}^{n_i} Q_{\text{in, res}}^{t_i}\Delta t_i, \qquad (106)$$

$$Q_{\text{out, res}}^{t} = \sum_{i=1}^{n_i} (R_{\text{min}}^{t_i} + R^{t_i}), \qquad (107)$$

where $n_i$ refers to the number of iterations within model time step $t$ and $Q_{\text{in, res}}^{t_i}$, $R_{\text{min}}^{t_i}$, $R^{t_i}$ and $\Delta t_i$ are as previously defined.

### 2.7.2 Lakes

Lakes can be included in the kinematic-wave routing for river flow, and as with the reservoirs in wflow_sbm, a mass balance approach is used for modelling lakes:

$$\frac{S_{\text{lake}}^{t_i+\Delta t_i}}{\Delta t_i} = \frac{S_{\text{lake}}^{t_i}}{\Delta t_i} + Q_{\text{in, lake}}^{t_i+\Delta t_i}$$
$$+ \frac{0.001(P_{\text{lake}}^{t_i+\Delta t_i} - E_{\text{lake}}^{t_i+\Delta t_i})A_{\text{lake}}}{\Delta t_i} - Q_{\text{out, lake}}^{t_i+\Delta t_i}, \quad (108)$$

where $S_{\text{lake}}$ is lake storage [m$^3$], $t_i$ is the kinematic-wave time step of length $\Delta t_i$ [s], $Q_{\text{in, lake}}$ is the sum of inflows [m$^3$ s$^{-1}$], $Q_{\text{out, lake}}$ is the lake outflow at the outlet [m$^3$ s$^{-1}$],

$P_{\text{lake}}^{t_i+\Delta t_i}$ is the precipitation amount [mm] during $\Delta t_i$ (converted from $P_{\text{lake}}^{t}$ [mm $t^{-1}$]), $E_{\text{lake}}^{t_i+\Delta t_i}$ is lake evaporation [mm] during $\Delta t_i$ (converted from $E_{\text{lake}}^{t}$ [mm $t^{-1}$]) and $A_{\text{lake}}$ is the lake surface [m$^2$]. Most of the terms in Eq. (108) are known at the current or previous time step, except $S_{\text{lake}}^{t_i+\Delta t_i}$ and $Q_{\text{out, lake}}^{t_i+\Delta t_i}$. For lakes characterized by a storage curve of the form $S_{\text{lake}} = A_{\text{lake}} H_{\text{lake}}$ and the rating curve

$$Q_{\text{out, lake}} = \alpha_{\text{lake}}(H_{\text{lake}} - H_{0, \text{lake}})^{\beta_{\text{lake}}}, \qquad (109)$$

where $H_{0, \text{lake}}$ is the minimum water level under which the outflow is zero, $\alpha_{\text{lake}}$ [m s$^{-1}$] is a parameter that depends on lake outlet characteristics and the $\beta_{\text{lake}}$ exponent has a value of 2 (parabolic weir), the modified Puls approach is used. Then, $S_{\text{lake}}$ can be expressed as follows:

$$S_{\text{lake}} = A_{\text{lake}} H_{\text{lake}} = A_{\text{lake}}(h + H_{0, \text{lake}})$$
$$= \frac{A_{\text{lake}}}{\sqrt{\alpha_{\text{lake}}}}\sqrt{Q_{\text{out, lake}}} + A_{\text{lake}} H_{0, \text{lake}}, \qquad (110)$$

where $h = H_{\text{lake}} - H_{0, \text{lake}}$. Inserting this equation in the mass balance equation gives

$$\frac{A_{\text{lake}}}{\Delta t_i \sqrt{\alpha_{\text{lake}}}}\sqrt{Q_{\text{out, lake}}^{t_i+\Delta t_i}} + Q_{\text{out, lake}}^{t_i+\Delta t_i} = \frac{S_{\text{lake}}^{t_i}}{\Delta t_i} + Q_{\text{in, lake}}^{t_i+\Delta t_i}$$
$$+ \frac{0.001(P_{\text{lake}}^{t_i+\Delta t_i} - E_{\text{lake}}^{t_i+\Delta t_i})A_{\text{lake}}}{\Delta t_i}$$
$$- \frac{A_{\text{lake}} H_{0, \text{lake}}}{\Delta t_i}. \qquad (111)$$

The solution for $Q_{\text{out, lake}}^{t_i+\Delta t_i}$ is then

$$Q_{\text{out, lake}}^{t_i+\Delta t_i} =$$
$$\begin{cases} \left(\frac{-f_{\text{lake}} + \sqrt{f_{\text{lake}}^2 + 4\left(\text{SI}_{\text{lake}} - \frac{A_{\text{lake}} H_{0, \text{lake}}}{\Delta t_i}\right)}}{2}\right)^2 & \text{if } \text{SI}_{\text{lake}} > \frac{A_{\text{lake}} H_{0, \text{lake}}}{\Delta t_i} \\ 0 & \text{if } \text{SI}_{\text{lake}} \leq \frac{A_{\text{lake}} H_{0, \text{lake}}}{\Delta t_i}, \end{cases} \qquad (112)$$

where $f_{\text{lake}} = \frac{A_{\text{lake}}}{\Delta t_i \sqrt{\alpha_{\text{lake}}}}$ and $\text{SI}_{\text{lake}} = \frac{S_{\text{lake}}^{t_i}}{\Delta t_i} + Q_{\text{in, lake}}^{t_i+\Delta t_i} + \frac{0.001(P_{\text{lake}}^{t_i+\Delta t_i} - E_{\text{lake}}^{t_i+\Delta t_i})A_{\text{lake}}}{\Delta t_i}$.

The modified Puls approach is not applicable for lakes characterized by a rating curve (Eq. 109) with $\beta_{\text{lake}} \neq 2$ (non-parabolic weir; for a rectangular weir, usually a value of 3/2 is used) or a rating curve from measurements (linear interpolation of $Q_{\text{out, lake}}$ and $H_{\text{lake}}$ values in a lookup table) in combination with a storage curve from measurements (linear interpolation of $S_{\text{lake}}$ and $H_{\text{lake}}$ values in a lookup table) or computed from the relationship $S_{\text{lake}} = A_{\text{lake}} H_{\text{lake}}$. For these lakes $Q_{\text{out, lake}}$ is first computed for each time step based on $H_{\text{lake}}$ at the previous time step. Then, $S_{\text{lake}}$ is updated with Eq. (108), and $H_{\text{lake}}$ is updated with the storage curve based on the updated $S_{\text{lake}}$. For nearby lakes which are connected, it is possible to link the lakes and return flow can be allowed from the downstream to the upstream lake. An average lake

water level ($H_{\text{lake, avg}}$ [m]) should be provided as an initial state when wflow_sbm is initialized with default values in the code (cold start); see also Table A2. The total lake inflow $Q^t_{\text{in, lake}}$ [m$^3$ $t^{-1}$] and outflow $Q^t_{\text{out, lake}}$ [m$^3$ $t^{-1}$] for model time step $t$ are calculated as follows:

$$Q^t_{\text{in, lake}} = \sum_{i=1}^{n_i} Q^{t_i}_{\text{in, lake}} \Delta t_i, \tag{113}$$

$$Q^t_{\text{out, lake}} = \sum_{i=1}^{n_i} Q^{t_i}_{\text{out, lake}} \Delta t_i, \tag{114}$$

where $n_i$ refers to the number of iterations within model time step $t$ and $Q^{t_i}_{\text{in, lake}}$, $Q^{t_i}_{\text{out, lake}}$ and $\Delta t_i$ are as previously defined.

## 3 Computational performance

One of the reasons to switch to the Julia programming language is that it offers high performance, which is required for large-scale high-resolution hydrological model applications. Here we compare the simulation times of wflow_sbm between the Julia (van Verseveld et al., 2024) and Python (Schellekens et al., 2020) versions for three large catchments: the Moselle, Meuse and Rhine (Fig. 2). We used HydroMT-Wflow (Eilander et al., 2022) to set up the models for the three catchments at a resolution of 30″ ($\sim 1\,\text{km} \times 1\,\text{km}$). The models were run at a daily time step for 5 years (2000–2005) with ERA5 forcing data. We excluded the input/output (I/O) operations to allow for a clean comparison between the Julia and Python version and ran the simulation on a machine with the following specifications: a desktop with an Intel Xeon Gold 6144 CPU (with four cores, four threads exposed to the user) and 16 GB RAM.

The switch to Julia results in substantially smaller simulation times, independent of the size of the catchment (Fig. 2). By enabling threads (Julia version), the simulation times decrease further, leading to a model that runs 4–5 times faster compared to the Python version. For the Rhine catchment, the simulation time for 1 year is 120 min for the wflow_sbm Python version; for the Julia version, this is 36 and 23 min with one and four threads, respectively. These simulation times take up most of the total computational time (e.g. $\sim 98\,\%$ for the Rhine model with Julia running on one and multiple threads). These results show that the wflow_sbm Julia version is suitable for large-scale high-resolution model applications.

## 4 Applications

The wflow_sbm model has been applied to a number of specific cases. Below we describe these specific applications and the a priori parameter estimation, including forcing, with HydroMT-Wflow for a variety of hydroclimates and hydrological processes.

## 4.1 Parameter estimation with HydroMT-Wflow

The estimation of wflow_sbm model parameters is based on earlier work by Imhoff et al. (2020) that focused on the Rhine Basin and on the development of the Iterative Hydrography Upscaling (IHU) method by Eilander et al. (2021) to derive flow direction and representative river length, slope and width parameters. Eilander et al. (2021) showed an improved accuracy with IHU-upscaled flow direction maps, applied to MERIT Hydro, compared to other often-used upscaling methods. Furthermore, for a case study of the Rhine Basin, Eilander et al. (2021) showed that with IHU applied to MERIT Hydro, errors in the timing and magnitude of simulated peak discharge compared to simulations at the native data resolution are minimized. Imhoff et al. (2020) used available point-scale (pedo)transfer functions (PTFs) from the literature to generate seamless parameter maps for the Rhine Basin. Following a multiscale parameter regionalization (MPR) technique (Samaniego et al., 2010), parameters were estimated at the original data resolution ("level 0"), and upscaled to the model resolution ("level 1") with upscaling operators. Although universal scaling rules for hydrological model parameters are not available, the correct upscaling operator is found when model parameters are characterized by a constant mean and standard deviation across different spatial resolutions. Additionally, model fluxes and states should be consistent across different spatial resolutions. For the Rhine Basin, Imhoff et al. (2020) found that modelled actual evapotranspiration fluxes were consistent across different spatial resolutions (1.2, 2.4, 3.6 and 4.8 km). Routed discharge in headwater basins was not consistent across scales (KGE decreased from the finest to the coarsest resolution), while for the main Rhine River, routed discharge was consistent. For recharge fluxes, relatively large differences were found for regions with high drainage densities.

The transfer function and upscaling operators to derive wflow_sbm model parameters for any region in the world are part of the HydroMT-Wflow software (Eilander et al., 2022) and are listed in Table 1. For two sensitive wflow_sbm model parameters, the temperature threshold ($s_{\text{fall}, T \text{ threshold}}$) and the multiplication factor $f_{\text{Kh0}}$, a PTF is not available (Imhoff et al., 2020). For $s_{\text{fall}, T \text{ threshold}}$ and $f_{\text{Kh0}}$, a uniform default value of 0.0 °C and 100.0 is applied (Table 2), respectively. The a priori parameter estimation for wflow_sbm provides a model setup without the need for much further calibration; in most cases only the model parameter $f_{\text{Kh0}}$ is tuned (e.g. Wannasin et al., 2021a; Sperna Weiland et al., 2021).

Table 1 also includes references to examples of global datasets that can be used to set up a wflow_sbm model with HydroMT-Wflow. For soil properties, SoilGrids (Hengl et al., 2017) at 250 m resolution is available. For land cover the datasets GlobCover 2009 (Arino et al., 2010) at 300 m resolution, VITO v2.0.2 (Buchhorn et al., 2019) at 100 m resolution and CORINE Land Cover (CLC) 2018 (European Environment Agency, 2018) are currently available.

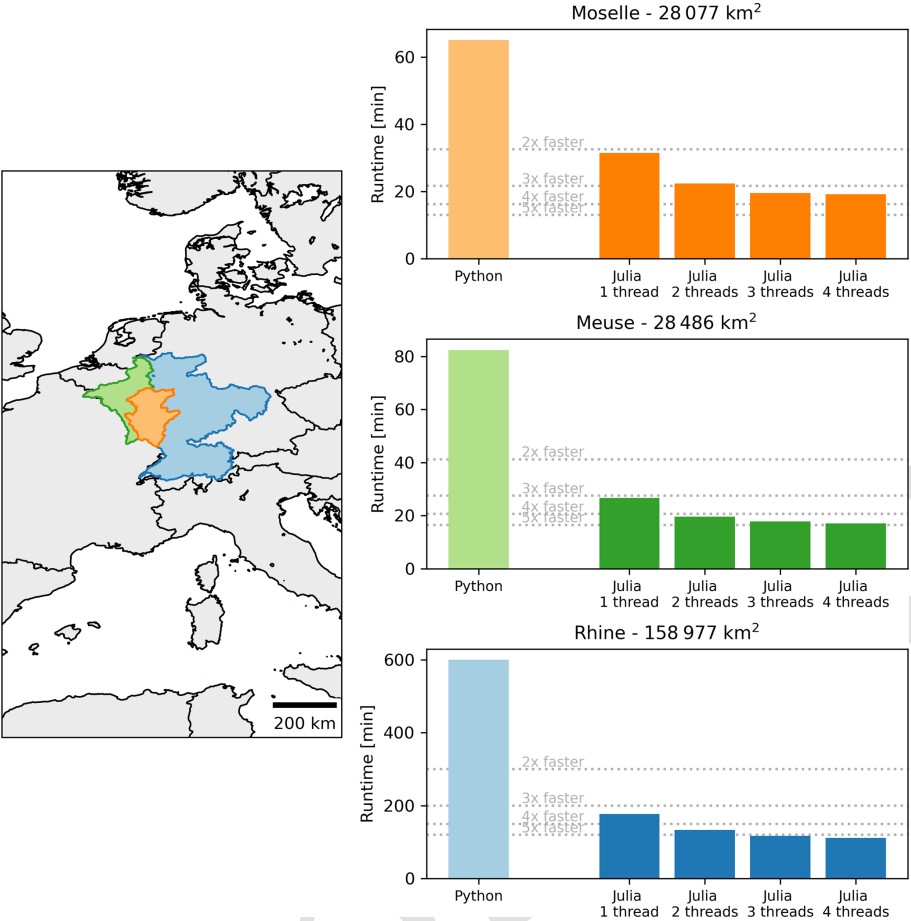

**Figure 2.** Simulation times of the wflow_sbm model in three large catchments with wflow Python version 2020.1.2 (Schellekens et al., 2020) and Wflow.jl v0.7.3 (van Verseveld et al., 2024), including multithreading in the Julia version.

Leaf area index climatology is based on the MCD15A3H Version-6 leaf area index product, at 500 m resolution (Myneni et al., 2015). For elevation and derived data, the MERIT-DEM-based hydrography map (MERIT Hydro; Yamazaki et al., 2019) at 90 m resolution is used. It contains a global flow direction map derived from the Multi-Error-Removed Improved-Terrain DEM dataset (MERIT DEM; Yamazaki et al., 2017) and a synthetic water layer map that consists of a combination of the Global "1-s" Water Body Map (G1WBM; Yamazaki et al., 2015), Global Surface Water Occurrence (GSWO; Pekel et al., 2016) and water-related features from OpenStreetMap. The fine-resolution MERIT Hydro flow direction map is upscaled to the wflow_sbm model resolution with the Iterative Hydrography Upscaling (IHU) method (Eilander et al., 2021). River width ($w_{\mathrm{river}}$) and bankfull depth ($h_{\mathrm{bankfull}}$) are based on MERIT Hydro (river mask based on a minimum upstream area) and the global reach-level bankfull river width dataset from Lin et al. (2019). River bankfull depth $h_{\mathrm{bankfull}}$ is estimated from bankfull discharge ($Q_{\mathrm{bankfull}}$) data in Lin et al. (2019) with the following power law relationship:

$$h_{\mathrm{bankfull}} = c\, Q_{\mathrm{bankfull}}^{p}, \tag{115}$$

with $c = 0.27$ (default) and $p = 0.30$ (default).

For glacier-related model parameters, the Randolph Glacier Inventory 6.0 (Pfeffer et al., 2014) is available. Lake-related parameters are derived from the HydroLAKES Version 1.0 (Messager et al., 2016) dataset, and reservoir parameters are based on a combination of the Global Reservoir and Dam Database (GRanD), Version 1, Revision 01 (v1.01) (Lehner et al., 2011); HydroLAKES Version 1.0 (Messager et al., 2016); and GSWO. For more details on the setup of a wflow_sbm model, from global (or regional/local) datasets with HydroMT-Wflow, we refer to the documentation and code of HydroMT-Wflow (Eilander et al., 2022).

Figure 3 shows examples of model parameter maps set up with HydroMT-Wflow for the Moselle catchment (see also Sect. 4.2.4). The parameters are related to elevation (slope), elevation and flow direction (Strahler stream order), land cover (rooting depth), and soil properties (vertical saturated hydraulic conductivity).

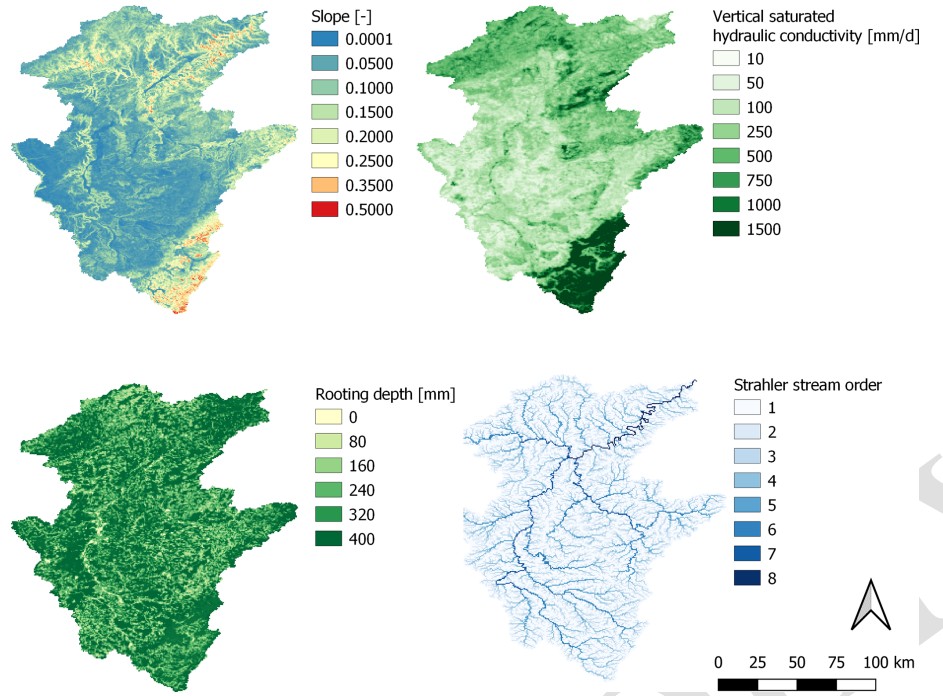

**Figure 3.** Wflow_sbm static model parameter maps slope, vertical saturated hydraulic conductivity, rooting depth and Strahler stream order for the Moselle catchment.

## 4.2 Wflow_sbm model cases

The wflow_sbm model cases have been set up with HydroMT-Wflow at a resolution of $30''$ based on the (pedo)transfer functions listed in Table 1 and HydroMT-Wflow constant default model parameters listed in Table 2. We present two model cases illustrating the model sensitivity to the model parameter horizontal hydraulic conductivity $f_{Kh0}$ (Whanganui catchment; see Sect. 4.2.1) and the parameter $f_{Kv}$ (Crystal River catchment; see Sect. 4.2.2) that controls the exponential decline in vertical saturated hydraulic conductivity. We also illustrate how wflow_sbm performs based on the a priori parameter estimation with only changing the model parameter $f_{Kh0}$ for the Umeälven catchment (Sect. 4.2.3), where reservoir operations and snow processes play an important role; for the Moselle catchment (Sect. 4.2.4) including discharge and catchment-average soil moisture as output; and for the Rhine catchment (Sect. 4.2.5) to demonstrate the ability of wflow_sbm to represent the spatial distribution of actual evapotranspiration $E_a$ and snow storage $S_{snow}$. Finally, we present a model case for the Ouémé catchment (Sect. 4.2.6), where groundwater loss plays an important role. The locations of the wflow_sbm model cases on a global map are shown in Fig. 4a. For each model case, a map of the catchment with elevation and rivers is presented in Fig. 4b–f. For each model case, four soil layers are defined as (default) 0–100, 100–400, 400–1200 and 1200 mm up to the maximum soil depth $z_{soil}$ (Table 1) to capture changes in soil hydraulic properties and roots, and thus

soil moisture fluxes, with depth. Wflow_sbm determines the actual soil layer thickness for each layer per grid cell based on $z_{soil}$. For river and overland flow, the time step is set to a fixed value of 900 and 3600 s, respectively. When snow is enabled, a reduction factor to infiltration in soils because of frozen conditions is not applied. Other more specific model settings are described per model case in Sect. 4.2.1–4.2.6.

The model performance of the wflow_sbm applications is here assessed with the modified Kling–Gupta efficiency (KGE; Kling et al., 2012) for discharge and catchment-average soil moisture as

$$KGE = 1 - \sqrt{(r-1)^2 + (\beta-1)^2 + (\gamma-1)^2}, \quad (116)$$

where $r$ is the correlation coefficient between observed and simulated values, $\beta$ is a bias term, and $\gamma$ is the variability ratio:

$$KGE = 1 - \sqrt{(r-1)^2 + \left(\frac{\mu_{sim}}{\mu_{obs}} - 1\right)^2 + \left(\frac{\sigma_{sim}/\mu_{sim}}{\sigma_{obs}/\mu_{obs}} - 1\right)^2}, \quad (117)$$

where $\mu_{sim}$ is the mean of simulated values, $\mu_{obs}$ is the mean of observed values, $\sigma_{sim}$ is the standard deviation of simulated values and $\sigma_{obs}$ is the standard deviation of observed values. $KGE = 1$ means a perfect match between simulated and observed values, and $KGE \approx -0.41$ indicates the model simulation is as accurate as the observed mean (Knoben et al., 2019). For the assessment of the reproduction of spatial patterns for the model case of the Rhine catchment, the spa-

**Table 1.** List of wflow_sbm parameters estimated with a (pedo)transfer function (PTF). Upscaling operators are abbreviated as follows: $A$ – arithmetic mean, $\log A$ – arithmetic mean of the natural logarithm.

| Parameter | Equation or section | The (pedo)transfer function | Upscaling operator | Additional notes |
|---|---|---|---|---|
| $c_n$ | Eqs. (48) and (49) | Rawls and Brakensiek (1989) applied to SoilGrids | $\log A$ | $\lambda_n$ upscaled with $\log A$, $c_n$ determined from $\lambda_n$ at model resolution |
| $k$ | Eq. (21) | van Dijk and Bruijnzeel (2001) | $A$ | Lookup table (land cover) |
| $K_{v0}$ | Eq. (50) | Brakensiek et al. (1984) applied to SoilGrids | $\log A$ | For the soil depths $z$: 0, 50, 150, 300, 600, 1000 and 2000 mm |
| LAI | Eqs. (20) and (21) | Myneni et al. (2015) | $A$ | LAI climatology from the period 2003–2017 |
| $f_{Kv}$ | Eq. (50) | | | Fitting exponential function between $K_{vz}$ and $z$ |
| $n_{land}$ | Eq. (94) | Engman (1986), Kilgore (1997) | $A$ | Lookup table (land cover) |
| $n_{river}$ | Eq. (94) | Liu et al. (2005) | $A$ | Derived at model resolution, lookup table (Strahler order) |
| $z_{rooting}$ | Eq. (70) | Schenk and Jackson (2002), Fan et al. (2016) | $A$ | $d_{75}$ rooting depth, lookup table |
| $S_{leaf}$ | Eq. (20) | Pitman (1989), Liu (1998) | $A$ | Lookup table (land cover) |
| $c_{land slope}$ | Eqs. (86) and (94) | Horn (1981) | $A$ | Derived from MERIT DEM |
| $c_{river slope}$ | Eq. (94) | | | Derived from MERIT Hydro |
| $x_{river}$ | Sect. 2.6 | | | Derived from MERIT Hydro |
| $w_{river}$ | Sect. 2.6 | Lin et al. (2019) | | River mask from MERIT Hydro |
| $h_{bankfull}$ | Sect. 2.6 | Lin et al. (2019) | | River mask from MERIT Hydro |
| $z_{soil}$ | Eq. (40) | Hengl et al. (2017), ESDAC (2004) | $A$ | |
| $S_{wood, max}$ | Sect. 2.2.3 | Pitman (1989), Liu (1998) | $A$ | Lookup table (land cover) |
| $\theta_s$ | Eq. (40) | Tóth et al. (2015) | $A$ | |
| $\theta_r$ | Eq. (40) | Tóth et al. (2015) | $A$ | |
| $f_{open water}$ | Eq. (60) | | $A$ | Lookup table (land cover) |
| $f_{paved}$ | Sect. 2.4.1 | | $A$ | Lookup table (land cover) |
| $S_{glacier}, f_{glacier}$ | Eq. (36) | Pfeffer et al. (2014) | | |
| $H_{lake, avg}, \alpha_{lake}, A_{lake}$ | Sect. 2.7.2 | Messager et al. (2016) | | Lake parameters, fixed $\beta_{lake} = 2$ |
| $A_{res}, S_{res}, f_{res, min}, f_{res, max}, Q_{min req.}, Q_{max, res}$ | Sect. 2.7.1 | Lehner et al. (2011), Messager et al. (2016), Pekel et al. (2016) | | Reservoir parameters |

tial efficiency metric ($E_{SP}$; Dembélé et al., 2020) is used:

$$E_{SP} = 1 - \sqrt{(r_s - 1)^2 + (\alpha - 1)^2 + (\gamma - 1)^2}, \text{ with} \quad (118)$$

$$r_s = 1 - \frac{6\sum_1^n d_i^2}{n(n^2 - 1)}, \quad (119)$$

$$\alpha = 1 - E_{RMS}(Z_{X_{sim}}, Z_{X_{obs}}) \text{ and} \quad (120)$$

$$\gamma = \frac{\sigma_{sim}/\mu_{sim}}{\sigma_{obs}/\mu_{obs}}, \quad (121)$$

where $r_s$ is the Spearman rank-order correlation coefficient; $d_i$ is the difference between the two ranks (observed variable $X_{obs}$ and simulated variable $X_{sim}$) of each cell $i$ in $n$ grid cells; $\alpha$ is a term that determines the spatial location matching, calculated as the root mean squared error ($E_{RMS}$) of the standardized values ($z$ scores, $Z_X$) of $X_{sim}$ and $X_{obs}$; and $\gamma$ is the variability ratio (equal to the $\gamma$ term of KGE). As with KGE, $E_{SP}$ ranges from $-\infty$ to 1 (a perfect match between $X_{sim}$ and $X_{obs}$).

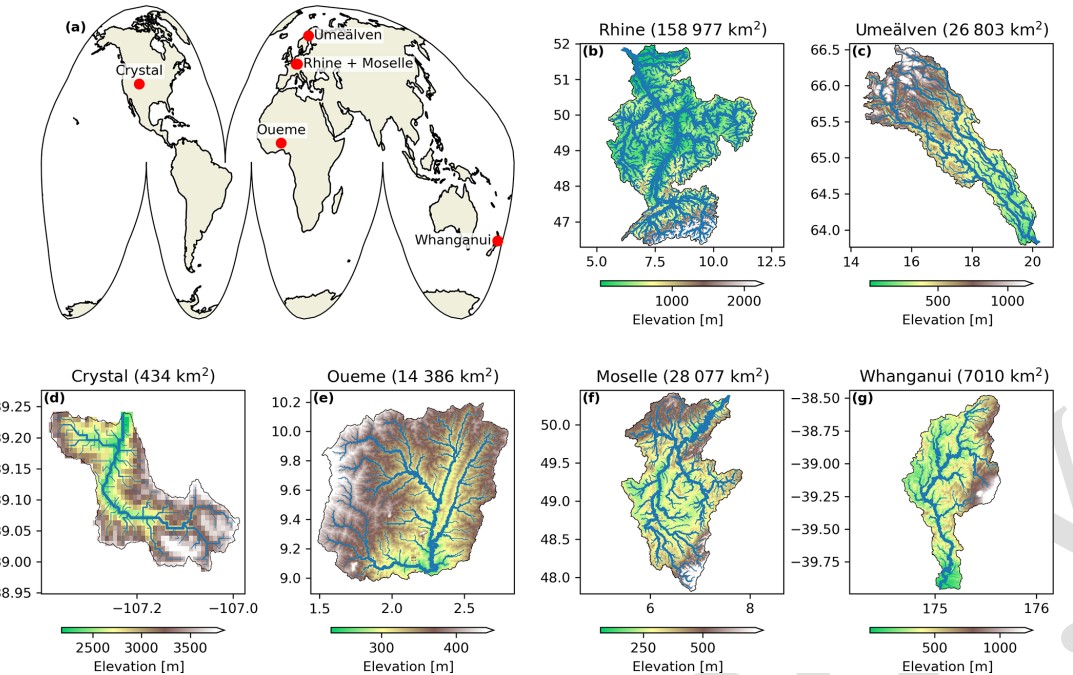

**Figure 4.** Locations and maps of wflow_sbm model cases: **(a)** model case locations on a global map, **(b)** Europe – Rhine (Sect. 4.2.5), **(c)** Europe – Umeälven (Sect. 4.2.3), **(d)** Europe – Moselle (Sect. 4.2.4), **(e)** USA – Crystal River (Sect. 4.2.2), **(f)** Africa – Ouémé River (Sect. 4.2.6) and **(g)** New Zealand – Whanganui River (Sect. 4.2.1).

**Table 2.** Constant wflow_sbm model parameter values defined in HydroMT-Wflow (Eilander et al., 2022).

| Parameter | Equation | Value |
| --- | --- | --- |
| $\frac{E_{sat}}{P_{sat}}$ | Eq. (1) | 0.11 |
| $c_{\text{infiltration, paved}}$ | Eq. (45) | $5.0\,\text{mm}\,\text{d}^{-1}$ |
| $c_{\text{infiltration, unpaved}}$ | Eq. (44) | $600.0\,\text{mm}\,\text{d}^{-1}$ |
| $f_{\text{red, frozen}}$ | Eq. (42) | 0.038 |
| $s_{\text{fall, }T\text{ threshold}}$ | Eq. (22) | $0.0\,°\text{C}$ |
| $s_{\text{fall, }T\text{ interval}}$ | Eq. (22) | $2.0\,°\text{C}$ |
| $s_{\text{ddf}}$ | Eqs. (25) and (26) | $3.75653\,\text{mm}\,\text{d}^{-1}\,°\text{C}^{-1}$ |
| $s_{\text{melt, }T\text{ threshold}}$ | Eqs. (25) and (26) | $0.0\,°\text{C}$ |
| $s_{\text{whc}}$ | Eq. (29) | 0.1 |
| $g_{\text{ddf}}$ | Eq. (34) | $5.3\,\text{mm}\,\text{d}^{-1}\,°\text{C}^{-1}$ |
| $g_{\text{snow to firn}}$ | Eq. (32) | $0.002\,\text{d}^{-1}$ |
| $g_{\text{melt, }T\text{ threshold}}$ | Eq. (34) | $1.3\,°\text{C}$ |
| $c_{\text{rd}}$ | Eq. (70) | $-500.0\,\text{mm}^{-1}$ |
| $f_{\text{Kh0}}$ | Eq. (86) | 100.0 |
| $L_{\text{max}}$ | Eq. (85) | $0.0\,\text{mm}\,\text{d}^{-1}$ |

For each model case, we use the first year as a warm-up period. For the simulations of all but three model cases, we use ERA5 forcing, temperature and potential evapotranspiration (using the de Bruin method; de Bruin et al., 2016), which are derived based on downscaled ERA5 fields using a fixed lapse rate of $-0.0065\,°\text{C}\,\text{m}^{-1}$. For the model case of the Crystal River catchment (Sect. 4.2.2), forcing is based on the dataset by Maurer et al. (2002); for the model case of the Rhine catchment (Sect. 4.2.5), we use the Multi-Source Weighted-Ensemble Precipitation version-2.8 (MSWEP V2.8) (Beck et al., 2019) global precipitation product instead of ERA5 rainfall; and for the case of the Ouémé catchment (Sect. 4.2.6), we use Climate Hazards group Infrared Precipitation with Stations (CHIRPS) rainfall (Funk et al., 2015) estimates instead of ERA5 rainfall.

### 4.2.1 New Zealand – Whanganui River – effect of model parameter $f_{\text{Kh0}}$

The wflow_sbm model parameter $f_{\text{Kh0}}$ is a multiplication factor applied to the vertical saturated hydraulic conductivity at the soil surface $K_{v0}$ to calculate the horizontal saturated hydraulic conductivity used for computing lateral subsurface flow. This parameter compensates for anisotropy, small-scale saturated hydraulic conductivity (soil core) measurements that do not represent larger-scale hydraulic conductivity and smaller flow length scales (hillslope) in reality not represented by the model resolution. Land-cover-derived model parameters are based on VITO v2.0.2 (Buchhorn et al., 2019). For this model case, the snow (including the snow avalanche routine) and glacier model are enabled. To illustrate the effect of different $f_{\text{Kh0}}$ values (1, 20, 50, 100), Fig. 5 shows the discharge simulation (daily time step) and KGE values for Global Runoff Data Centre (GRDC) station 5865600 of the Whanganui River catchment in New Zealand, for the year 1996. Figure 5 clearly shows that higher $f_{\text{Kh0}}$

values generally result in higher baseflow values and flattened peaks. The $f_{Kh0}$ value of 20 results in the highest KGE of 0.71 for the year 1996. For the complete period of simulation 1979–2009, the KGE values were 0.63, 0.79, 0.68 and 0.55 for $f_{Kh0}$ values of 1, 20, 50 and 100, respectively.

By changing the parameter $f_{Kh0}$, it is expected that the contribution of overland flow and lateral subsurface flow to river discharge will change. We show in Fig. 6a the effect of different $f_{Kh0}$ values (1 and 100) on the average lateral inflow components subsurface flow $Q_{\text{subsurface, to river}}$ and overland flow $Q_{\text{land, to river}}$ to the river. With an $f_{Kh0}$ value of 1, the contribution of $Q_{\text{subsurface, to river}}$ is minimal (maximum contribution is $0.0011\,\mathrm{m^3\,s^{-1}}$), and river inflow consists mainly of $Q_{\text{land, to river}}$, with high values during discharge peaks that quickly drop to low values under baseflow conditions (Fig. 6a). The average water table depth $z_{\text{watertable}}$ is low without much variation with an $f_{Kh0}$ value of 1 (Fig. 6b). With an $f_{Kh0}$ value of 100, the contribution of $Q_{\text{subsurface, to river}}$ is higher and $Q_{\text{land, to river}}$ is lower during peaks and higher under baseflow conditions compared to an $f_{Kh0}$ value of 1. On average $z_{\text{watertable}}$ is lower and shows more variation with an $f_{Kh0}$ value of 100 compared to an $f_{Kh0}$ value of 1 (Fig. 6b). Thus, $f_{Kh0}$ controls the distribution of $Q_{\text{subsurface, to river}}$ and $Q_{\text{land, to river}}$, related to the overall wetness of the catchment and the magnitude of lateral subsurface flows ($Q_{\text{subsurface}}$), which has an effect on the peak discharges and baseflow values of the hydrograph.

### 4.2.2 USA – Crystal River – effect of exponential decline in $K_{v0}$

The wflow_sbm parameter $f_{Kv}$ controls the exponential decline in the vertical saturated hydraulic conductivity $K_{v0}$ at the soil surface with depth and thus vertical flow and lateral subsurface flow. A priori, $f_{Kv}$ is estimated with HydroMT-Wflow through the use of two different fitting methods using non-linear least squares (curve_fit) and a least-squares solution (linalg.lstsq) applied to the estimated vertical saturated hydraulic conductivity $K_{vz}$ at different depths from SoilGrids (see also Table 1). Figure 7 shows simulated (daily time step) discharge for the Crystal River near Redstone (Colorado, USA) in 2003, with land-cover-derived model parameters based on VITO v2.0.2 (Buchhorn et al., 2019). For this model case, the snow (including the snow avalanche routine) model is enabled. For both fitting methods, the performance is good (KGE 0.9 or higher); the fitting method using non-linear least squares (curve_fit) results in a higher KGE value. This fitting method captures the rising limb (partially caused by snowmelt) during the period 15 March–15 May and generally the falling limb (less overestimation) of the hydrograph better. The average $f_{Kv}$ value for the catchment was 0.0027 and 0.0011 for fitting using non-linear least squares and the least-squares solution, respectively. A higher $f_{Kv}$ value results in a more exponential decline in $K_{v0}$ with depth.

As with the parameter $f_{Kh0}$, it is expected that by changing the $f_{Kv}$ parameter the contribution of overland flow and lateral subsurface flow to river discharge will change. We show the effect of the different $f_{Kv}$ values as a result of different fitting methods on the average lateral inflow components $Q_{\text{subsurface, to river}}$ and $Q_{\text{land, to river}}$ to the river in Fig. 8a and on $z_{\text{watertable}}$ in Fig. 8b. The curve_fit fitting method captures the rising limb of the hydrograph better during the period 15 March–15 May because the contribution of $Q_{\text{land, to river}}$ is higher compared to the linalg.lstsq fitting method, while the $Q_{\text{subsurface, to river}}$ contribution is similar (Fig. 8a). With the curve_fit fitting method, the catchment is wetter; $z_{\text{watertable}}$ has a shallower pattern compared to the linalg.lstsq fitting method (Fig. 8b), causing higher $Q_{\text{land, to river}}$ values. During the falling limb of the hydrograph, the curve_fit fitting method results in less overestimation caused by a lower $Q_{\text{subsurface, to river}}$ contribution, while the $Q_{\text{land, to river}}$ contribution is similar (Fig. 8a).

### 4.2.3 Europe – Umeälven – snow and reservoirs

The Ume River (Umeälven) is one of the largest rivers in Sweden and has been extensively cultivated for hydroelectric power. From a hydrometeorological aspect, snowfall and snowmelt play an important role here. Furthermore, accounting for these hydropower stations in the hydrological model is done on the basis of the information in the GRanD, HydroLAKES and GWSO datasets. This information is rather uncertain as hydropower operators have their own release curves or optimization schemes. Here, we simulated discharge for the period 1979 to 2019 at a daily time step, with land-cover-derived model parameters based on CLC 2018 (European Environment Agency, 2018). The values derived for the vertical saturated hydraulic conductivity are quite large, on the order of a few metres per day, and hence no anisotropy factor for horizontal saturated hydraulic conductivity was applied ($f_{Kh0} = 1$). For this model case, the snow (including the snow avalanche routine) and glacier models are enabled and lakes and reservoirs are included.

Figure 9 presents KGE scores for Swedish Meteorological and Hydrological Institute (SMHI) stations simulated with E-HYPE (Swedish Meteorological and Hydrological Institute, 2022) and wflow_sbm. Observed discharges for the stations were obtained from Swedish Meteorological and Hydrological Institute (2024). Because of differences in forcing, simulation periods and the use of KGE methods, a quantitative comparison of model performance scores is not possible. However, it is obvious that for locations (20010 and 20047) largely influenced by reservoirs and lakes, and thus reservoir operations, both models show poor performance. For a more upstream location like station 1630, E-HYPE performance is good (KGE > 0.8) and wflow_sbm performance is satisfactory (KGE = 0.66). Wflow_sbm shows lower KGE values for locations 2237 and 2238, while for locations 1733 and 1734 wflow_sbm shows a better performance. Wflow_sbm per-

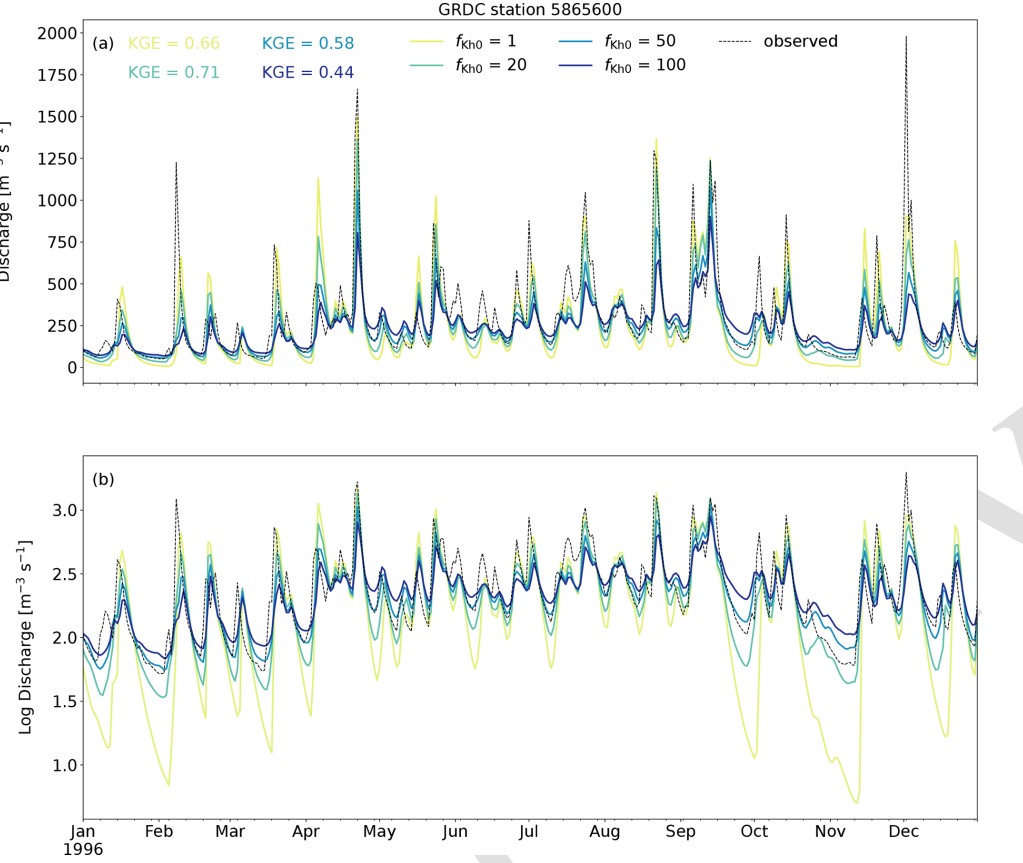

**Figure 5.** Effect of $f_{Kh0}$ on simulated discharge for GRDC station 5865600 of the Whanganui River catchment in 1996: **(a)** discharge and **(b)** $\log_{10}$ of the discharge.

formance for the unregulated Vindel River (locations 1630, 2237 and 2238), a tributary to the Ume River, could probably be improved on the basis of E-HYPE performance. This could be done for example through a further analysis of the impact of different snow model parameters like $s_{fall, T \, threshold}$ and $s_{ddf}$ on model performance, lake model settings of Lake Storovan, and possibly forcing datasets other than ERA5.

Figure 10 shows simulated and observed discharge for the SMHI stations within the Umeälven catchment for the year 1993. The performance for the most downstream stations 1733 and 1734 is satisfactory. The underestimation of discharge peaks for station 1733 during July and August is mostly caused by underestimating discharge peaks at station 20010 (downstream of Lake Storuman) during the same period. The actual release scheme of Lake Storuman is very likely not captured well enough by wflow_sbm. Wflow_sbm overestimates baseflow and underestimates peak flows for station 2238 (Fig. 10), downstream of Lake Storovan. This may be caused for example by the lake model settings of Lake Storovan or upstream model parameters related to lateral subsurface flow (see also Sect. 4.2.2 and 4.2.1) or snow dynamics, and it has an effect on the underestimation of peak

flows at the downstream station 1734 by wflow_sbm. Overall, we show that with the a priori parameter estimation, the performance for the Umeälven catchment is (qualitatively) similar to the E-HYPE performance for this catchment.

### 4.2.4 Europe – Moselle – soil moisture

Besides comparing simulated discharge with observed discharge, it can be useful to compare model state variables like soil moisture or snow water equivalent to actual observations or satellite-based datasets for the purposes of calibration and validation of the hydrological model. Here we perform simulation for the period 1979 to 2019 at a 6-hourly time step, with land-cover-derived model parameters that are based on CLC 2018 (European Environment Agency, 2018). We use an $f_{Kh0}$ value of 250 based on previous hydrological modelling work of the Rhine Basin in Imhoff et al. (2020). For this model case, the snow (including the snow avalanche routine) model is enabled and lakes and reservoirs are included. The simulated soil moisture dynamics of the first soil layer (0–10 cm) are compared to the SMAP Enhanced L3 Radiometer dataset (O'Neill et al., 2021), averaged over the catchment, for the period 2015 to 2019. Figure 11 shows

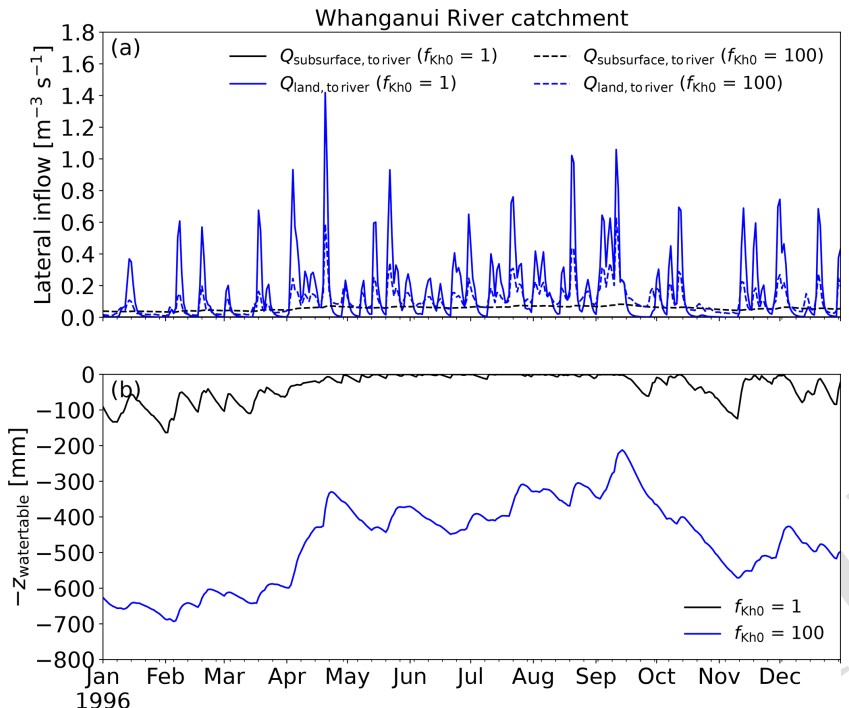

**Figure 6.** Simulated **(a)** average lateral inflow (subsurface flow $Q_{\text{subsurface, to river}}$ and overland flow $Q_{\text{land, to river}}$) and **(b)** average water table depth $z_{\text{watertable}}$, of the Whanganui River catchment in 1996, with $f_{\text{Kh0}} = 1$ and $f_{\text{Kh0}} = 100$.

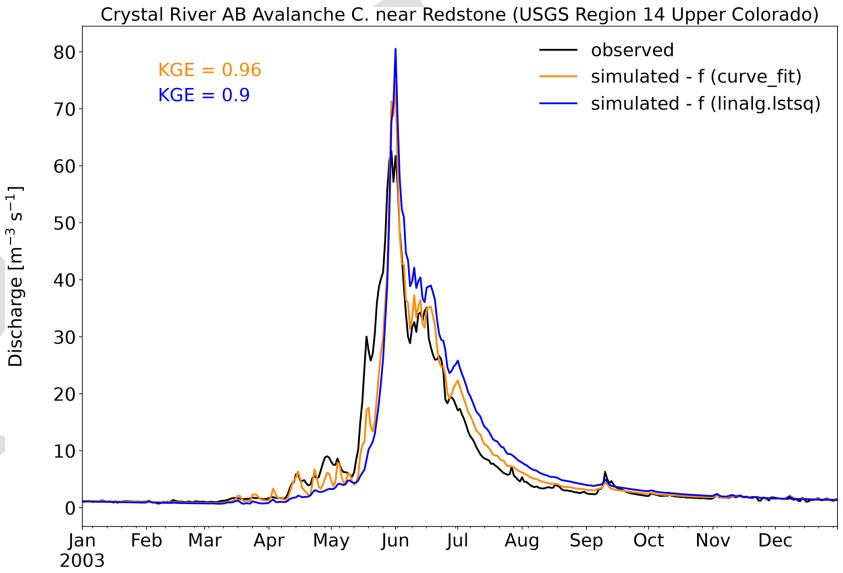

**Figure 7.** Simulated discharge for the Crystal River near Redstone (Colorado, USA) in 2003, with different fitting methods for $f_{\text{Kv}}$.

that wflow_sbm captures the average soil moisture dynamics quite well, with a KGE score of 0.83. Some of the lower and higher soil moisture values of the SMAP Enhanced L3 Radiometer dataset are not captured by wflow_sbm. This could be caused by using the default first soil layer thickness of 10 cm here, while the SMAP Enhanced L3 Radiometer dataset represents the top 5 cm of the soil column. Addi-

tionally, difference between wflow_sbm and the SMAP Enhanced L3 Radiometer in the saturated and residual water contents used can also play a role; for example the average saturated and residual water content for the wflow_sbm model is 0.44 and 0.17, respectively, while the SMAP soil moisture product shows values outside of this range (Fig. 11). Figure 12 shows simulated and observed daily discharge

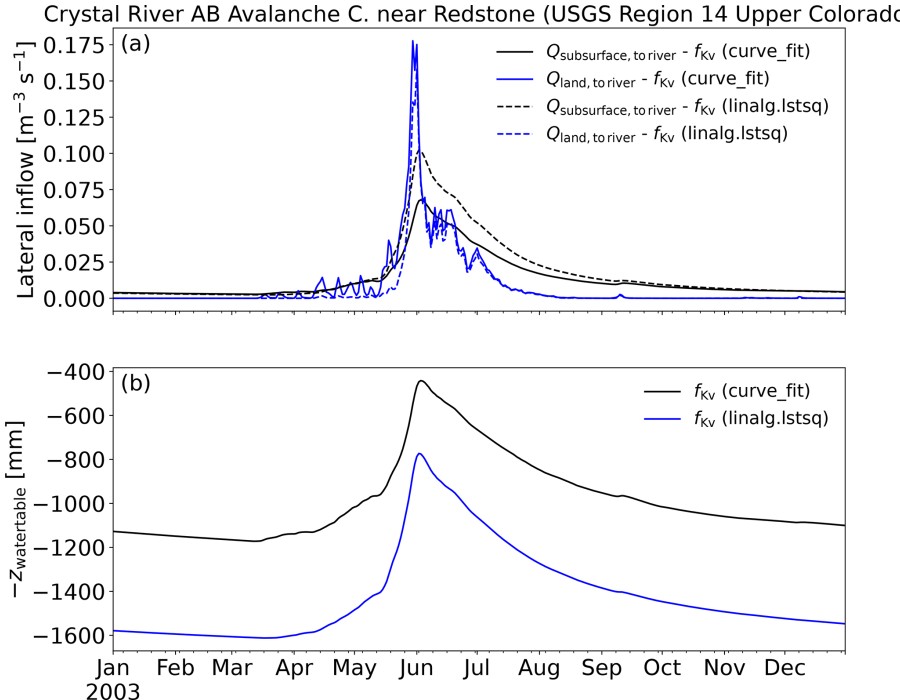

**Figure 8.** Simulated **(a)** average lateral inflow (subsurface flow $Q_{\text{subsurface, to river}}$ and overland flow $Q_{\text{land, to river}}$) and **(b)** average water table depth $z_{\text{watertable}}$, for the Crystal River near Redstone (Colorado, USA) in 2003, with different fitting methods for $f_{\text{Kv}}$.

for GRDC station 6336050 (Cochem, near the outlet of the Moselle River into the Rhine River), for the period 2007–2008, with a KGE score of 0.74. The KGE score for GRDC station 6336050 for the complete simulation period is 0.71, and thus the overall performance of simulated soil moisture dynamics and discharge at the catchment scale is good.

### 4.2.5 Europe – Rhine – actual evapotranspiration and snow storage

As a spatially distributed hydrological model, wflow_sbm can easily make direct use of spatial datasets for model calibration, evaluation and data assimilation. Because of the increasing availability of satellite-based earth observations, also at finer temporal and spatial resolutions, hydrological modelling studies increasingly make use of these datasets for calibration, evaluation and data-assimilation purposes (e.g. López López et al., 2016; Demirel et al., 2018; Dembélé et al., 2020). Here, we demonstrate the ability of wflow_sbm to represent the spatial distribution of actual evapotranspiration $E_a$ and snow storage $S_{\text{snow}}$, with the a priori estimation of the model parameters and only changing the $f_{\text{Kh0}}$ parameter, for the Rhine catchment. Simulation is for the period 2014 to 2019 at a daily time step, with land-cover-derived model parameters that are based on CLC 2018 (European Environment Agency, 2018). We use a regionally optimized $f_{\text{Kh0}}$ map, initially based on hydrological modelling work of the Rhine Basin in Imhoff et al. (2020) with further im-

provements as part of hydrological modelling work for Rijkswaterstaat (part of the Dutch Ministry of Infrastructure and Water Management). For this model case, the snow (including the snow avalanche routine) and glacier models are enabled and lakes and reservoirs are included. The KGE score for GRDC station 6435060 (Lobith) at the Dutch–German border for the complete simulation period is 0.85. For the actual evapotranspiration comparison, we use The Global Land Evaporation Amsterdam Model (GLEAM) v3.7b daily actual evapotranspiration data with a spatial resolution of $\sim 28$ km (Martens et al., 2017; Miralles et al., 2011) for the period 2015 to 2019. For the forcing variable precipitation, GLEAM and wflow_sbm both use the global precipitation product MSWEP V2.8. Wflow_sbm $E_a$ is upscaled to the GLEAM resolution using average resampling. Figure 13 shows the average annual $E_a$ for the period 2015–2019 for wflow_sbm and GLEAM. The spatial variability is quite similar ($\gamma = 1.07$) and the spatial correlation is moderate ($r_s = 0.70$), but the matching of the spatial location of grid cells ($\alpha = 0.19$) is not satisfactory, leading to an $E_{\text{SP}}$ of 0.13. GLEAM and wflow_sbm both show a higher region of $E_a$ in northern Switzerland, which is characterized by lower elevations and flatter terrain than the southern part of Switzerland. For other parts in the Rhine catchment GLEAM shows spatial clusters with higher $E_a$ values that are not matched by wflow_sbm. While an $E_{\text{SP}}$ value of 0.13 may seem low, $E_{\text{SP}}$ can be considered a tough criterion. Koch et al. (2018) identified the SPAtial EFficiency

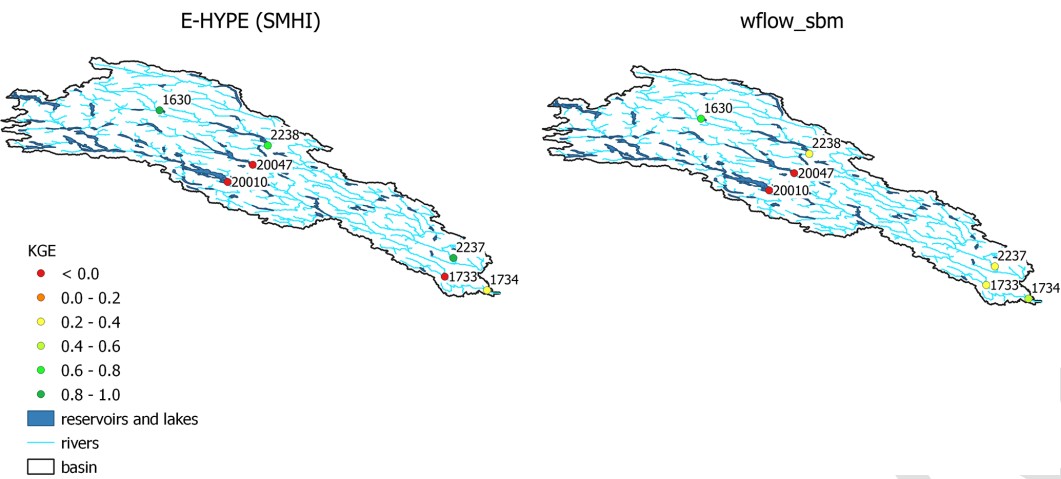

**Figure 9.** KGE scores for stations within the Umeälven catchment with E-HYPE and wflow_sbm. E-HYPE and wflow_sbm performance is assessed with KGE from Gupta et al. (2009) and the modified KGE from Kling et al. (2012), respectively.

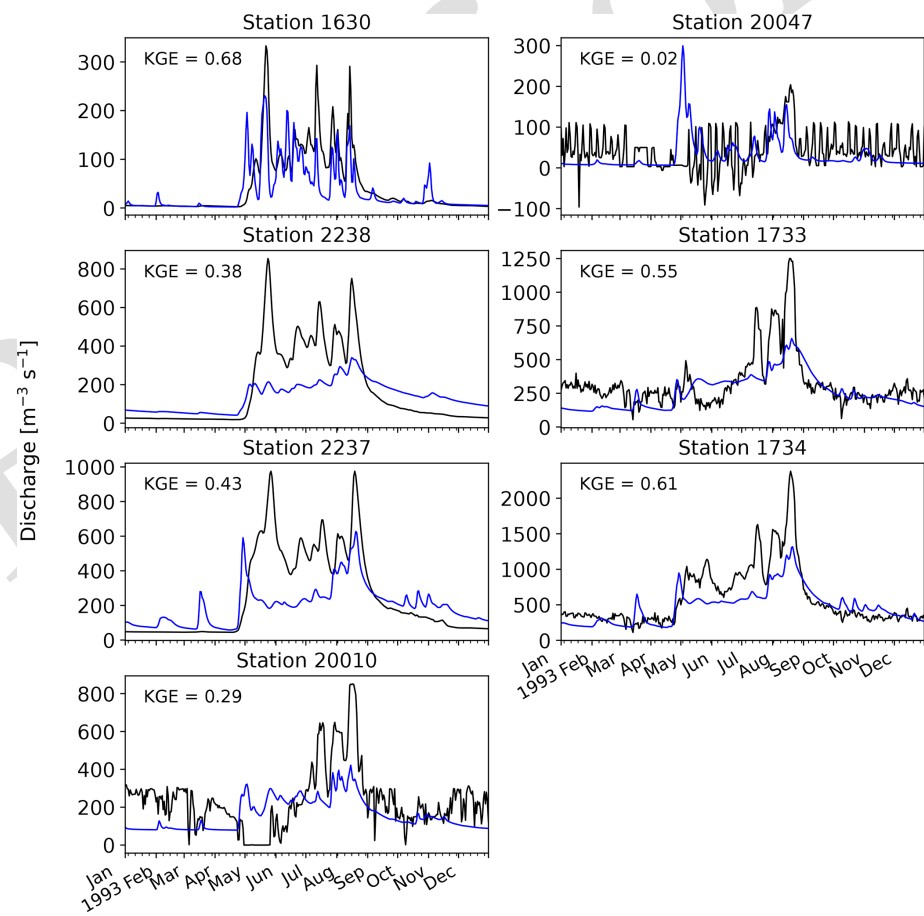

**Figure 10.** Simulated (blue) and observed (black) discharge for the SMHI stations within the Umeälven catchment for the year 1993.

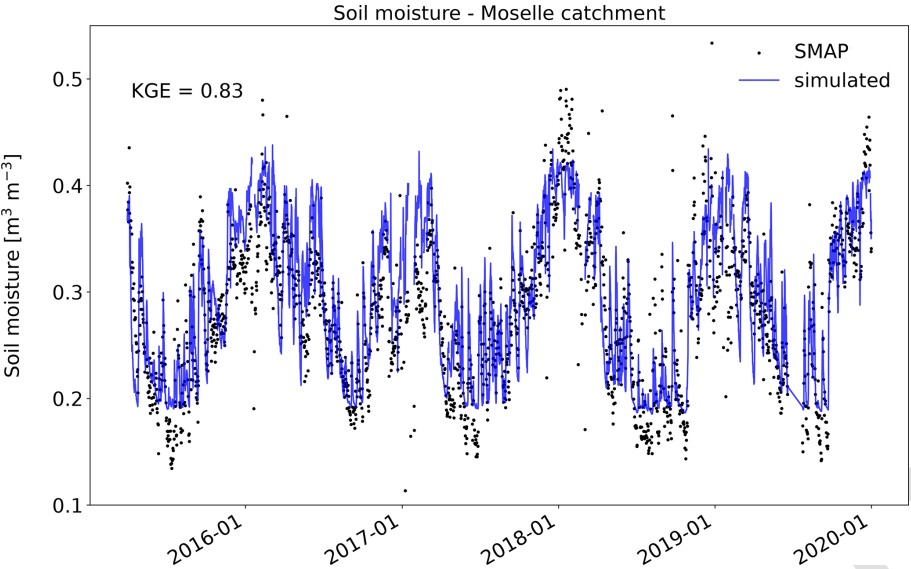

**Figure 11.** Catchment-average-simulated and SMAP soil moisture for the Moselle catchment. The date format is year-month.

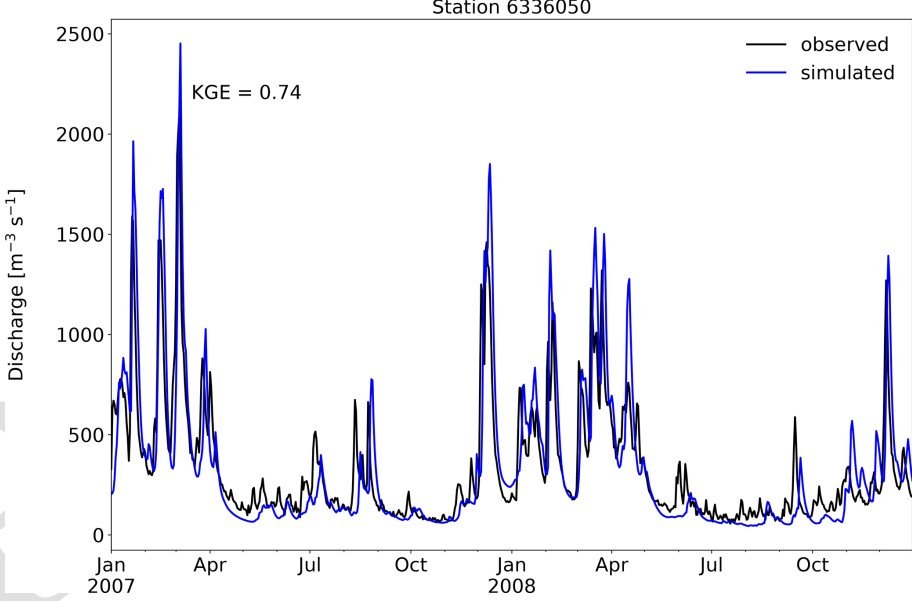

**Figure 12.** Simulated and observed discharge for GRDC station 6336050 (Cochem).

metric (SPAEF) as such, a spatial efficiency metric equivalent to $E_{SP}$. Additionally, according to Koch et al. (2018) more detailed investigation into the relationship between spatial variability and spatial efficiency metrics such as SPAEF and $E_{SP}$ is required to be able to put these metrics into context, for example for the comparison of different models or catchments. The temporal KGE scores for each grid cell (not shown) show a satisfactory to good performance; the median KGE is 0.79, with a range between 0.54 and 0.91, indicating wflow_sbm can represent $E_a$ from GLEAM. Comparing catchment-average daily $E_a$ from GLEAM and wflow_sbm

(Fig. 14) shows a good agreement (KGE = 0.87), with generally an underestimation of $E_a$ by wflow_sbm during the months July and August, and GLEAM showing more variability during autumn and winter. Overall, these results indicate that wflow_sbm can represent daily $E_a$ variability from GLEAM, although the matching of the spatial location of grid cells is not well represented. This might be caused by, for example, the difference in resolution between GLEAM and wflow_sbm (especially in regions with complex terrain), the potential-evaporation method used by each model (GLEAM uses the Priestley–Taylor equation), irrigation (not

(yet) included in wflow_sbm, while in GLEAM this is partly corrected for by the assimilation of satellite soil moisture; Miralles et al., 2011), and differences in snow evaporation approaches between wflow_sbm (only evaporation of inter-
5 cepted snow) and GLEAM (snow evaporation based on a modified Priestley–Taylor equation).

The C-SNOW project provides Sentinel-1 (S1) snow depth observations over the Northern Hemisphere mountains at a spatial resolution of ∼ 1 km (Lievens et al., 2019, 2022). For
the European Alps the S1 snow depth dataset of Lievens et al. (2019) was improved (Lievens et al., 2022) over the period 2017–2019, and we use this dataset at ∼ 1 km resolution for the comparison with wflow_sbm $S_{snow}$. Because C-SNOW provides snow depth observations and $S_{snow}$ rep-
resents snow water equivalent, a direct comparison of spatial patterns is not feasible with the spatial efficiency metric $E_{SP}$. Therefore, we determine the temporal correlation for each grid cell (Fig. 15) and the spatio-temporal correlation between S1 snow depth retrievals and $S_{snow}$, estimated with
the Spearman rank-order correlation coefficient $r_s$. The S1 snow depth retrievals are matched to the wflow_sbm grid using nearest-neighbour resampling, excluding wet snow conditions detected by S1. The spatio-temporal correlation between S1 snow depth retrievals and $S_{snow}$ is 0.88. Figure 15
shows generally a strong to a very strong positive correlation (median of $r_s = 0.87$), except in the valleys. Lievens et al. (2022) indicated that the main uncertainties in S1 snow depth retrievals are mainly caused by wet, shallow and occasional snow cover and forest cover. Regional model simulations of
snow depth to assess the S1 snow depth retrievals showed a strong positive temporal correlation if the maximum snow depth reached above ∼ 1 m at an elevation above ∼ 1000 m or with a forest cover fraction below ∼ 80 % (Lievens et al., 2022). A similar pattern is revealed by the temporal corre-
lation between the S1 snow depth retrievals and wflow_sbm $S_{snow}$ in Fig. 15.

### 4.2.6 Africa – Ouémé River – groundwater loss

The Ouémé mesoscale site (Benin), Africa, is part of the AMMA-CATCH observation network covering a 14 000 km²
basin in Sudanian climate on a crystalline basement. It is an interesting test case for wflow_sbm and the automated model setup including a priori estimation of the model parameters. Various studies using a variety of hydrological model concepts, including similar model concepts to wflow_sbm,
have been conducted (see Cornelissen et al., 2013, and references therein) for this area. Séguis et al. (2011) reported that around 15 % of water is being lost to the groundwater which is disconnected from the river system. Here, we run the wflow_sbm model for the period 1981 to 2019 at a daily
time step both without and with groundwater loss ($L_{max} = 0 \, \mathrm{mm \, d^{-1}}$ vs. $L_{max} = 0.6 \, \mathrm{mm \, d^{-1}}$; 15 % of ∼ 4 mm d⁻¹ of annual average daily rainfall) and analyse the model results for a variety of locations. $L_{max}$ represents a maxi-

mum groundwater loss value, and the actual groundwater loss (computed by wflow_sbm) is controlled by the verti-
55 cal hydraulic conductivity at the soil bottom and the saturated store (see Eq. 85) and may vary spatially and in time. Land-cover-derived model parameters are based on VITO v2.0.2 (Buchhorn et al., 2019). For this model case, the snow model is disabled and lakes and reservoirs are not included.
Figure 16 shows the KGE scores of discharge for the stations within the Ouémé mesoscale site without and with groundwater loss. Generally the performance of wflow_sbm increases with groundwater loss with a median increase of 0.86 for all stations. Figure 17 shows simulated discharge for
the station TEBOU for 2010 with and without groundwater loss. The simulation with groundwater loss clearly shows less overestimation of discharge during the rising limb and peaks of the hydrograph, also reflected in the higher KGE score for the simulation with groundwater loss.

While the simulation with groundwater loss shows generally a better performance, we expect further improvement is possible by checking the effect of different groundwater loss values. For example, during the start of the rising limb of the hydrograph, the simulation underestimates the discharge for
2010 (Fig. 16) and other years (not shown). Applying a lower uniform groundwater loss value (we use the upper range of 15 % reported by Séguis et al., 2011) or applying spatially distributed groundwater loss values based on discharge measurements and a water balance approach for the upstream
area (sub-catchment) of each station could further improve simulation results.

### 5  Conclusions and future work

We presented the wflow_sbm hydrological model as part of the Wflow.jl (v0.7.3) open-source modelling framework
for distributed hydrological modelling in Julia, a continuation of the wflow development in the PCRaster Python framework. Wflow_sbm has been applied in various catchments around the world with satisfactory to good performance. With wflow_sbm we aim to strike a balance be-
tween low-resolution, low-complexity and high-resolution, high-complexity hydrological models. Most wflow_sbm parameters are based on physical characteristics, and at the same time wflow_sbm has a runtime performance well suited for large-scale high-resolution modelling. We demon-
strated some examples of wflow_sbm applications with Wflow.jl (v0.7.3), using HydroMT-Wflow to set up the wflow_sbm model, including forcing, in an automated way. The wflow_sbm applications illustrate that the a priori model parameter estimates in combination with a manual adjust-
ment of the $f_{Kh0}$ model parameter result in generally satisfactory to good performance for discharge and catchment-average soil moisture (Moselle catchment), with, for the Umeälven catchment, similar performance for discharge (qualitatively) to E-HYPE. For the Rhine catchment we

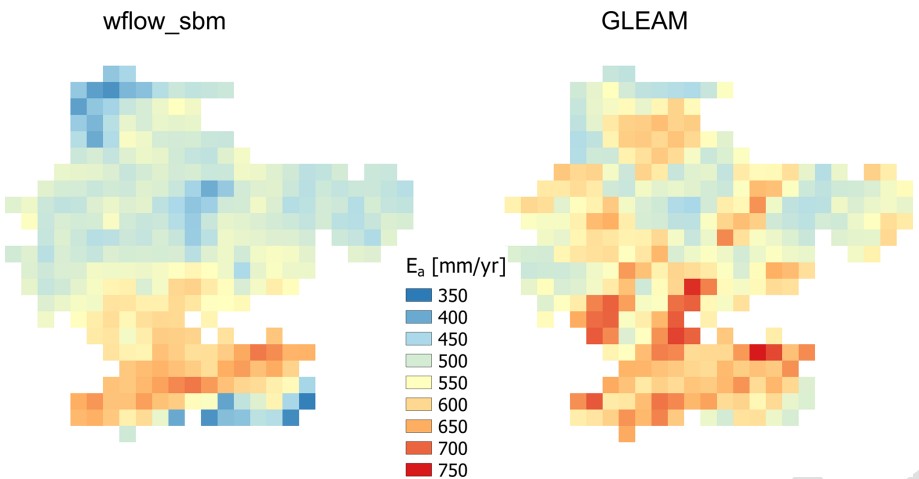

**Figure 13.** Wflow_sbm simulated and GLEAM long-term (2015–2019) average annual evapotranspiration ($E_a$) for the Rhine catchment.

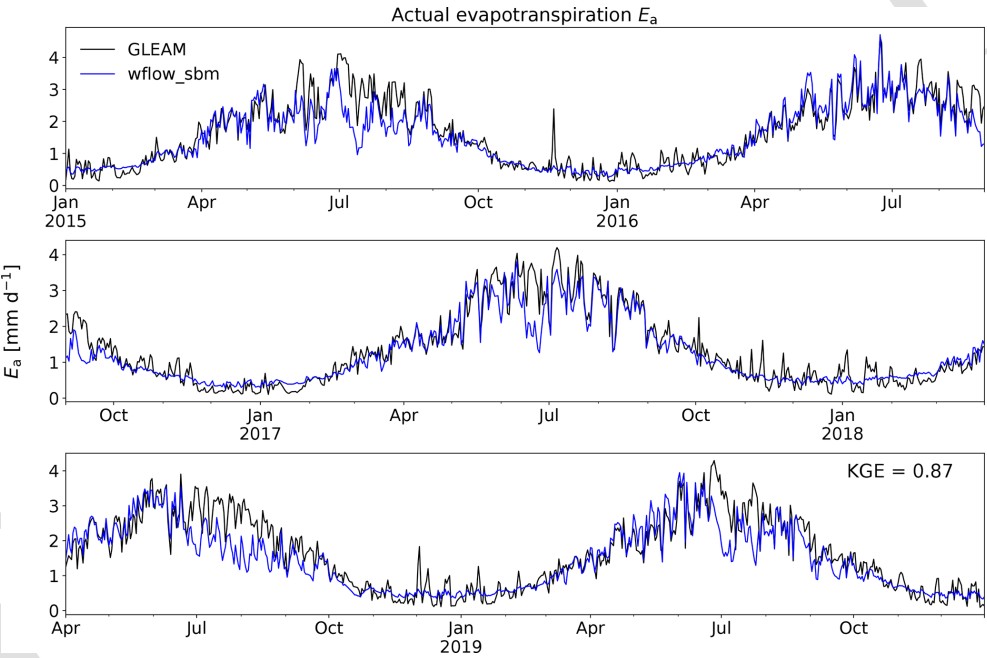

**Figure 14.** Wflow_sbm simulated and GLEAM catchment-average daily evapotranspiration ($E_a$) for the Rhine catchment for the period 2015–2019.

demonstrated the ability of wflow_sbm to represent the spatial distribution of actual evapotranspiration $E_a$ from GLEAM and snow storage $S_{snow}$ for the Alps by comparing to C-SNOW S1 snow depth observations. Wflow_sbm can represent daily $E_a$ variability from GLEAM, although the spatial location matching is not satisfactory, which could be due to, amongst other things, the difference in resolution between GLEAM and wflow_sbm and different or missing process representations such as snow evaporation and irrigation. For the comparison of $S_{snow}$ and S1 snow depth retrievals, a good performance indicated by the Spearman rank-order correlation coefficient (temporal (median of $r_s = 0.87$) and

spatio-temporal ($r_s = 0.88$)) is obtained. The Ouémé River case illustrates the use of the model parameter $L_{max}$ (maximum leakage) based on data from the literature, resulting in an overall significant increase in model performance. Including the process of leakage to deeper groundwater results in loss of water outside of the model domain, and we recommend including this process only if the scientific literature or geological data indicate that leakage to deeper groundwater is of importance. With the a priori parameter estimation, a working wflow_sbm model is set up quickly and incorrect process representations become apparent. The results, for example for the Ouémé River, indicate that local information

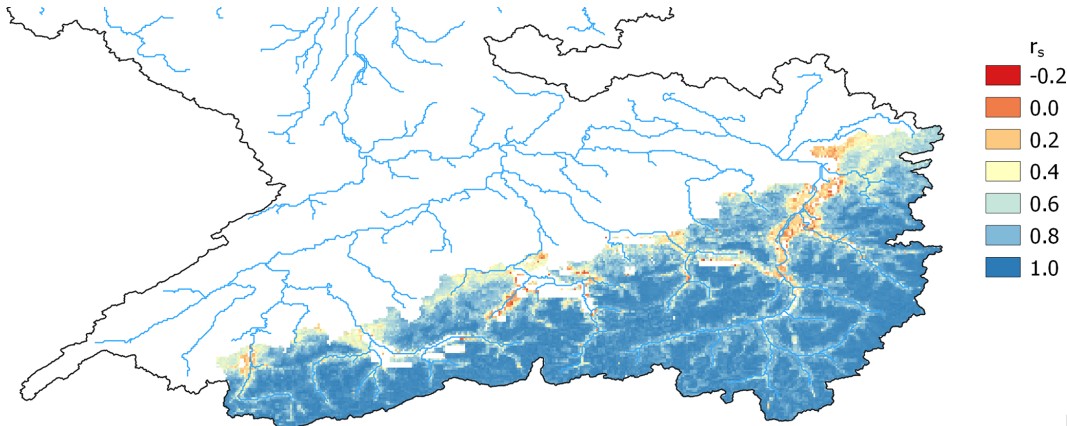

**Figure 15.** Temporal correlation (Spearman rank-order coefficient $r_s$) for each grid cell between wflow_sbm $S_{snow}$ and S1 snow depth retrievals from C-SNOW for the Alps in the Rhine catchment for the period 2017–2019.

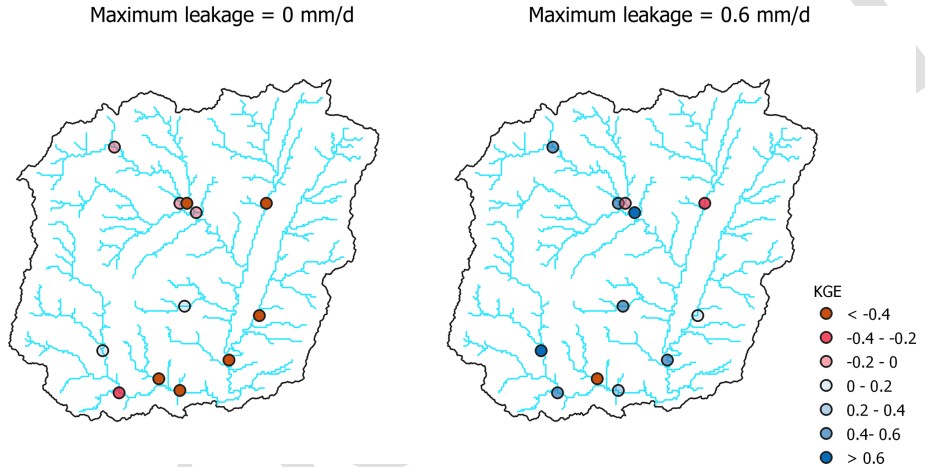

**Figure 16.** KGE scores of discharge for stations within the Ouémé mesoscale site without and with groundwater loss.TS6

and literature studies can help in improving process representation, and if they cannot, this opens the way for a better focus on the missing process representation. This is something that is lacking when a hydrological model is directly calibrated for a given catchment.

While (pedo)transfer functions are available for most of the sensitive wflow_sbm model parameters, this is not the case for the $f_{Kh0}$ model parameter. An interesting approach, as part of future work focusing on model data, could be to develop a transfer function for this model parameter by estimating the transfer function with function space optimization (FSO), a method presented by Feigl et al. (2020). Relevant hydrological processes such as glacier and snow processes, evapotranspiration processes, unsaturated zone dynamics, (shallow) groundwater, and surface flow routing including lakes and reservoirs are part of wflow_sbm. Floodplain dynamics (backwater effects and floodplain storage) are not part of the kinematic-wave routing in Wflow.jl, and this may be problematic for accurately simulating discharge and water depths when backwater effects and floodplain storage cannot be ignored (e.g. Zhao et al., 2017). Additionally, the kinematic-wave approach is mostly applicable when slopes are steep and less reliable for low-gradient rivers. A recent wflow_sbm development, which is part of v0.7.3, is the improvement of the routing scheme for river-floodplain dynamics and the improvement of discharge and water depth estimates for low-gradient rivers. The improved routing scheme includes the following options: (1) a 1-D local inertial solution for river channel flow and a 2-D local inertial solution for floodplain and overland flow, similar to Neal et al. (2012), and (2) a 1-D local inertial solution for river channel flow with optional 1-D floodplain schematization. Future possible developments related to the improved local inertial routing scheme are (1) to improve the multithreaded performance for the 2-D local inertial solution, (2) vector-based routing (e.g. Mizukami et al., 2021) allowing for more flexible channel routing configurations that are less computationally intensive and (3) to combine different routing solu-

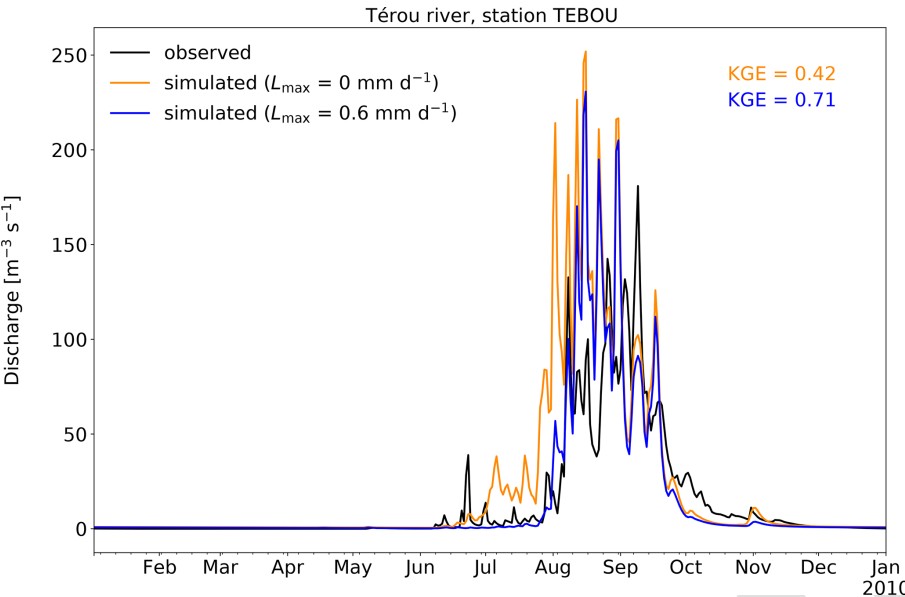

**Figure 17.** Simulated (with and without groundwater loss) and observed discharge for the station TEBOU for 2010.

tions (e.g. kinematic wave and local inertial) at the submodel-domain scale (e.g. local inertial for the floodplain). For water resource modelling studies, wflow_sbm is often linked to a network-based water allocation model (e.g. Meijer et al., 2021). The development of a water demand module (irrigation, livestock, industrial and domestic) and water allocation module is foreseen to fully exploit the gridded capabilities of wflow_sbm. The standard soil column of wflow_sbm extends to 2 m below surface level based on SoilGrids data, and although the soil column depth can be increased, the process modelled by wflow_sbm consists of shallow lateral subsurface flow, with an exponential decline in $K_{v0}$ with depth, which may not be appropriate for simulating deep groundwater. While for many applications deep groundwater processes can be ignored, for the coupling with a groundwater model like MODFLOW or for the extraction of groundwater as part of the foreseen water demand and allocation developments, implementation of a deep groundwater concept is important. Finally, speedup of the wflow code is ongoing work: recently multithreading (single node) was added to the wflow code, and further developments may include distributed computing (using for example Julia's implementation Distributed.jl or the Julia interface to message passing interface (MPI) MPI.jl) and graphics processing unit (GPU) acceleration. In view of these future developments and the current status of the Wflow.jl framework, we have developed the wflow_sbm model, which is applicable worldwide and serves as an important tool to provide relevant information for operational and water resource planning challenges.

## Appendix A

**Table A1.** Wflow_sbm state and flux variables (non-exhaustive).

| Symbol | Description | Unit | Wflow.jl name |
|---|---|---|---|
| $S_{\text{canopy}}$ | Canopy storage | mm | canopystorage |
| $S_{\text{snow}}$ | Snow storage | mm | snow |
| $S_{\text{snow,liquid}}$ | Amount of liquid water in the snowpack | mm | snowwater |
| $S_{\text{glacier}}$ | Glacier storage | mm | glacierstore |
| $S_{\text{unsat},n}$ | Amount of water in the unsaturated zone, for layer $n$ | mm | ustorelayerdepth |
| $S_{\text{sat}}$ | Amount of water in saturated store | mm | satwaterdepth |
| $S_{\text{res}}$ | Storage of reservoir | m$^3$ | volume |
| $S_{\text{lake}}$ | Lake storage | m$^3$ | storage |
| $H_{\text{lake}}$ | Water level lake | m | waterlevel |
| $P$ | Precipitation | mm $t^{-1}$ | precipitation |
| $I_{\text{total}}$ | Total interception | mm $t^{-1}$ | interception |
| $P_{\text{throughfall}}$ | Throughfall | mm $t^{-1}$ | throughfall |
| $F_{\text{available}}$ | Infiltration rate of available water | mm $t^{-1}$ | avail_forinfilt |
| $F_{\text{excess}}$ | Infiltration excess water rate | mm $t^{-1}$ | infiltexcess |
| $F_{\text{excess, sat}}$ | Rate of water that cannot infiltrate due to saturated soil | mm $t^{-1}$ | waterexcess |
| $F_{\text{act}}$ | Actual infiltration rate | mm $t^{-1}$ | actinfilt |
| $R_{\text{exfilt, sat}}$ | Water exfiltrating under saturation excess conditions | m $t^{-1}$ | exfiltwater |
| $R_{\text{exfilt, unsat}}$ | Water exfiltrating from unsaturated store because of change in water table | mm $t^{-1}$ | exfiltustore |
| $R_{\text{river}}$ | Runoff from river fraction | mm $t^{-1}$ | runoff_river |
| $R_{\text{open water}}$ | Runoff from open-water fraction (excluding rivers) | mm $t^{-1}$ | runoff_land |
| $E_{\text{open water}}$ | Evaporation from open waterbodies (excluding rivers) | mm $t^{-1}$ | ae_openw_l |
| $E_{\text{river}}$ | Evaporation from rivers | mm $t^{-1}$ | ae_openw_r |
| $E_{\text{act, sat}}$ | Soil evaporation from the saturated store | mm $t^{-1}$ | soilevapsat |
| $E_{\text{act, soil}}$ | Soil evaporation from the unsaturated store | mm $t^{-1}$ | – |
| $E_{\text{trans, sat}}$ | Transpiration from the saturated store | mm $t^{-1}$ | actevapsat |
| $\sum_{n=1}^{m} E_{\text{trans, unsat},n}$ | Transpiration from the unsaturated store for $m$ unsaturated soil layers | mm $t^{-1}$ | ae_ustore |
| $C_{\text{act}}$ | Actual capillary rise | mm $t^{-1}$ | actcapflux |
| $L$ | Leakage | mm $t^{-1}$ | actleakage |
| $R_{\text{input}}$ | Net recharge to the saturated store | mm $t^{-1}$ | recharge |
| $Q_{\text{subsurface}}$ | Subsurface flow | m$^3$ d$^{-1}$ | ssf |
| $Q_{\text{transfer, act},m}$ | Transfer of water from unsaturated soil layer $m$ to saturated store | mm $t^{-1}$ | transfer |
| $Q$ | Surface flow in the kinematic wave | m$^3$ s$^{-1}$ | q_av* |
| $Q_{\text{in, lake}}$ | Lake inflow | m$^3$ $t^{-1}$ | inflow |
| $Q_{\text{out, lake}}$ | Lake outflow | m$^3$ $t^{-1}$ | totaloutflow |
| $P_{\text{lake}}$ | Lake average precipitation | mm $t^{-1}$ | precipitation |
| $E_{\text{lake}}$ | Lake average evaporation | mm $t^{-1}$ | evaporation |
| $Q_{\text{in, res}}$ | Reservoir inflow | m$^3$ $t^{-1}$ | inflow |
| $Q_{\text{out, res}}$ | Reservoir outflow | m$^3$ $t^{-1}$ | totaloutflow |
| $P_{\text{res}}$ | Reservoir average precipitation | mm $t^{-1}$ | precipitation |
| $E_{\text{pot, res}}$ | Reservoir average evaporation | mm $t^{-1}$ | evaporation |

Note that variables are defined for model time step $t$. Some variables use the same Wflow.jl names to improve code readability, as they are handled in separate structs. *q_av represents the average surface flow during model time step $t$.

**Table A2.** Wflow_sbm model parameters and forcing.

| Symbol | Description | Unit | Wflow.jl name | Default value |
|---|---|---|---|---|
| $P$ | Precipitation | $mm\,t^{-1}$ | `precipitation` | – |
| $E_{pot,\,total}$ | Potential evapotranspiration | $mm\,t^{-1}$ | `potential_evaporation` | – |
| $T_{air}$ | Mean air temperature | °C | `temperature` | – |
| $z_{soil}$ | Soil depth | mm | `soilthickness` | 2000.0 |
| $\theta_s$ | Saturated soil water content | $mm\,mm^{-1}$ | $\theta_s$ | 0.6 |
| $\theta_r$ | Residual soil water content | $mm\,mm^{-1}$ | $\theta_r$ | 0.01 |
| $f_{paved}$ | Fraction of compacted soil (or paved) | – | `pathfrac` | 0.01 |
| $f_{open\,water}$ | Open-water fraction (excluding rivers) | – | `waterfrac` | 0.0 |
| $f_{river}$ | River fraction | – | `riverfrac` | – |
| $c_{infiltration,\,unpaved}$ | Infiltration capacity of non-compacted soil | $mm\,t^{-1}$ | `infiltcapsoil` | $100.0\,mm\,d^{-1}$ |
| $c_{infiltration,\,paved}$ | Infiltration capacity of compacted soil | $mm\,t^{-1}$ | `infiltcappath` | $10.0\,mm\,d^{-1}$ |
| $z_{rooting}$ | Rooting depth | mm | `rootingdepth` | 750.0 |
| $\frac{E_{sat}}{P_{sat}}$ | Gash interception model parameter | – | `e_r` | 0.1 |
| LAI | Leaf area index | $m^2\,m^{-2}$ | `leaf_area_index` | – |
| $S_{leaf}$ | Specific leaf storage | mm | `sl` | – |
| $S_{canopy,\,max}$ | Canopy storage capacity | mm | `cmax` | 1.0 |
| $f_{canopygap}$ | Canopy gap fraction | – | `canopygapfraction` | 0.1 |
| $S_{wood,\,max}$ | Storage capacity, woody parts of vegetation | mm | `swood` | – |
| $k$ | Extinction coefficient | – | `kext` | - |
| $c_{rd}$ | Model parameter controlling the sigmoid function, for the fraction of wet roots | $mm^{-1}$ | `rootdistpar` | −500.0 |
| $z_{cap,\,maxdepth}$ | Critical water depth beyond which capillary rise ceases | mm | `cap_hmax` | 2000.0 |
| $n_{cap}$ | Empirical coefficient controlling capillary rise | – | `cap_n` | 2.0 |
| $K_{v0}$ | Vertical saturated hydraulic conductivity at the soil surface | $mm\,t^{-1}$ | `kv₀` | $3000.0\,mm\,d^{-1}$ |
| $f_{Kv}$ | Scaling parameter for vertical saturated hydraulic conductivity | $mm^{-1}$ | `f` | 0.001 |
| $c_n$ | Brooks–Corey power coefficient | – | `c` | 10.0 |
| $h_b$ | Air entry value | cm | `hb` | 10.0 |
| $L_{max}$ | Maximum allowed leakage | $mm\,t^{-1}$ | `maxleakage` | $0.0\,mm\,d^{-1}$ |
| $f_{Kh0}$ | Multiplication factor applied to $K_{v0}$ (for lateral subsurface flow) | – | – | 1.0 |
| $f_{Kv,\,n}$ | Multiplication factor (correcting vertical saturated hydraulic conductivity) | - | `kvfrac` | 1.0 |
| $s_{ddf}$ | Degree-day-melt factor snow | $mm\,t^{-1}\,°C^{-1}$ | `cfmax` | $3.75653\,mm\,d^{-1}\,°C^{-1}$ |
| $s_{fall,\,T\,threshold}$ | Temperature threshold for snowfall | °C | `tt` | 0.0 |
| $s_{fall,\,T\,interval}$ | Temperature threshold interval for snowfall | °C | `tti` | 1.0 |
| $s_{melt,\,T\,threshold}$ | Temperature threshold for snowmelt | °C | `ttm` | 0.0 |
| $s_{whc}$ | Water-holding capacity of snow | – | `whc` | 0.1 |
| $g_{melt,\,T\,threshold}$ | Temperature threshold for glacier melt | °C | `g_tt` | 0.0 |
| $g_{ddf}$ | Degree-day-melt factor glacier | $mm\,t^{-1}\,°C^{-1}$ | `g_cfmax` | $3.0\,mm\,d^{-1}\,°C^{-1}$ |
| $f_{glacier}$ | Fraction covered by a glacier | – | `glacierfrac` | 0.0 |
| $S_{glacier}$ | Glacier storage | mm | `glacierstore` | 5500.0 |
| $g_{snow\,to\,firn}$ | Fraction of snow that is converted into firn | $t^{-1}$ | `g_sifrac` | $0.001\,d^{-1}$ |
| $w_{soil}$ | Weighting coefficient for near-surface soil temperature | $t^{-1}$ | `w_soil` | $0.1125\,d^{-1}$ |
| $f_{red,\,frozen}$ | Controlling infiltration reduction factor | – | `cf_soil` | 0.038 |
| $c_{land\,slope}$ | Slope of the land surface | $m\,m^{-1}$ | $\beta_l$ | – |
| $c_{river\,slope}$ | Slope of river | $m\,m^{-1}$ | `sl` | – |
| $A_{res}$ | Reservoir area | $m^2$ | `area` | – |
| $Q_{min\,req.}$ | Minimum flow requirement downstream of the reservoir | $m^3\,s^{-1}$ | `demandrelease` | – |
| $Q_{max,\,res}$ | Maximum release capacity spillway | $m^3\,s^{-1}$ | `maxrelease` | – |
| $S_{res,\,max}$ | Maximum storage of the reservoir | $m^3$ | `maxvolume` | – |
| $f_{res,\,min}$ | Target minimum storage fraction | – | `targetminfrac` | – |
| $f_{res,\,max}$ | Target maximum storage fraction | – | `targetmaxfrac` | – |
| $A_{lake}$ | Lake area | $m^2$ | `area` | – |
| $H_{0,\,lake}$ | Water level lake threshold (below this threshold no outflow occurs) | m | `threshold` | – |
| $H_{lake}$ | Water level lake | m | `waterlevel` | - |
| $\alpha_{lake}$ | Lake rating curve coefficient | $m\,s^{-1(a)},\,m^{\frac{3}{2}}\,s^{-1(b)}$ | `b` | – |
| $\beta_{lake}$ | Lake rating curve exponent | – | `e` | – |
| $n_{land}$ | Manning's roughness (overland flow) | $s\,m^{-\frac{1}{3}}$ | `n` | 0.072 |
| $n_{river}$ | Manning's roughness (river flow) | $s\,m^{-1/3}$ | `n` | 0.036 |
| $x_{river}$ | River length | m | `dl` | – |
| $w_{river}$ | River width | m | `width` | – |
| $h_{bankfull}$ | River bankfull depth | m | `bankfull_depth` | 1.0 |

Note that $t$ represents the model time step. Some parameters use the same Wflow.jl names to improve code readability, as they are handled in separate structs. [a] Unit for parabolic weir. [b] Unit for rectangular weir.

*Code and data availability.* Wflow.jl is open-source and distributed under the terms of the MIT License. The code and documentation are provided through the following GitHub repository: https://github.com/Deltares/Wflow.jl (last access: 2 April 2024 ). Wflow v0.7.3 is available through https://github.com/Deltares/Wflow.jl/releases/tag/v0.7.3 (last access: 2 April 2024), Zenodo, with the associated DOI https://doi.org/10.5281/zenodo.10495638 (van Verseveld et al., 2024), and is available as a Julia package. The wflow_sbm model cases presented in this paper are available at Zenodo with the associated DOI https://doi.org/10.5281/zenodo.10370017 (van Verseveld et al., 2023). Development and maintenance of Wflow.jl are conducted by Deltares, and we welcome contributions from external parties.

*Author contributions.* WJvV and MV wrote the majority of the source code, and JB, HB, DE, LB and MH contributed to the software development and documentation. WJvV wrote the original draft of the paper. AHW, ROI and WJvV performed the setup, analysed the model simulations and wrote Sect. 4.2. JB wrote Sect. 3 and produced Fig. 2. LB produced Fig. 1 (adapted from van Verseveld et al., 2024). All co-authors contributed to the review and editing of the original draft paper.

*Competing interests.* The contact author has declared that none of the authors has any competing interests.

ther geographical representation in this paper. While Copernicus Publications makes every effort to include appropriate place names, the final responsibility lies with the authors.

*Acknowledgements.* This work has received funding from the EU Horizon 2020 Programme for Research and Innovation, under grant agreement no 776613 (EUCP: European Climate Prediction system; https://www.eucp-project.eu, last access: 2 April 2024) and from the European High Performance Computing Joint Undertaking (JU) under grant agreement no. 955648 (ACROSS; https://www.acrossproject.eu/, last access: 2 April 2024). The research was co-funded by the Deltares Strategic Research Program "Natural Hazards and Real-Time Information".

*Financial support.* This research has been supported by Horizon 2020 (EUCP (grant no. 776613), ACROSS (grant no. 955648)) and the Deltares Strategic Research Program "Natural Hazards and Real-Time Information". TS7

*Review statement.* This paper was edited by Mauro Cacace and reviewed by two anonymous referees.

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

**Remarks from the typesetter**

TS1      Please confirm the equation.

TS2      Please give an explanation of why the ratio in Eq. (57) needs to be changed. We have to ask the handling editor for approval. Thanks.

TS3      Please give an explanation of why the equation needs to be changed. We have to ask the handling editor for approval. Thanks.

TS4      Please confirm the equation.

TS5      Please give specific instructions how the equation should be changed and provide an explanation of why the equation needs to be changed. We have to ask the handling editor for approval. Thanks.

TS6      Thank you for your feedback. Please note that the bottom is not cut off in the figure (if you zoom in you can see that it is complete).

TS7      Please confirm both Acknowledgements and Financial support sections.

TS8      Please confirm reference list entry.

TS9      Please confirm reference list entry.

TS10      Please confirm DOI.

TS11      Please confirm the title.

TS12      Please confirm addition.

TS13      Please confirm journal name.

TS14      Please confirm journal name.

TS15      Please confirm journal name.