# Peer review of "Wflow\_sbm v0.6.1, a spatially distributed hydrologic model: from global data to local applications"

_Geoscientific Model Development, 2022_

## Author Comment (AC1)

We thank the reviewer for taking the time to review our manuscript and for providing very constructive comments. Below, we give a response (in blue in 'normal' font) to the reviewer comments (included in italic for general, major and minor comments).

**General comments**

The manuscript is a model description paper that presents the Wflow\_sbm v0.6.1 hydrological model developed by Deltares. The model structure and equations are presented in detail followed by case studies of its application in various catchments across the world. The presented model has a great potential in contributing to large scale and high resolution hydrological modelling. Overall, the paper is well written, the model is presented in detail and the applications demonstrate the capability of the model to simulate major hydrological processes in different regions. I appreciate the effort of the authors to make it public and provide transparency in the model functioning. I have enjoyed the paper, but I am missing a key component when it comes to "spatially fully distributed" hydrological models, which concerns the Wflow\_sbm ability to be spatially calibrated and evaluated with gridded data (not catchment average) and its performance in representing the spatial patterns, which is the major feature of grid-based models, as compared to lumped or semi-distributed models. Therefore, I urge the authors to demonstrate the performance of their model in reproducing the spatial patterns of major hydrological processes like actual evapotranspiration, soil moisture, and terrestrial water storage and snow accumulation as global data exist to do so.

Yes, we agree that a comparison with gridded data is missing in the manuscript, while representing spatial patterns is a major feature of wflow\_sbm. In the revised version of the manuscript we propose to extend the Moselle case (4.2.4) to demonstrate the ability of wflow\_sbm to represent the spatial distribution of at least two major hydrological processes like actual evapotranspiration, soil moisture and snow.

**Specific comments**

**Major Comments**

The key strength for spatially distributed hydrological models is the ability to simulate hydrological processes in space and provide their spatial variations. I strongly recommend demonstrating that your model can be calibrated and evaluated on spatial patterns as it is becoming the state-of-the-art in this field (e.g. Dembele et al. 2020, Demirel et al. 2018, Zink et al 2018).

Demonstrating that wflow\_sbm can be calibrated and evaluated on spatial patterns is indeed missing in the manuscript. We propose to extend the Moselle case (4.2.4) to demonstrate this in the revised version of the manuscript. Since the focus of the manuscript is on a-priori model parameter estimates in combination with a manual adjustment of the multiplication factor applied to vertical saturated hydraulic conductivity, and not on fully calibrating a hydrological model, we propose to follow the same approach for demonstrating the ability of wflow\_sbm to represent the spatial distribution of at least two major hydrological processes like actual evapotranspiration, soil moisture and snow.

**L672-674:* Why the use of ERA5 for Europe and other products for other regions (e.g. CHIRPS for Oueme in Africa)? Were these datasets evaluated or previously found suitable for hydrological modelling in these regions?**

CHIRPS rainfall is based on merge of satellite and gauge observations. ERA5 is reanalysis data and in general performance less good where convection is important. See also Figure 6 from Beck et al. (2017), where you see that satellite information gets much higher weight than reanalysis above the

tropics/ Africa. We could also have used MSWEP, but we preferred CHIRPS (Africa, 0.05 degree) as it is readily available like ERA5 reanalysis. All our other examples are outside of the tropics.

**Minor comments**

**What are the available objective functions for model calibration? Is multivariate calibration supported by the model?**

The focus of the wflow\_sbm model as part of the Wflow.jl hydrological modelling framework is on the computations (computational engine), see also L.154-156. Therefore, we do not provide objective functions for model calibration as part of Wflow.jl. However, the Wflow.run function can be easily extendend/changed for custom use (custom model run function), for calibration purposes or for example to run wflow\_sbm in ensemble mode.

*Be consistent with the use of the term "hydrological" or "hydrologic" (see e.g. lines 1 and 783). Choose one and keep it throughout the paper.*

Good point about using the term "hydrological" or "hydrologic" consistent, we will make this consistent (using the term "hydrological") in the revised version of the manuscript.

L10-11: Mention clearly that this is the model performance for discharge.

We will change line 10-11 in the revised version of the manuscript and mention clearly that this is model performance for discharge.

L511: A variable name should not have several meanings. Here P is defined as the wetted perimeter while it refers to precipitation in Table A1. Please correct this.

Thanks for noticing this about the variable names, we will correct the variable name to  $P_W$  for wetted perimeter in the revised version of the manuscript.

**L382: is f\_canopygap time dependent? There is no exponent t in the name in Table A2.**

f\_canopygap is indeed time dependent, and for Table A2 this is also true for a couple of other variables like for example precipitation (P) and leaf area index (LAI). For Table A2 (and A1) the focus is on the variable (Symbol) used (without exponent t, as for the equations in the manuscript) in the manuscript and the corresponding Wflow.jl name (including description, unit and default value).

**Technical corrections**

Many thanks for providing these technical corrections, we will include these corrections in the revised version of the manuscript.

**References**

Beck, H. E., van Dijk, A. I. J. M., Levizzani, V., Schellekens, J., Miralles, D. G., Martens, B., and de Roo, A.: MSWEP: 3-hourly 0.25° global gridded precipitation (1979–2015) by merging gauge, satellite, and reanalysis data, Hydrol. Earth Syst. Sci., 21, 589–615, https://doi.org/10.5194/hess-21-589-2017, 2017.

---

## Author Comment (AC2)

We thank the reviewer for taking the time to review our manuscript and for providing valuable comments to further improve the manuscript. Below, we give a response (in blue in 'normal' font) to the reviewer comments (included in italic).

**Comments**

1. wflow\_sbm simulates multiple hydrological processes with different time scales using flexible temporal discretization. Modeling time steps can vary from daily to sub-daily, depending on the size and characteristics of the catchment of interest (and that's good). However, without proper disaggregation or aggregation of temporal variables, a potential problem may arise, where changing time steps can lead to unreliable model responses. Except for the sub-daily Rutter interception model, it remains unclear whether varying model time steps are available or not for most of the other hydrological components. In particular, temperature-related variables, such as soil temperature, infiltration and snow melting, may be sensitive to the selection of the model time step. For instance, the infiltration through multiple soil layers described in sub-sections 2.4.1 and 2.4.2 is a sub-daily process, which may lead to over- or under-estimation of the process if a daily time step is applied. Therefore, it is recommended to specify how different hydrological processes with varying temporal scales can be integrated within wflow\_sbm and to outline the precautions that should be taken to avoid potential drawbacks.

Good point. The Rutter interception model is indeed used for sub-daily model time steps. For kinematic wave river flow and overland flow routing sub model time steps (within the model time step) are possible (see also Lines 521-527). For the infiltration through multiple soil layers a maximum change in soil water is allowed (to prevent "overshooting") and thus smaller time steps than the wflow\_sbm model time step are possible. This information is actually missing in sub-section 2.4.2 and we will add it in the revised version of the manuscript. Additionally, we propose to add this information also to section 2.1 of the revised version of the manuscript, including precautions that should be taken (for example for the lateral subsurface flow routing component the model timestep is not yet adjusted, for certain grid size and model time step combinations this may result in loss of accuracy (also related to comment 4), and the use of an explicit scheme means that results from a daily time step model may be different from those with an hourly time step).

2. In most of the example simulations, especially in Fig. 12, significant mismatches between observations and simulations are found in the timing of flood peaks. Although not clearly evident since hydrographs are drawn for several months or a year, the peak timing differences seem to exceed several days, which is a drawback for a hydrologic model. It is uncertain whether the multiplication parameters, f\_Kh0 and f\_v0, are properly calibrated for example cases. In my view, inadequate integration of hydrologic processes with different time scales may also affect the timing errors of flood peaks.

We agree, there are mismatches between observations and simulations in the timing of hydrograph peaks. The main purpose of the first two model cases is to illustrate the model sensitivity to model parameters  $f_{Kh0}$  and  $f_{Kv}$  and how this affects the contribution of lateral subsurface and overland flow to river discharge and the water table depth dynamics, and not so much to perform a (full) calibration. This is also the case for the other simulations, based on the a-priori parameter estimation we show the wflow\_sbm performance by only changing the model parameter  $f_{Kh0}$  or  $L_{max}$ . Additionally, we make use of global available forcing datasets (except for the model case in subsection 4.2.2) for the simulations, that very likely also contribute to mismatches between observed and simulated hydrograph peaks. Finally, to get a better match it is in our view required to also

include other model parameters (for example manning's roughness) as part of a calibration procedure.

3. While improving overall model performance, considering constant groundwater loss seems to significantly reduce the volume of small to mid-size runoff events, potentially causing another problem. As shown in Fig. 14, different from observations, the runoff simulation with groundwater loss for small events before Aug 2010 is nearly zero.

Yes, the applied uniform maximum groundwater loss value of 0.6 mm/d for the catchment results in an underestimation of observed discharge before August 2009. This is not a "constant groundwater loss", but a maximum groundwater loss value. The actual groundwater loss (computed by wflow\_sbm) is controlled by the vertical hydraulic conductivity at the soil bottom and the saturated store (see also Eq. 81) and may vary spatially and in time. The maximum groundwater loss value is based on literature. Thanks for pointing this out, we propose to add this to the revised version of the manuscript, including some recommendations for possible further improvement. For example, by using a lower uniform maximum groundwater loss value (we use 15% of 4 mm/day, and this is the upper range value (10-15%) reported by Séguis et al. (2011)), or by using the discharge observations of the different stations within the catchment for deriving upstream maximum groundwater loss values within the reported range by Séguis et al. (2011).

4. While using a global data set, wflow\_sbm seems to target local applications. The size of application cases ranges between 434 and 28,000 square kilometers. For those local applications, a typical spatial resolution of 1 km may not be considered high resolution. Even for small-sized catchments, wflow\_sbm seems not to be tested for a spatial grid finer than 1 km. Please describe the potential challenges of the model when applying the high-resolution grid finer than 1 km.

The focus of the manuscript is indeed on local applications using a global dataset, however local data (that may be more accurate) can of course also be used to setup a wflow\_sbm model and wflow\_sbm has been applied to larger basins (for example the Rhine basin) and at the continental scale (Europe (not reported yet)). When applying wflow\_sbm at a higher resolution than 1 km, computational time, memory usage and data storage (depending on gridded output) will increase. Finally, as for the lateral subsurface flow routing component the model timestep is not yet adjusted (see also our response to comment 1), a high-resolution grid may result in loss of accuracy depending on the model time step and model parameters related to lateral subsurface flow (e.g. land slope and horizontal saturated hydraulic conductivity). With a daily model time step, we estimate to use a minimum grid resolution of 200 m for generally accurate lateral subsurface flow results. For more information with regard to using wflow\_sbm on different spatial resolutions, we refer to the paper by Aerts et al. (2022).

**5. Please check "groundwater loss" in the sub-section 4.2.5 is the correct expression. When the water is lost in the stream to the groundwater, it is usually called channel (transmission) loss or river water leakage.**

The term "groundwater loss" is the correct term, as it is water that is lost from the saturated store. To clarify, we will add this to sub-section 4.2.5 in the revised version of the manuscript.

**6. typos**

**Line 812: 1) improve -> 1) to improve**

Thanks for catching this typo, we will change this in the revised version of the manuscript.

**References**

Aerts, J. P. M., Hut, R. W., van de Giesen, N. C., Drost, N., van Verseveld, W. J., Weerts, A. H., and Hazenberg, P.: Large-sample assessment of varying spatial resolution on the streamflow estimates of the wflow\_sbm hydrological model, Hydrol. Earth Syst. Sci., 26, 4407–4430, https://doi.org/10.5194/hess-26-4407-2022, 2022.

Séguis, L., Kamagaté, B., Favreau, G., Descloitres, M., Seidel, J.-L., Galle, S., Peugeot, C., Gosset, M., Le Barbé, L., Malinur, F., Van Exter, S., Arjounin, M., Boubkraoui, S., Wubda, M.: Origins of streamflow in a crystalline basement catchment in a sub-humid Sudanian zone: The Donga basin (Benin, West Africa): Inter-annual variability of water budget, Journal of Hydrology, Volume 402, Issues 1–2, 2011, Pages 1-13, ISSN 0022-1694, https://doi.org/10.1016/j.jhydrol.2011.01.054.

---

## Author Response (AR1)

**Author's response to review of gmd-2022-182.**

**Title:** Wflow_sbm v0.6.1, a spatially distributed hydrologic model: from global data to local applications

**Authors:** Willem J. van Verseveld, Albrecht H. Weerts, Martijn Visser, Joost Buitink, Ruben O. Imhoff, Hélène Boisgontier, Laurène Bouaziz, Dirk Eilander, Mark Hegnauer, Corine ten Velden, and Bobby Russell.

We thank the reviewers for taking the time to review our manuscript and for providing constructive comments, which have significantly improved our manuscript. Below, we give a response (in blue in 'normal' font) to the reviewer comments (included in italic for general, major and minor comments).

During the review process, we did find two issues:

1. With the modified Rutter interception model in Wflow.jl, that resulted in an overestimation of throughfall and stemflow, see also the following reported issue on Github: https://github.com/Deltares/Wflow.jl/issues/299
2. With the solution of the Modified Puls approach for lakes, see also the following Pull Request on Github: https://github.com/Deltares/Wflow.jl/pull/323

We have fixed these issues in Wflow v0.7.3, and have used this version now throughout the revised manuscript. The modified Rutter interception model (sub-daily time step) as well as the Modified Puls approach for lakes is used in the wflow_sbm model case for the Moselle catchment (sub-section 4.2.4), and the KGE score for SMAP is a bit lower in the revised version of the manuscript (the KGE score was 0.86 and has changed to 0.83, Figure 11 at p. 39), while the KGE score for discharge did not change. Related to issue (1) we have improved the description of the computation of the stemflow fraction (p. 8, L. 211-212 and Eq. 2 (and related Eq. 1, Eq. 4), that was not correctly implemented in the code for the modified Rutter interception model, and changed accordingly Eq. (10) at p. 9. Also Eq. (19) at p. 10 was changed as part of fixing the issue with the modified Rutter interception model. For the Umealven case (sub-section 4.2.3) that includes lakes that are solved with the Modified Puls approach, the KGE scores for discharge did not change. We did also rerun the other wflow model cases, and results were not different. For the computational performance the reported run times are slightly faster with v0.7.3 (36 min (was 37 min) and 23 min (was 25 min) with 1 and 4 threads for the Rhine model (p. 25, L. 672), and for the percentage of total computational time we changed the description from >98% (valid for the Rhine catchment on one thread) to ~98% for one and multiple threads (P. 25, L. 673). Because of issue (2) we did change Eq. (112) at p. 24 in the manuscript. Finally, we slightly changed the mentioned planned developments in the abstract (p. 1, L. 15-16), description of the improved routing scheme (p. 46, L. 971-974) and planned developments (p. 46, L. 986-987) in section 5, as Wflow v0.7.3 is now the reference.

For the variables in section 2 we made a clear distinction between flux and state variables in the description and units (for example in sub-section 2.2.1, the variable $P^t_{sat,max}$ is now described as a rate variable with unit [mm/t]). Furthermore we explain in L. 169-173 (p. 7) that external flux parameters are converted to the model time step t, and that for flux variables the model time step t is implicitly embedded in the equations (e.g. for subtracting a flux variable from a storage variable, the flux variable is not multiplied by t in the equation).

We also improved the equations in sub-section 2.4.2, including that the vertical hydraulic conductivity (addition of Eq. 50) and unsaturated soil layer thickness (addition of Eq. 51) vary in time

(because of a varying water table depth). Other (small) changes that are mostly related to equations and units:

- For model time step size and base model time step size $\Delta t$ [s] and $\Delta t_b$ [s] is used.
- Rename unit day to d.
- Added description how $E^t_{sat}/P^t_{sat}$ is determined in L. 207-210.
- Split Eq. (12) of the original manuscript into two equations (Eq. (13) and Eq. (14)) for clarity.
- Added explanation why Eq. (11) is required (L. 239).
- Extended description of LAI input in L. 257-258 (static, climatology or part of forcing).
- Describe $S^{t-1}_{snow,liquid}$ in L. 288 in the same way as in L. 293 (liquid water content of snow) and add unit [mm] in L. 288.
- Add correct units to $g_{snow\,to\,ice}$ (L. 315).
- Mention for more cases when variables of an equation are previously defined (e.g. L. 389).
- Eq. (22) (for the last term/case parentheses were missing).
- L. 353-354: use [mm $t^{-1}$] for units and not [mm/day] (per day applies to external units, see also L. 169-170).
- L. 361 rename parameter $w$ to $w_{soil}$, and fix unit [$t^{-1}$].
- L. 366-367: use the term "m" unsaturated soil layers, to specify the total number of unsaturated soil layers, use this also in the equations (e.g. Eq. 43) handling unsaturated soil layers.
- Add variable $Q^t_{transfer,act,n}$ for the actual transfer of water (Eqs. 54 and 58), and rename $Q^t_{transfer,n}$ to $Q^t_{transfer,pot,n}$ (Eqs. 48 and 53).
- Add that $E^t_{act,soil,sat}$ is set at zero (L. 450-451) (single soil layer or water table not in upper soil layer). Because of this for Eq. (71) and Eq. (72) case statements are not required.
- Add unit for $c_{rd}$ parameter (L. 462).
- Add layer "n" to terms of Eq. (75).
- Remove Eq. (77), since in the updated manuscript this variable is already defined in Eq. (50).
- Fix if statement and rename parameter "m" to "$n_{cap}$" in Eq. (84), and specify when capillary rise is computed (L. 504-505).
- As part of Wflow v0.7.3 (compared to v0.6.1), for the computation of vertical saturated hydraulic conductivity at the bottom of the soil column the optional multiplication factor (to correct the vertical saturated hydraulic conductivity) is used: Eq. (85) has been changed to reflect this change.
- Fix unit for lateral subsurface flow [m3/d], L. 531.
- Add parameter $f_{ssf,\,Kv}$, the scaling parameter used for lateral subsurface flow based on $f_{Kv}$ (L. 535).
- Improve description of Eq. (87) and Eq. (88) with R [m/d] based on the SBM concept variable $R_{input}$ [mm/t].
- Split Eq. (85) of the original manuscript into two equations: Eq. (89) and Eq. (90) to improve readability. Use $\Delta t_{ssf}$ [d] for the model time step size of the lateral subsurface component.
- Fix unit Manning's coefficient in L. 575.
- Use unit [m/m] for slope (L. 592).
- Describe reservoir equations (sub-section 2.7.1) at kinematic wave time step $t_i$.
- Split up Eq. (97) in the original manuscript in to three separate equations (Eqs. 102-104) for clarity.
- Add the total reservoir inflow and outflow: Eq. (106) and Eq. (107).
- Describe lake equations (sub-section 2.7.2) at kinematic wave time step $t_i$.
- Add subscript lake to $\alpha$ and $\beta$ parameters (sub-section 2.7.2).

- Describe parameter $\alpha_{lake}$ (L. 639) and add how h is computed (L. 643).
- Add the total lake inflow and outflow: Eq. (113) and Eq. (114).
- Remove Kvz from Table 1 (this parameter is not directly estimated from data, it is computed in Wflow.jl).
- Use [mm] as units in soil depths description from Soilgrids in Tabel 1.
- Table 2: fix units for $c_{rd}$ and $g_{snow\ to\ ice}$.
- Table A1: Use "m" (number of unsaturated soil layers) for variable "Transpiration from the unsaturated store…", Rename $Q_{transfer,n}$ to $Q_{transfer,act,m}$. Add variable $Q_{out,res}$.
- Table A2: correct units for $w_{soil}$, $g_{snow\ to\ ice}$, $n_{land}$, $n_{river}$ and $\alpha_{lake}$ and use for LAI unit $[m^2\ m^{-2}]$. Switch descriptions $\alpha_{lake}$ and $\beta_{lake}$.
- Added some notes below Tables A1 and A2.

**Reviewer 1**

*General comments*

*The manuscript is a model description paper that presents the Wflow_sbm v0.6.1 hydrological model developed by Deltares. The model structure and equations are presented in detail followed by case studies of its application in various catchments across the world. The presented model has a great potential in contributing to large scale and high resolution hydrological modelling. Overall, the paper is well written, the model is presented in detail and the applications demonstrate the capability of the model to simulate major hydrological processes in different regions. I appreciate the effort of the authors to make it public and provide transparency in the model functioning. I have enjoyed the paper, but I am missing a key component when it comes to "spatially fully distributed" hydrological models, which concerns the Wflow_sbm ability to be spatially calibrated and evaluated with gridded data (not catchment average) and its performance in representing the spatial patterns, which is the major feature of grid-based models, as compared to lumped or semi-distributed models. Therefore, I urge the authors to demonstrate the performance of their model in reproducing the spatial patterns of major hydrological processes like actual evapotranspiration, soil moisture, and terrestrial water storage and snow accumulation as global data exist to do so.*

Yes, we agree that a comparison with gridded data is missing in the manuscript, while representing spatial patterns is a major feature of wflow_sbm. In the revised version of the manuscript we have added a new model case for the Rhine catchment (sub-section 4.2.5, p. 40) to demonstrate the ability of wflow_sbm to represent the spatial distribution of actual evapotranspiration and snow storage. For the actual evapotranspiration comparison, we did use The Global Land Evaporation Amsterdam Model (GLEAM) v3.7b daily actual evapotranspiration data with a spatial resolution of ~28 km (Martens et al., 2017; Miralles et al., 2011), and for snow storage we did use Sentinel-1 (S1) snow depth observations from the C-SNOW project (Lievens et al., 2019, 2022).

*Specific comments*

*Major Comments*

*The key strength for spatially distributed hydrological models is the ability to simulate hydrological processes in space and provide their spatial variations. I strongly recommend demonstrating that your model can be calibrated and evaluated on spatial patterns as it is becoming the state-of-the-art in this field (e.g. Dembele et al. 2020, Demirel et al. 2018, Zink et al 2018).*

Demonstrating that wflow_sbm can be calibrated and evaluated on spatial patterns is indeed missing in the manuscript. We did add a new wflow_sbm model case for the Rhine catchment (subsection 4.2.5, p. 40) to demonstrate this in the revised version of the manuscript. Since the focus of the manuscript is on a-priori model parameter estimates in combination with a manual adjustment of the multiplication factor applied to vertical saturated hydraulic conductivity, and not on fully calibrating a hydrological model, we did follow the same approach for demonstrating the ability of wflow_sbm to represent the spatial distribution of actual evapotranspiration and snow storage. For the actual evapotranspiration comparison, we did use The Global Land Evaporation Amsterdam Model (GLEAM) v3.7b daily actual evapotranspiration data with a spatial resolution of ~28 km (Martens et al., 2017; Miralles et al., 2011), and for snow storage we did use Sentinel-1 (S1) snow depth observations from the C-SNOW project (Lievens et al., 2019, 2022).

*L672-674: Why the use of ERA5 for Europe and other products for other regions (e.g. CHIRPS for Oueme in Africa)? Were these datasets evaluated or previously found suitable for hydrological modelling in these regions?*

CHIRPS rainfall is based on merge of satellite and gauge observations. ERA5 is reanalysis data and in general performance less good where convection is important. See also Figure 6 from Beck et al. (2017), where you see that satellite information gets much higher weight than reanalysis above the tropics/ Africa.  We could also have used MSWEP for Oueme, but we preferred CHIRPS (Africa, 0.05 degree) as it is readily available like ERA5 reanalysis. All our other examples are outside of the tropics. For the new wflow_sbm model case for the Rhine catchment we use MSWEP instead of ERA5 because GLEAM is also using MSWEP (p. 40, L. 878).

***Minor comments***

*What are the available objective functions for model calibration? Is multivariate calibration supported by the model?*

The focus of the wflow_sbm model as part of the Wflow.jl hydrological modelling framework is on the computations (computational engine), see also p.7, L. 178-179. Therefore, we do not provide objective functions for model calibration as part of Wflow.jl. However, the Wflow.run function can be easily extended/changed for custom use (custom model run function), for calibration purposes or for example to run wflow_sbm in ensemble mode.

*Be consistent with the use of the term "hydrological" or "hydrologic" (see e.g. lines 1 and 783). Choose one and keep it throughout the paper.*

Good point about using the term "hydrological" or "hydrologic" consistent, we made this consistent (using the term "hydrological") in the revised version of the manuscript.

*L10-11: Mention clearly that this is the model performance for discharge.*

We have changed line 11 in the revised version of the manuscript to show clearly that this is model performance for discharge: "… good (KGE ≥ 0.7) performance for discharge a-priori (without further tuning)."

*L511: A variable name should not have several meanings. Here P is defined as the wetted perimeter while it refers to precipitation in Table A1. Please correct this.*

Thanks for noticing this about the variable names, we did correct the variable name to $P_W$ for wetted perimeter in the revised version of the manuscript (p. 21, L. 573 (Eq. 94), L. 574, L. 575, L. 577).

*L382: is f_canopygap time dependent? There is no exponent t in the name in Table A2.*

f_canopygap can indeed be time dependent (when using leaf area index (LAI) climatology or as part of forcing), and for Table A2 this is also true for a couple of other variables like for example precipitation (P) and LAI. For Table A2 (and A1) the focus is on the variable (Symbol) used (without exponent t, as for the equations in the manuscript) in the manuscript and the corresponding Wflow.jl name (including description, unit and default value).

***Technical corrections***

*L73: "most gauging" ---> "most discharge gauging"*

Changed (p. 3, L. 74).

*L127: "then" ---> "than"*

Changed (p.5, L. 128).

*L135: "water when" ---> "water occurs when"*

Changed (p. 5, L. 136).

*L136: "in" ---> "from"*

Changed (p. 5, L. 137).

*L137: "river" ---> "river routing"*

Changed (p. 5, L. 138).

*L155: "extendig" ---> "extending"*

Changed (p. 7, L. 179).

*L203: delete the duplicated "is"*

Deleted (p. 9, L. 233).

*L348, 366, 383, 388, 396: "bucket" ---> "unsaturated soil bucket "*

The "bucket" in wflow_sbm refers to a soil column with a certain depth that contains an unsaturated zone as well as a saturated zone (depending on the water depth in the soil column). Because of this, we did not change "bucket" to "unsaturated soil bucket ", but to "soil bucket" in  lines 382, 412, 431, 436, 446.

*L564: "expresses" ---> "expressed"*

Changed (p. 24, L. 641).

*L569: In the second part of the equation 103 (i.e. if SI_lake <=…), "A" should be "A_lake" in the numerator of the fraction.*

Changed this in equation 112 (p. 24, L. 646).

*L585: Give the definition of "I/O".*

Definition is added (p. 25, L. 666).

*L607: "orginal" ---> "original"*

Changed (p. 27, L. 688).

*L681, 704, 726, 755: "avalance" ---> "avalanche"*

Changed "avalance" to "avalanche" in lines 776, 800, 822, 851.

*L720: "hydrometeorlogical" ---> "hydrometeorological"*

Changed (p. 36, L. 816).

*L733: "1630" ---> "station 1630"*

Changed (p. 37, L. 829).

*L774: "yearly average rainfall" ---> "annual average daily rainfall"*

Changed (p. 43, L. 920).

*L776: "disables" ---> "disabled"*

Changed (p. 43, L. 924).

*L776: "scores" ---> "scores of discharge"*

Changed (p. 43, L. 924).

*L800: "function" ---> "functions"*

Changed (p. 45, L. 961).

*Figure 11: In the caption: "average" ---> "catchment-average"*

We have changed this in the caption of Figure 11 (p. 39).

*Figure 13: In the caption: "scores" ---> "scores of discharge"*

We have changed this in the caption of Figure 16 (p. 44).

*Table A1: No Wflow.jl name for E_act,soil?*

Yes, correct, the variable E_act,soil is not available as a field of the vertical SBM struct in Wflow.jl (only internal).

*Table A1: Should the unit of R_input be mm/t?*

Yes, indeed R_input should be in  mm/t, we have changed this in Table A1.

*Table A1: Should the unit of Q_in,res be m3/t?*

Yes, thanks for noting this, we changed the unit in Table A1 for Q_in,res.

*Table A2: It would be better to separate the list of the parameters from the forcing.*

We think it is fine to keep the forcing variables in Table A2, these variables are also stored as part of the vertical SBM struct in Wflow.jl.

*Table A2: Is this mean air temperature? Say it explicitly.*

Yes, we have changed this to mean air temperature in Table A2.

*Table A2: For f_paved, should read "compacted" in the description.*

Thanks for catching this, we have changed it to "compacted" in Table A2.

*Table A2: Add f_river to the table.*

Good point, we have added f_river to Table A2.

**Reviewer 2**

**Comments**

*1. wflow_sbm simulates multiple hydrological processes with different time scales using flexible temporal discretization. Modeling time steps can vary from daily to sub-daily, depending on the size and characteristics of the catchment of interest (and that's good). However, without proper disaggregation or aggregation of temporal variables, a potential problem may arise, where changing time steps can lead to unreliable model responses. Except for the sub-daily Rutter interception model, it remains unclear whether varying model time steps are available or not for most of the other hydrological components. In particular, temperature-related variables, such as soil temperature, infiltration and snow melting, may be sensitive to the selection of the model time step. For instance, the infiltration through multiple soil layers described in sub-sections 2.4.1 and 2.4.2 is a sub-daily process, which may lead to over- or under-estimation of the process if a daily time step is applied. Therefore, it is recommended to specify how different hydrological processes with varying temporal scales can be integrated within wflow_sbm and to outline the precautions that should be taken to avoid potential drawbacks.*

Good point. The Rutter interception model is indeed used for sub-daily model time steps. For kinematic wave river flow and overland flow routing sub model time steps (within the model time step) are possible (see also lines 584-589, p. 22). We have added an extra paragraph to section 2.1 (p. 6-7, L. 155-173) that describes how the equations of the hydrological processes are solved and whether varying internal time steps (within the model time step) are possible, including precautions that should be taken. We have also added information in sub-section 2.4.2 (p. 15, L. 409-411) about internal time stepping for the unsaturated flow.

*2. In most of the example simulations, especially in Fig. 12, significant mismatches between observations and simulations are found in the timing of flood peaks. Although not clearly evident since hydrographs are drawn for several months or a year, the peak timing differences seem to exceed several days, which is a drawback for a hydrologic model. It is uncertain whether the multiplication parameters, f_Kh0 and f_v0, are properly calibrated for example cases. In my view, inadequate integration of hydrologic processes with different time scales may also affect the timing errors of flood peaks.*

We agree, there are mismatches between observations and simulations in the timing of hydrograph peaks. The main purpose of the first two model cases is to illustrate the model sensitivity to model parameters $f_{Kh0}$ and $f_{Kv}$ and how this affects the contribution of lateral subsurface and overland flow to river discharge and the water table depth dynamics, and not so much to perform a (full) calibration. This is also the case for the other simulations, based on the a-priori parameter estimation we show the wflow_sbm performance by only changing the model parameter $f_{Kh0}$ or $L_{max}$. Additionally, we make use of global available forcing datasets (except for the model case in sub-section 4.2.2) for the simulations, that very likely also contribute to mismatches between observed and simulated hydrograph peaks. Finally, to get a better match it is in our view required to also include other model parameters (for example manning's roughness) as part of a calibration procedure.

*3. While improving overall model performance, considering constant groundwater loss seems to significantly reduce the volume of small to mid-size runoff events, potentially causing another problem. As shown in Fig. 14, different from observations, the runoff simulation with groundwater loss for small events before Aug 2010 is nearly zero.*

Yes, the applied uniform maximum groundwater loss value of 0.6 mm/d for the catchment results in an underestimation of observed discharge before August 2009. This is not a "constant groundwater loss", but a maximum groundwater loss value. The actual groundwater loss (computed by wflow_sbm) is controlled by the vertical hydraulic conductivity at the soil bottom and the saturated store (see also Eq. 85) and may vary spatially and in time. The maximum groundwater loss value is based on literature. Thanks for pointing this out, we have added this to the revised version (p. 43, L. 921-923) of the manuscript, including some recommendations for possible further improvement (p.43-44, L. 930-935).

*4. While using a global data set, wflow_sbm seems to target local applications. The size of application cases ranges between 434 and 28,000 square kilometers. For those local applications, a typical spatial resolution of 1 km may not be considered high resolution. Even for small-sized catchments, wflow_sbm seems not to be tested for a spatial grid finer than 1 km. Please describe the potential challenges of the model when applying the high-resolution grid finer than 1 km.*

The focus of the manuscript is indeed on local applications using a global dataset, however local data (that may be more accurate) can of course also be used to setup a wflow_sbm model and wflow_sbm has been applied to larger basins (for example the Rhine basin (now added as a model case to the revised version of the manuscript, see also sub section 4.2.5)) and at the continental scale (Europe; not reported yet). When applying wflow_sbm at a higher resolution than 1 km, computational time, memory usage and data storage (depending on gridded output) will increase. Finally, as for the lateral subsurface flow routing component the model timestep is not yet adjusted (see also our response to comment 1), a high-resolution grid may result in loss of accuracy depending on the model time step and model parameters related to lateral subsurface flow (e.g. land slope and horizontal saturated hydraulic conductivity). With a daily model time step, we estimate to use a minimum grid resolution of 200 m for generally accurate lateral subsurface flow results and have added this in lines 166-167 (p. 7). For more information with regard to using wflow_sbm on different spatial resolutions, we refer to the paper by Aerts et al. (2022).

*5. Please check "groundwater loss" in the sub-section 4.2.5 is the correct expression. When the water is lost in the stream to the groundwater, it is usually called channel (transmission) loss or river water leakage.*

The term "groundwater loss" is the correct term, as it is water that is lost from the saturated store. To clarify, we have added this to sub-section 4.2.6 in the revised version of the manuscript (p. 43, L. 921-923).

*6. typos*

*Line 812: 1) improve -> 1) to improve*

Thanks for catching this typo, we have changed this in the revised version of the manuscript (p. 46, L. 974).

**References**

Aerts, J. P. M., Hut, R. W., van de Giesen, N. C., Drost, N., van Verseveld, W. J., Weerts, A. H., and Hazenberg, P.: Large-sample assessment of varying spatial resolution on the streamflow estimates of the wflow_sbm hydrological model, Hydrol. Earth Syst. Sci., 26, 4407–4430, https://doi.org/10.5194/hess-26-4407-2022, 2022.

Beck, H. E., van Dijk, A. I. J. M., Levizzani, V., Schellekens, J., Miralles, D. G., Martens, B., and de Roo, A.: MSWEP: 3-hourly 0.25° global gridded precipitation (1979–2015) by merging gauge, satellite, and reanalysis data, Hydrol. Earth Syst. Sci., 21, 589–615, https://doi.org/10.5194/hess-21-589-2017, 2017.

Lievens, H., Brangers, I., Marshall, H.-P., Jonas, T., Olefs, M., and De Lannoy, G. J. M.: Sentinel-1 snow depth retrieval at sub-kilometer resolution over the European Alps, The Cryosphere, 16, 159-177, 2022.

Lievens, H., Demuzere, M., Marshall, H.-P., Reichle, R.H., Brucker, L., Brangers, I., de Rosnay, P., Dumont, M., Girotto, M., Immerzeel, W.W., Jonas, T., Kim, E.J., Koch, I., Marty, C., Saloranta, T., Schöber J., and De Lannoy, G.J.M.: Snow depth variability in the Northern Hemisphere mountains observed from space, Nature Communications, 10, 4629, 2019.

Martens, B., Miralles, D.G., Lievens, H., van der Schalie, R., de Jeu, R.A.M., Fernández-Prieto, D., Beck, H.E., Dorigo, W.A., and Verhoest, N.E.C.: GLEAM v3: satellite-based land evaporation and root-zone soil moisture, Geoscientific Model Development, 10, 1903–1925, doi: 10.5194/gmd-10-1903-2017, 2017.

Miralles, D.G., Holmes, T.R.H., de Jeu, R.A.M., Gash, J.H., Meesters, A.G.C.A., Dolman, A.J.: Global land-surface evaporation estimated from satellite-based observations, Hydrology and Earth System Sciences, 15, 453–469, doi: 10.5194/hess-15-453-2011, 2011.